# Extensive germline-somatic interplay contributes to prostate cancer progression through HNF1B co-option of TMPRSS2-ERG

Nikolaos Giannareas[1,7], Qin Zhang[1,7], Xiayun Yang[1,7], Rong Na [2,7], Yijun Tian[3], Yuehong Yang[1], Xiaohao Ruan[4], Da Huang[4], Xiaoqun Yang[5], Chaofu Wang[5], Peng Zhang[6], Aki Manninen [1], Liang Wang [3] & Gong-Hong Wei [1,6] ✉

Genome-wide association studies have identified 270 loci conferring risk for prostate cancer (PCa), yet the underlying biology and clinical impact remain to be investigated. Here we observe an enrichment of transcription factor genes including HNF1B within PCa risk-associated regions. While focused on the 17q12/HNF1B locus, we find a strong eQTL for HNF1B and multiple potential causal variants involved in the regulation of HNF1B expression in PCa. An unbiased genome-wide co-expression analysis reveals PCa-specific somatic TMPRSS2-ERG fusion as a transcriptional mediator of this locus and the HNF1B eQTL signal is ERG fusion status dependent. We investigate the role of HNF1B and find its involvement in several pathways related to cell cycle progression and PCa severity. Furthermore, HNF1B interacts with TMPRSS2-ERG to co-occupy large proportion of genomic regions with a remarkable enrichment of additional PCa risk alleles. We finally show that HNF1B co-opts ERG fusion to mediate mechanistic and biological effects of the PCa risk-associated locus 17p13.3/VPS53/FAM57A/GEMIN4. Taken together, we report an extensive germline-somatic interaction between TMPRSS2-ERG fusion and genetic variations underpinning PCa risk association and progression.

Genome-wide association studies (GWASs) have successfully discovered thousands of risk-associated single nucleotide polymorphism (SNP) loci for cancers of prostate and others. While most SNP associations are cancer-specific, roughly one-third of such genomic loci with SNPs associates with multiple types of cancers, namely pleiotropic loci that may have shared mechanisms or hallmarks across cancers[1]. Due to the complexity of human genome, the vast majority of GWAS-identified SNPs fell within non-coding genomic regions that have been proven to possess regulatory functions in modulating gene expression through diverse mechanisms[2–6]. Growing evidence indicate that GWAS loci are often involved in expression quantitative trait locus (eQTL) conferring cancer risk via altering the DNA-binding affinity of critical transcription factors to causal SNP-containing regulatory elements such as enhancers, representing a major driving force of cancerous gene expression program[2,5,7]. However, it remains challenging to define somatic driver transcription factors, target genes and aberrant biological pathways from GWAS discoveries, thereby depicting the molecular mechanisms of

[1]Disease Networks Research Unit, Faculty of Biochemistry and Molecular Medicine & Biocenter Oulu, University of Oulu, Oulu, Finland. [2]Division of Urology, Department of Surgery, Li Ka Shing Faculty of Medicine, the University of Hong Kong, Hong Kong, China. [3]Department of Tumour Biology, H. Lee Moffitt Cancer Center and Research Institute, Tampa, FL, USA. [4]Department of Urology, Ruijin Hospital, Shanghai Jiaotong University School of Medicine, Shanghai, China. [5]Department of Pathology, Ruijin Hospital, Shanghai Jiaotong University School of Medicine, Shanghai, China. [6]Fudan University Shanghai Cancer Center & MOE Key Laboratory of Metabolism and Molecular Medicine and Department of Biochemistry and Molecular Biology of School of Basic Medical Sciences, Shanghai Medical College of Fudan University, Shanghai, China. [7]These authors contributed equally: Nikolaos Giannareas, Qin Zhang, Xiayun Yang, Rong Na. ✉e-mail: gonghong_wei@fudan.edu.cn

germline–somatic interplay and continuum underlying these cancer risk loci.

Prostate cancer (PCa) remains the second most common cancer in men and globally affects millions of individuals. It was estimated that more than 1.41 million men have been diagnosed with PCa and 375,000 PCa-associated deaths occurred worldwide in 2020[8]. PCa has complex etiology and a high estimate of heritability at 57%[9]. Therefore, unraveling PCa risk-associated genetic factors and investigating their underlying mechanisms and biological impacts are expected to greatly inform our understanding of PCa pathogenesis, progression, and clinical management. Comprehensive GWAS analyses have been performed in men with PCa across diverse ancestry groups, and these studies have jointly identified 270 susceptibility loci harboring over 400 SNPs that reach genome-wide significance ($P \leq 5 \times 10^{-8}$) in association with PCa risk and aggressiveness[10–13]. Of these PCa risk loci, the 17q12/HNF1B locus variants rs4430796, rs11263763 and rs11651052 have been reproducibly found to be associated with PCa susceptibility[14–21]. Moreover, SNPs in the 17q12/HNF1B region have also been reported in association with risk of several other cancer types, including pancreatic[22], ovarian[23,24], testicular[25], and endometrial[26] cancers. Yet, the mechanistic effects and underlying biology of the 17q12/HNF1B remain elusive for all of these associated cancer types. HNF1B as a 17q12 locus gene, belongs to HNF1 transcription factor family whose members possess a POU-homeodomain responsible for sequence-specific DNA binding to gene cis-regulatory elements (CRE)[14,27]. HNF1B has been reported to play roles in tumourigenesis and is used as a biomarker for clear cell carcinomas of the pancreas[28], colorectal cancer[29] and endometrial carcinoma[30], implying that HNF1B is a plausible causative gene for this pleiotropic association.

Results from our unbiased genome-wide co-expression and ChIP-seq analysis reveal that TMPRSS2-ERG is likely to be an effective transcription regulator of the 17q12/HNF1B PCa risk locus. TMPRSS2-ERG is the most frequent somatic fusion event in PCa, involving a chromosomal rearrangement of ERG transcription factor hijacking the 5′ androgen-responsive regulatory region of TMPRSS2 to form a constitutively activated mutant TMPRSS2-ERG fusion protein[31,32]. The frequency of ERG fusion is variable among different ethnic groups, showing the highest frequency (>50%) in Caucasians, followed by African American (20%-30%) and Asian men (<20%)[33–36]. Aberrant ERG fusion transcription factor influences various pathways and biological processes such as androgen receptor (AR) signaling, transforming growth factor beta 1 (TGF-β) signaling and cell invasion[37,38]. While the association of TMPRSS2-ERG fusion status with PCa clinical outcomes remains inconclusive[39], some reports have demonstrated the role of ERG fusion in PCa cellular growth and tumor progression. Studies have shown that knockdown of ERG inhibits cell proliferation, invasion and xenograft tumor growth of TMPRSS2-ERG-positive PCa cell line VCaP[40] while ERG overexpression lead to PCa precursor-like lesions in mice[41] and promoted cell invasion in vitro[40]. In addition, cooperation of prostate-specific expression of ERG and genetic activation of the PI3K/AKT pathway or loss of PTEN drive PCa progression in mouse models[42,43]. However, given the high rate of TMPRSS2-ERG genomic translocation in PCa, it remains to be investigated whether and how this somatic fusion is involved in germline risk loci for PCa discovered by GWASs.

In this work, employing an unbiased enrichment analysis, we show that the transcription factor genes, such as HNF1B, are greatly enriched nearby known PCa susceptibility loci, implying a hypothesis that an interconnected gene regulatory network by core genes, transcription factors in particular, may explain subtle effects of GWAS-discovered SNPs on complex diseases and traits. We therefore focus on the 17q12/HNF1B locus and seek to identify functional causal variants of this region as well as investigate role of HNF1B in PCa progression and predisposition to aggressive disease, which in turn may be an example for functional study of this cross-cancer pleiotropic genetic

association in other types of cancers. We show that TMPRSS2-ERG is a transcription regulator mediating mechanistic effects of the 17q12 locus and control the expression of HNF1B, which in turn regulates a set of cell cycle genes implicating PCa predisposition and progression. We find that HNF1B and TMPRSS2-ERG physically interact with each other, have a remarkable chromatin co-occupancy and cooperatively regulate genes implicated in PCa development. Lastly, we observe that common binding sites for HNF1B and TMPRSS2-ERG can explain more of genetic effects on PCa predisposition than that of their unique binding sites, and present a solid example of HNF1B co-option of TMPRSS2-ERG co-regulating the 17p13.3 PCa risk locus.

## Results

### Transcription factor genes including HNF1B are markedly enriched in PCa susceptibility loci

Transcription factors direct the chromatin binding to cis-acting regulatory DNA elements (CREs) and play central roles in gene expression networks[44,45]. We thereby examined whether the genes encoding transcription factors are more likely to be enriched in PCa susceptibility loci and thus extract nearby genes of each PCa GWAS variant for a statistical examination. This analysis revealed that transcription factor genes are indeed preferentially to be enriched in PCa risk loci ($P = 2.49 \times 10^{-6}$, hypergeometric distribution test; Fig. 1a), including many with poorly characterized causal roles (Fig. 1b), except for HOXB13, RFX6, and NKX3-1 that have been shown to be causally linked to PCa susceptibility and tumourigenesis[46–49]. Among these functionally uncharacterized PCa risk loci, the 17q12/HNF1B locus with several independent SNPs has been reproducibly found in association with PCa susceptibility[14–21], and interestingly the phenome-wide association analysis (PheWAS) using FinnGen cohort data ($n = 176,899$) observed a specific top-ranked association of the 17q12/HNF1B locus variants with PCa across 2,264 disease endpoints (Fig. 1c). Together with other observations describing 17q12/HNF1B in association with multiple types of cancer[22–26], this finding prompted us to delve into the molecular and biological mechanisms as well as the clinical implications of this locus, and the regulation of HNF1B in PCa.

We first evaluated the expression profiles of HNF1B across 31 cancer types using RNA-seq data of The Cancer Genome Atlas (TCGA) and found that HNF1B was highly expressed in approximately top 25% of cancer types including PCa (Supplementary Fig. 1a). To test the biological relevance of HNF1B in PCa, we performed a cell proliferation assay and found that several PCa cell lines including LNCaP, VCaP, DU145, and PC3 harboring siRNAs against *HNF1B* displayed reduced cell growth compared to cells harboring control siRNA (Supplementary Fig. 1b–i). In line with this observation, downregulation of HNF1B via lentivirus-mediated short hairpin RNA (shRNA) greatly attenuated cell proliferation and migration in the PCa V16A cell line while ectopic expression of HNF1B elevated cellular proliferation of RWPE1 cells (Fig. 1d–f and Supplementary Fig. 1j). Moreover, the data from a genome-wide CRISPR-mediated loss-of-function screen in the PCa cell line VCaP[50] demonstrated that HNF1B is a top-ranked essential gene for cell survival (Fig. 1g), further indicating the importance of HNF1B in PCa cell growth and survival.

To investigate the functional and clinical relevance of HNF1B in PCa, we analyzed multiple independent PCa expression profile datasets[51–56]. The results showed that HNF1B expression is greatly elevated in primary and metastatic prostate tumors compared to normal prostate gland (Fig. 1h, i and Supplementary Fig. 1k, l). Moreover, we observed that the mRNA levels of HNF1B are greatly increased in localized or metastatic PCa samples and in tumors of higher grade and Gleason score (Fig. 1j, k), suggesting a potential function for HNF1B in advanced prostate tumors. Together, these findings illustrated a potential role for HNF1B in PCa severity and progression, indicating that HNF1B is a plausible causative gene underlying the effects of the 17q12 PCa susceptibility locus variants.

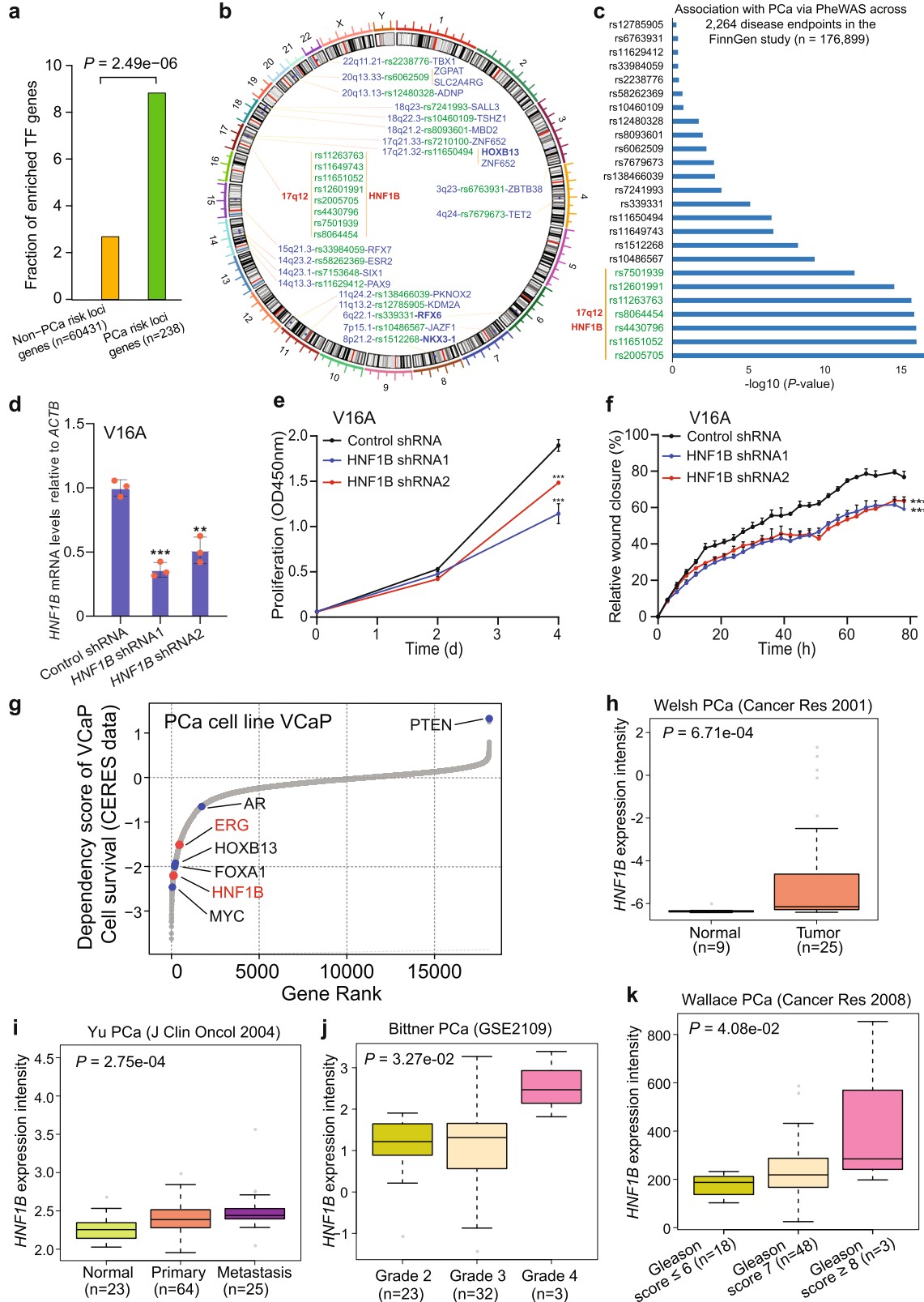

### The 17q12/HNF1B PCa risk locus harbors a single risk-associated gene and multiple causal variants

To identify functional causal variants at *HNF1B* locus, we screened a total of 13 SNPs highly associated with multiple PCa risk variants at 17q12 using the expression quantitative trait locus (eQTL) data from Wisconsin cohort of 466 prostate normal tissue samples[57–59]. We first examined the mRNA levels of different *HNF1B* isoforms and found that

*HNF1B* isoform 1 is dominantly expressed compared to the other two isoforms in prostate tissues (Supplementary Fig. 2a). The eQTL analysis revealed a strong significant association between *HNF1B* isoform 1 and 11 of the 13 SNPs (Fig. 2a and Supplementary Fig. 2b–o). Remarkably, further cis-eQTL in this Wisconsin cohort revealed the strongest association of *HNF1B* among all genes within one mega-base window with any of these 11 SNPs (Fig. 2b and Supplementary Data 1). We next

**Fig. 1 | Transcription factor genes are highly enriched nearby PCa risk loci and the 17q12 HNF1B is associated with PCa cell growth and tumor progression.**
**a** Transcription factor genes are more likely to be enriched in PCa susceptibility loci (*n* = 60669). *P* value was evaluated by the two-sided Fisher's exact test. **b** Circos visualization of enriched transcription factor genes and relevant PCa risk loci. **c** Phenome-wide association analysis (PheWAS) for an unbiased examination of potential associations between the transcription factor gene enriched PCa risk loci and 2,264 disease endpoints in the FinnGen study (*n* = 176,899). *P* values were obtained from the PheWAS study, and the genome-wide significance threshold was defined as $P = 5 \times 10^{-8}$. The y-axis indicates the SNPs of the loci described in **b**. Note that the 17q12 alleles are prevalent in association with PCa in this PheWAS assessment. **d** Depletion of *HNF1B* in V16A through lentivirus-mediated shRNA interference. *n* = 3 samples; *P* values based on the order of appearance: 2E–04, 2E–03. **e** *HNF1B* knockdown reduces PCa cell proliferation. *n* = 3 samples; *P* values HNF1B siRNA1: 5,4E–04, siRNA2: 4,9E–04. **f** Wound healing assay in the V16A cells infected with lentiviruses expressing shRNAs targeting *HNF1B*. *n* = 3 samples; *P* values HNF1B siRNA1: 2,3E–04, siRNA2: 4,1E–04. **g** Genome-wide loss-of-function screen in VCaP identified essential genes including *HNF1B* and *ERG* for cell survival. Lower scores indicate higher dependency on the gene for cell viability. *AR*, *HOXB13*, *FOXA1*, and *MYC* are well-known genes driving PCa cell proliferation and survival whereas tumor suppressor *PTEN* does not favor PCa cell growth and survival. *HNF1B* is highly expressed in primary prostate tumors (**h**, *n* = 34) and metastases (**l**, *n* = 112) compared to normal tissues. In (**h**), *P* value was assessed by two-sided Mann–Whitney U test. *HNF1B* expression level is increased in high-grade tumors indicated by higher clinical Grade (**j**, *n* = 58) and Gleason scores (**k**, *n* = 69). In (**i**–**k**), *P* values were evaluated by Kruskal-Wallis test. In (**d**–**f**), *n* = 3 technical replicates, error bars, mean ± SD, ** *P* < 0.01, *** *P* < 0.001, *P* values were evaluated using the two-tailed Student's *t* tests. In h-k, the interquartile range (IQR) is depicted by the box with the median represented by the center line. Whiskers maximally extend to 1.5 × IQR (with outliers shown). Source data are provided in Source Data file.

performed CRISPR/Cas9-mediated genome editing in the PCa cell lines 22Rv1 and V16A to delete each of the 13 SNP-containing regions and observed great downregulation of HNF1B in most of mixed clones with knockout (KO) of individual SNP-region (Fig. 2c and Supplementary Fig. 2p). To test whether the SNPs residing in *HNF1B* locus have an enhancer variant-like function in regulating *HNF1B*, we examined six SNP regions based on the above CRISPR/Cas9-mediated KO screening. Each SNP region with two possible alleles was cloned into upstream of *HNF1B* promoter in luciferase reporter constructs in both orientations. Enhancer report assays in 22Rv1 and LNCaP cells showed that rs11651052, rs12453443, rs9901746, rs7405696 and rs11263763 regions indicate enhancer-like function to activate luciferase gene expression compared to the HNF1B promoter (Fig. 2d and Supplementary Fig. 2q). Specifically in 22Rv1 cells, the rs9901746 region indicated an orientation-dependent effect on the enhancer activity with 3′–5′ orientation in both alleles. In addition, several SNP regions showed enhancer activity in the presence of specific alleles and, in particular the G and A alleles of rs7405696 and rs11651052, respectively, displayed stronger activity than the C alleles of rs7405696 or the allele G of rs11651052 towards HNF1B promoter in regulating luciferase gene expression (Fig. 2d). To further verify CRISPR/Cas9 screen and the enhancer report assays, we perform genomic deletion of those SNP-enhancer regions through CRISPR/Cas9-mediated KO at single cell levels in the PCa 22Rv1 cells. We observed a profound downregulation of *HNF1B* in every clone in which SNP-enhancer regions were deleted (Fig. 2e). These results pointed to HNF1B as a PCa risk-associated gene and defined several potential causal variants that are likely to be involved in the regulation of *HNF1B* expression at 17q12 locus.

### Somatic TMPRSS2-ERG fusion as a regulator of the 17q12/HNF1B PCa risk locus

Accumulating evidence indicates that GWAS loci are often involved in key transcription factors acting through genomic regulatory elements like enhancers to drive eQTL gene expression[2–6,46,60], and given eQTL genes often display co-expression with the responsible transcription factors in the clinical settings[46,60]. We thus performed a genome-wide co-expression analysis of *HNF1B* in several large clinical PCa datasets to figure out potential transcription factors that are regulating *HNF1B* via the causal variant containing regions. Intriguingly, ERG emerged as the most positively co-expressed gene with *HNF1B* in TCGA cohort comprised of 497 primary PCa tumors (Fig. 3a, b and Supplementary Fig. 3a). *ERG* was also found as a top-ranking gene showing positive expression correlation with *HNF1B* in another cohort of 264 PCa tumors (Supplementary Fig. 3b). By contrast, *ERG* exhibited no apparent expression correlation with HNF1B in normal prostate tissues (Supplementary Fig. 3c–e), suggesting that this association is specific to human prostate tumorigenesis. As mentioned above, a common somatic genomic rearrangement involves *TMPRSS2* and *ERG* genes that form an androgen signaling-regulated mutant transcription factor,

*TMPRSS2-ERG*[31,61–63]. We next assessed whether *ERG* contributes to the regulation of *HNF1B* expression by using siRNA-mediated knockdown assay in VCaP cells that harbors a *TMPRSS2-ERG* fusion[31] and LNCaP cell line negative to the fusion. This analysis revealed that depletion of *TMPRSS2-ERG* resulted in decreased mRNA levels of *HNF1B* in VCaP but not in LNCaP cells (Fig. 3c and Supplementary Fig. 3f).

To understand the regulatory mechanisms at *HNF1B* locus, we explored a large collection of genome-wide chromatin immunoprecipitation sequencing (ChIP-seq) data derived from PCa[64]. We found an enrichment of epigenetic marks (H3K4me1/2 and H3K27ac) for active enhancers and chromatin binding of ERG transcription factor across or nearby multiple potentially causal SNPs within HNF1B locus (Fig. 3d). We next performed ChIP assays with antibodies to ERG, HNF1B, AR, histone modifications, or with IgG as control. Using quantitative PCR (qPCR) in VCaP with or without androgen treatment, we confirmed specific chromatin enrichment of ERG, HNF1B, AR, H3K4me1, H3K4me2, H3K4me3, and H3K27ac at rs12453443, rs7405696, rs718960, rs11263763, and rs11651052 regions, respectively (Fig. 3e–i). In contrast, similar ChIP assays in fusion-negative LNCaP cells revealed no clear enrichment of ERG in these SNP regions and rather weak enrichment of AR, HNF1B and histone marks (Supplementary Fig. 3g–k). Through a motif analysis at the HNF1B locus, we found four SNPs rs4430796, rs718960, rs8064454 and rs11651052 to some extent residing within DNA-binding motifs of ERG and HNF1B (Supplementary Fig. 3l). Given that *ERG* hijacks the regulatory element of *TMPRSS2* gene to form the most frequent *ERG* fusion in the clinical setting of PCa[31], we further investigated in various cohorts whether the expression correlation between *ERG* and *HNF1B* is dependent on the *TMPRSS2-ERG* fusion status. These analyses revealed that *ERG* and *HNF1B* show apparent co-expression in *TMPRSS2-ERG* fusion-positive groups of PCa specimens, but not in fusion-negative groups (Fig. 3j, k and Supplementary Fig. 3m, n).

SNPs are germline inherited while *TMPRSS2–ERG* fusion is an acquired somatic genomic alteration. Concurrent presence of these germline-somatic features might exert synergistic impact on the target gene *HNF1B* with a positive eQTL signal in this region. In line with this, we found that none of these 13 SNPs (Fig. 2a) had eQTL association with *HNF1B* in TCGA cohort of PCa patients (Fig. 3l). When taking into account of *TMPRSS2-ERG* fusion status and stratifying patients into *TMPRSS2-ERG* fusion-positive and negative groups, we observed that the eQTL results from the *ERG* fusion-negative group are non-significant (Fig. 3m and Supplementary Fig. 4), whereas seven SNPs showed significant eQTL signal with *HNF1B* in the *TMPRSS2-ERG* fusion-positive PCa tumors (Fig. 3n and Supplementary Fig. 5a–n). Based on these observations, we next asked whether the frequency of the co-occurrence of *TMPRSS2-ERG* fusion-positive tumors that present the HNF1B locus SNP alleles increase the risk of PCa aggressiveness in the TCGA cohort. We thus performed an enrichment analysis of these SNPs in *TMPRSS2-ERG*

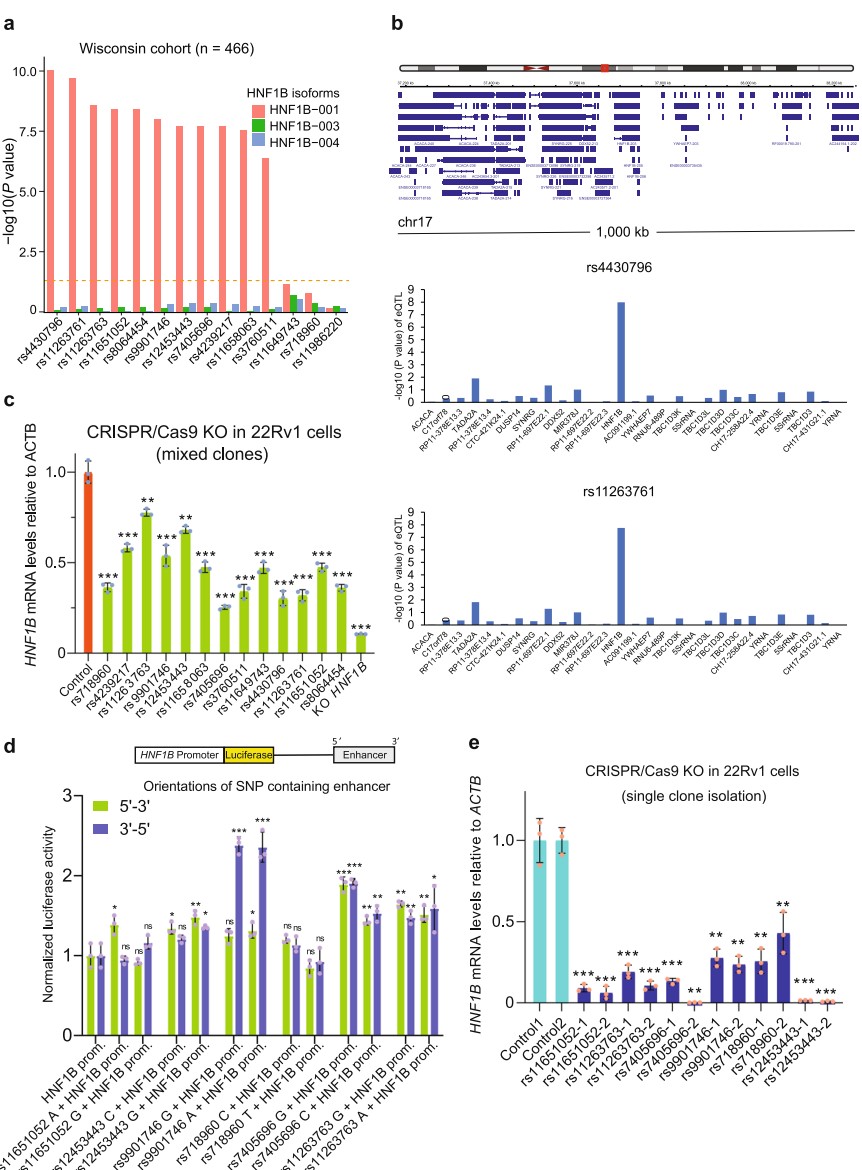

**Fig. 2 | Multiple eQTL SNPs at 17q12 locus are potential causal variants and involved in the regulation of *HNF1B* expression. a** Eleven of the thirteen SNPs in the HNF1B region were found to have significant eQTL association with *HNF1B* isoform 1 expressed in prostate glands (*n* = 466). The SNP rs11986220 within the 8q24 region was used as a non-relevant control site for the analysis. HNF1B-001 (NCBI source: NM_000458); HNF1B-003 (NCBI source: NM_001165923); HNF1B-004 (NCBI source: NM_001304286). Genotype to gene expression correlations were assessed with linear regression. **b** The 17q12/HNF1B locus eQTL analysis of genes within 1-Mbp region using the Wisconsin cohort of benign prostate tissues (*n* = 466). **c** RT-qPCR analysis to determine the mRNA expression levels of *HNF1B* in 22Rv1 cells with partial knockout (KO) of each SNP region or dampened *HNF1B* via CRISPR/Cas9 genome editing technology. *n* = 15 samples; *P* values based on the order of appearance: 7,39E−05, 3,6E−04, 3,8E−03, 6,8E−04, 1E−03, 1,7E−04, 3,14E−05, 9,29E−05, 1,8E−04, 8,08E−05, 6,59E−05, 1,5E−04, 6,46E−05, 1,44E−05.

**d** Reporter assays on six SNP regions with different alleles and in both orientations (5′–3′, 3′−5′) showing contribution of each region as an enhancer for *HNF1B* promoter in 22Rv1. Prom: Promoter. *n* = 25 samples; *P* values based on the order of appearance: 0,025, 0,589, 0,402, 0,17, 0,024, 0,085, 8,9E−03, 0,016, 0,071, 2E−04, 0,038, 6,1E−04, 0,091, 0,268, 0,22, 0,602, 9,5E−04, 6,1E−04, 9,4E−03, 7,1E−03, 2E−03, 8,6E−03, 8,1E−03, 0,035. **e** RT-qPCR analysis of *HNF1B* expression levels in 22Rv1 cell clones with CRISPR/Cas9-mediated deletion of each SNP region. Two independent clones were isolated for analysis from each KO-population. *n* = 14 samples; *P* values based on the order of appearance: 4,4E−05, 5,1E−05, 9,4E−05, 5E−05, 4,9E−05, 2,5E−05, 2E−04, 1,5E−04, 3E−04, 2,8E−03, 2,7E−05, 2,6E−05. In (**c–e**), *n* = 3 technical replicates, error bars, mean ± SD, * *P* < 0.05, ** *P* < 0.01, *** *P* < 0.001, ns: non-significant, *P* values were evaluated using the two-tailed Student's *t* tests. Source data are provided in Source Data file.

fusion-positive and -negative tumors. However, the results revealed no particular enrichment pattern for the SNP alleles in the PCa patients with TMPRSS2-ERG fusion-positive or -negative tumors (Supplementary Fig. 5o). We next additionally investigated whether genotype at SNPs in HNF1B is associated with *TMPRSS2-ERG* positive tumors in a Chinese PCa cohort (see "Methods"; Supplementary Table 1). We first performed a case-control study (791 PCa vs 752 non-PCa) and found that, among these 13 SNPs, rs8064454, rs7405696, rs11651052, rs9901746, rs11263763, rs11658063 and rs12453443 were

significantly associated with PCa in this cohort (Supplementary Table 2). We next carried out a PCa case-only study (136 ERG positive vs 655 ERG negative) while assuming that positive ERG expression based on immunohistochemistry (IHC) is due to the *TMPRSS2-ERG* fusion. Despite that we did not observe significant associations of the 13 HNF1B locus SNPs with ERG fusion status, this analysis revealed a moderate but significant association of the three LD SNPs rs79882976 (*P* = 0.01), rs11651496 (*P* = 0.01) and rs3744764 (*P* = 0.02) with ERG expression positive tumors (Supplementary Table 3). We

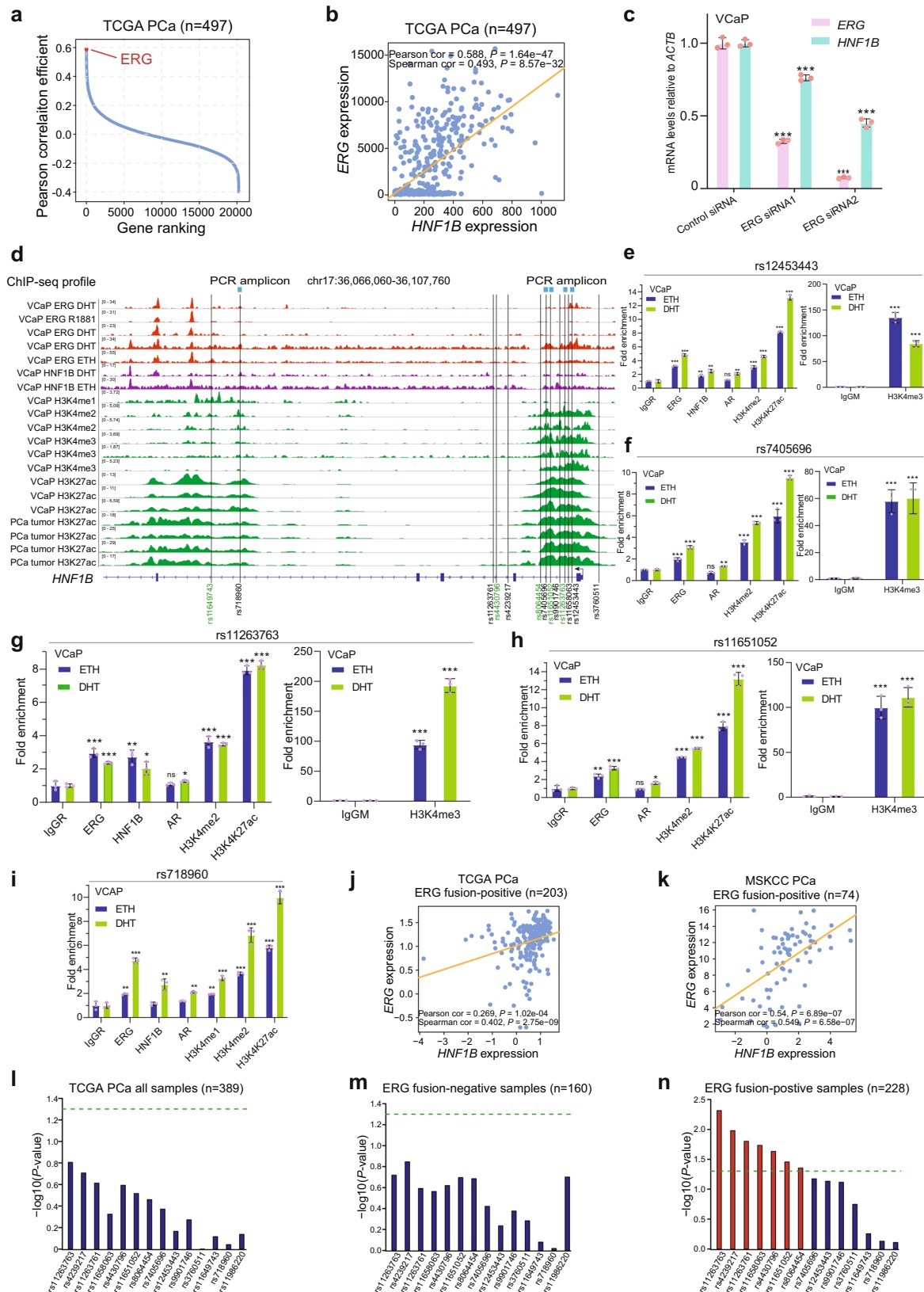

expect larger sample size may enable the detection of more associations between the HNF1B SNP genotypes and *TMPRSS2-ERG* fusion-positive PCa tumors. Taken together, these results suggest that the effect of these variants on the target gene HNF1B are further orchestrated by the acquired somatic event of TMPRSS2-ERG gene fusion.

## HNF1B regulates cell cycle progression pathways implicating PCa severity

To understand the role of HNF1B in PCa, we performed RNA-seq analysis of differentially expressed genes upon depletion of HNF1B with siRNA control or two different siRNAs against *HNF1B* in the PCa VCaP cells (Fig. 4a). In each experimental group, two biological replicates

**Fig. 3 | ERG is a responsible transcriptional regulator of the 17q12 locus and regulates HNF1B in PCa cells and clinical tissues in a manner dependent on TMPRSS2-ERG fusion status. a, b** Unbiased genome-wide analysis showing *ERG* as the most co-expressed gene positively correlated with *HNF1B* in PCa specimens (*n* = 497). *P* values were assessed by the two-sided Pearson's product-moment correlation and Spearman's rank correlation rho tests. **c** Depletion of *ERG* results in reduced mRNA levels of *HNF1B* in *TMPRSS2-ERG* fusion positive VCaP cell. *n* = 3 samples; *P* values based on the order of appearance: 9E−06, 2E−04, 2E−06, 1E−05. **d** Genome browser representation of ChIP-seq signals of active enhancer marks and transcription factor ERG at *HNF1B* regions harboring several potential causal SNPs as indicated. **e–i** ChIP-qPCR validation for chromatin enrichment of ERG, HNF1B, AR, H3K4me1, H3K4me2, H3K4me3, and H3K27ac at given SNP regions in VCaP cells treated with 100 nM DHT and without (ETH-treated). (**e, g**) *n* = 16 samples, (**f, h, i**) *n* = 14 samples; (**e**) *P* values based on the order of appearance: 1,2E−05, 2,2E−05, 1,6E−03, 1,6E−03, 0,053, 2,8E−03, 1E−04, 2,4E−05, 1,2E−07, 9,7E−07, 2E−05, 9,7E−07; (**f**) *P* values based on the order of appearance: 4,5E−04, 4,6E−05, 0,13, 4,1E−03, 2,9E−05, 1,4E−06, 1,6E−04, 3,2E−07, 3,1E−04, 3,2E−07; (**g**) *P*

values based on the order of appearance: 9E−04, 5,1E−05, 3,9E−03, 0,012, 0,63, 0,0204, 4,2E−04, 6,9E−06, 5,9E−06, 1,4E−06, 2,2E−05, 8,6E−06; (**h**) *P* values based on the order of appearance: 4,1E−03, 7,1E−05, 0,73, 0,021, 5,8E−05, 6,3E−07, 3,2E−05, 9,3E−06, 1,7E−04, 6,1E−05; (**i**) *P* values based on the order of appearance: 9E−03, 3,6E−05, 0,54, 4,7E−03, 0,131, 1,4E−03, 8,9E−03, 2E−04, 2,2E−04, 1E−04, 3,1E−05, 1,E−05 **j, k** Scatter plots showing positive correlations between *ERG* and *HNF1B* expression in PCa specimens with *TMPRSS2-ERG* fusion in the TCGA (*n* = 203) and MSKCC (*n* = 74) cohorts. *P* values were assessed by the two-sided Pearson's product-moment correlation and Spearman's rank correlation rho tests. Seven SNPs individually had a significant eQTL association with *HNF1B* expression in *TMPRSS2-ERG* fusion positive PCa group (**n**, *n* = 160) but not in the whole cohort (**l**, *n* = 389) or in ERG fusion negative group (**m**, *n* = 228). *P* values were assessed by the linear regression. In (**c**) and (**e–i**), *n* = 3 technical replicates, error bars, mean ± SD, * *P* < 0.05, ** *P* < 0.01, *** *P* < 0.001, ns: non-significant, *P* values were evaluated using two-tailed Student's *t* tests. Source data are provided in Source Data file.

were included and showed high correlations (Supplementary Fig. 6a–c). This RNA-seq analysis identified 207 significantly upregulated and 132 downregulated genes by HNF1B, respectively (DESeq2, *P* < 0.05; Fig. 4b). Representative differentially expressed genes were further validated via qPCR after siRNA-mediated knockdown of HNF1B (Supplementary Fig. 6d, e). To investigate potential functional categories of HNF1B knockdown target genes, we performed gene set enrichment analysis (GSEA) and revealed several cell cycle relevant terms highly enriched in HNF1B upregulated genes (Fig. 4c, d), further supporting the above described role of *HNF1B* in promoting PCa cell proliferation and invasion (Fig. 1d–f and Supplementary Fig. 1b–j). To explore the clinical relevance of HNF1B target genes in PCa, we generated a cell cycle or knock-down signature of HNF1B target genes (see "Methods"). We evaluated the clinical significance in multiple independent cohorts and found that HNF1B cell cycle signature score is positively correlated with the cell cycle progression (CCP) scores[65] (Fig. 4e, f and Supplementary Fig. 6f–l). Consistently, the HNF1B knockdown signature derived from differentially expressed genes by HNF1B also displays noteworthy positive linear correlation with the CCP scores (Fig. 4g and Supplementary Fig. 6m–t). We next investigated the clinical relevance of HNF1B cell cycle signature with PCa tumor progression and severity. The results indicated that the HNF1B cell cycle signature score is significantly elevated in the metastatic group than primary PCa and normal prostate glands (Fig. 4h–j and Supplementary Fig. 7a, b). Patients with higher levels of HNF1B cell cycle signature score are strongly associated with PCa severity, including Gleason score, lymph node metastasis, tumor stage, and biochemical recurrence (Fig. 4k–m and Supplementary Fig. 7c–j). In addition, we observed that patient group with higher levels of HNF1B cell cycle signature score are associated with shorter overall survival, elevated biochemical relapse and metastasis risk (Fig. 4n–p and Supplementary Fig. 7k, l). To confirm the robustness of the association, we next additionally performed a meta-analysis to systematically review, integrate, and provide an overall interpretation of the association between HNF1B cell cycle signature and patient prognostic outcomes across different studies[66]. The meta-analysis revealed that the elevated levels of HNF1B cell cycle signature were significantly associated with poor overall survival (*P* = 3.5e−03, lnHR: 0.85; 95% CI: 0.28–1.41) and biochemical recurrence (*P* = 6.1e−03, lnHR: 0.50; 95% CI: 0.14−0.85) in PCa patients. To validate the effect of the HNF1B cell cycle signature among a set of clinical variables to patient prognosis, we conducted multivariate analysis by incorporating clinical variables including age, Gleason score, PSA, tumor stage, ERG-fusion status, seminal vesical status and extraprostatic extension status in PCa patients using both the continuous and categorical HNF1B cell cycle signature by the median stratification. The multivariate results still showed significant association of the HNF1B cell cycle signature with patient overall survival (Supplementary Fig. 7o) and biochemical recurrence

(Supplementary Fig. 7p) among other clinical variables. To further investigate the clinical importance of HNF1B target genes, we derived HNF1B knockdown upregulated signature, consisting of 207 upregulated genes. We found that patients with higher levels of the score are associated with PCa tumor progression and severity (Supplementary Fig. 7q–x). Collectively, these findings indicate a strong association of HNF1B target genes with PCa progression and implicate the role of HNF1B in PCa severity.

Given that *HNF1B* is directly regulated by and co-expressed with ERG transcription factor, we examined whether HNF1B targeted genes are also regulated by ERG. Thus, we first performed a co-expression analysis of HNF1B and 339 differentially expressed genes upon HNF1B knockdown across multiple clinical PCa datasets. We found that the fraction of genes co-expressed with HNF1B was higher, showing significant co-expression with ERG (Fig. 4q and Supplementary Fig. 7y). Based on RNA-seq analysis of *HNF1B* knockdown, we next curated a panel of 25-gene as HNF1B co-expression signature and measured the degree of their expression correlation with *ERG* in prostate tumors. The results indicated a similar expression pattern for the 25 genes when compared ERG in two large independent cohorts of PCa (Fig. 4r and Supplementary Fig. 7z), further supporting a functional interplay between the PCa susceptibility gene *HNF1B* and the PCa-specific somatic ERG fusion transcription factor.

## HNF1B and ERG display physical interaction, chromatin co-occupancy and cooperative regulation of PCa genes

Given that ERG fusion correlates with and regulates *HNF1B* expression and is a transcriptional mediator of *HNF1B* locus variants, we next examined whether HNF1B and ERG may interact physically with each other. To this end we performed co-immunoprecipitation (Co-IP) assays that revealed indeed an interaction between endogenous HNF1B and TMPRSS2-ERG in VCaP cells treated with ETH (ethanol) and DHT (Dihydrotestosterone) (Fig. 5a). We applied nuclease treatment to rule out the possibility that the interaction between ERG and HNF1B was mediated by a DNA/RNA moiety (Supplementary Fig. 8a). In addition, treatment of VCaP cells with enzalutamide showed partial abolishment of the interaction (Fig. 5b) and slight downregulation of ERG and HNF1B (Supplementary Fig. 8b), suggesting possible therapeutic implications. Moreover, our experiments showed protein-protein interaction between ectopically expressed V5-tagged TMPRSS2-ERG and flag-tagged HNF1B (Supplementary Fig. 8c). We found that ERG interacts with the POU domain of HNF1B protein (Fig. 5c, d and Supplementary Fig. 8d). Consistent with the physical cooperation of TMPRSS2-ERG and HNF1B, it was observed that more than 70% (3,632/5,133) of HNF1B genome-wide binding sites were co-occupied by TMPRSS2-ERG (Fig. 5e). To investigate whether HNF1B global chromatin binding sites vary in different cell lines, we compared HNF1B genome-wide binding regions in VCaP cells with the ones

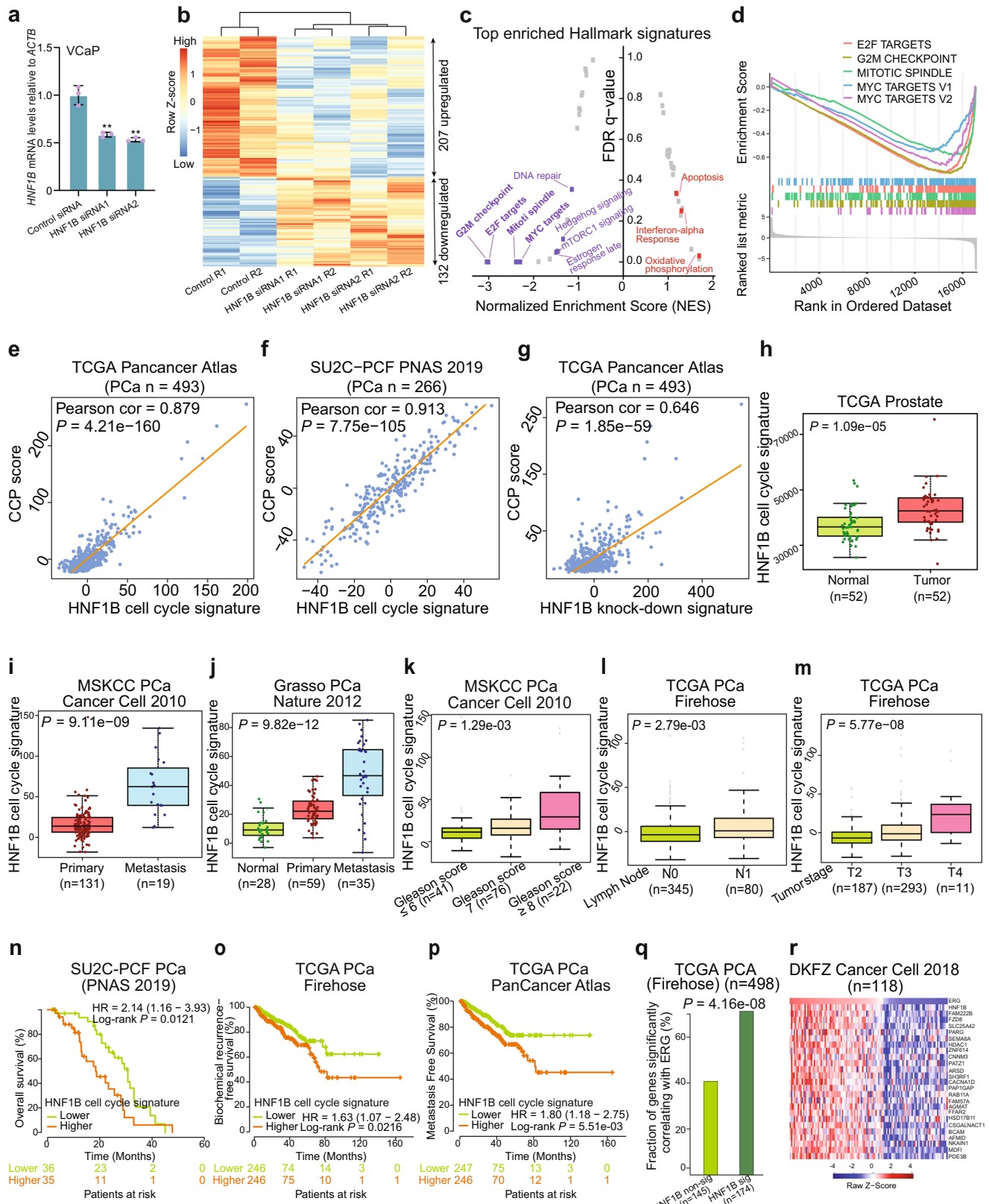

previously reported in ERG fusion-negative and androgen-unresponsive cell line DU145[67]. We observed approximately 19% of overlapping peaks for HNF1B between the two cell lines, indicating the chromatin binding profile varies greatly between ERG fusion-positive and negative cell lines (Supplementary Fig. 8e). We also compared the global binding profiles of ERG in VCaP and of HNF1B in DU145 cells and

observed nearly 48% of common binding sites (Supplementary Fig. 8f), which is extensively overlapping but rather lower than that of their overlap rate in the same cell line VCaP (Fig. 5e).

We next derived a list of 51 genes as the joint target signature of HNF1B and ERG via an integrated analysis of RNA-seq-defined HNF1B upregulated genes and ChIP-seq profiled genomic co-occupancies of

**Fig. 4 | HNF1B gene signatures correlate with PCa progression in the clinical setting. a** Depletion of *HNF1B* in VCaP cells treated with DHT 100 nM 72hrs. *n* = 3 technical replicates, error bars, mean ± SD, ** *P* < 0.01; two-tailed Student's *t* test. *n* = 3 samples; *P* values based on the order of appearance: 2,3E−03, 1,4E−03. **b** Heatmap of HNF1B regulated genes measured by RNA-seq (*p* < 0.05). *P* values were evaluated by the Wald test and not adjusted for multiple comparisons. **c** Top-ranked GSEA enriched pathways associated with genes upregulated and down-regulated by HNF1B, respectively. Categories were ranked by Normalized Enrich-ment Score (NES). **d** GSEA enrichment plots displaying enrichment of cell-cycle related pathways among HNF1B upregulated genes. **e, f** Cell-cycle gene signature based on z-score sum of the 33-cell cycle relevant genes differentially regulated by HNF1B showing strong positive correlations with cell cycle progression (CCP) score in the TCGA (*n* = 493) and SU2C-PCF (*n* = 266) cohorts, respectively. **g** Pearson correlation test displays significant positive linear correlation of HNF1B knock-down signature with CCP score in the TCGA cohort (*n* = 493). In **e**–**g**, *P* values were

assessed by the two-sided Pearson's product-moment correlation test. Higher HNF1B cell cycle signature scores are associated with tumor development and progression to metastasis (**h**, *n* = 108), (**i**, *n* = 150), (**j**, *n* = 122), higher Gleaso*n* score (**k**, *n* = 139), lymph node-positive PCa (**i**, *n* = 425), and higher tumor stages (**m**, *n* = 491); two-sided Man*n*–Whitney U test or Kruskal-Wallis test. The inter-quartile range (IQR) is depicted by the box with the median represented by the center line. Whiskers maximally extend to 1.5 × IQR (with outliers shown). Kaplan-Meier curves depicting the associations between overall survival (**n**, *n* = 71), bio-chemical relapse (**o**, *n* = 492), or metastasis (**p**, *n* = 493) of PCa patients and the HNF1B cell cycle signature scores in PCa patients; log-rank test. **q** Differentially expressed genes that are significantly co-expressed with *HNF1B* are more likely to be co-expressed with *TMPRSS2-ERG* in TCGA cohort (*n* = 498); two-sided Fisher's exact test. **r** A 25 gene co-expression signature with HNF1B displayed a similar expression pattern as ERG in DKFZ cohort (*n* = 118). Source data are provided in Source Data file.

HNF1B and ERG in VCaP (see Methods; Fig. 5f). Notably, HNF1B itself is also a direct target gene of HNF1B and ERG (Fig. 5f) and has several co-binding sites of the two transcription factors (Fig. 5g and Supple-mentary Fig. 8g). DNA-binding motifs of HNF1B and ERG were fre-quently observed within the co-binding sites at HNF1B (Supplementary Fig. 8g). Using ChIP-qPCR, we further validated chromatin co-binding of HNF1B and ERG within HNF1B genomic region and three additional representative genes *TFF1*, *RASSF7*, and *POLR3F* (Fig. 5g and Supple-mentary Fig. 8h–k). We next explored clinical relevance of this HNF1B/ERG joint target signature and examined their correlations with CCP scores. This analysis revealed a significant positive linear correlation in multiple independent PCa cohorts (Fig. 5h, i and Supplementary Fig. 8l–o). Furthermore, patients with higher levels of the signature score is associated with shorter overall survival (Fig. 5j, k).

Taken together, those results indicate additional layers of germline-somatic interplays between HNF1B and TMPRSS2-ERG, including their protein-protein interaction, genome-wide chromatin co-localization and a potential synergistic role for the activation of cis-regulatory elements driving gene expression for PCa progression and prognosis.

## HNF1B and ERG co-occupied chromatin regions indicate a greater enrichment of germline variants across PCa risk loci including 17p13.3

Given that ERG fusion is a transcriptional regulator of PCa risk locus in the 17q12/HNF1B region and physically interacts and shares chromatin binding sites with HNF1B, we asked whether HNF1B alone or together with ERG exerts the genetic impacts that explain PCa risk associations. Hence, we performed an enrichment analysis of SNPs in high linkage disequilibrium (LD, $R^2 > 0.8$) with PCa GWAS variants in ERG unique, ERG and HNF1B common as well as HNF1B unique binding sites, respectively. Strikingly, we found that PCa risk SNPs are more likely to be enriched in the common binding sites of TMPRSS2-ERG and HNF1B compared to their unique DNA-binding regions (Fig. 6a). We next examined a genotype-gene expression association of these enriched SNPs from three sources of eQTL datasets, including GTEx[68], PancanQTL[69] and ncRNA-eQTL[70], and jointly identified 17 eQTL genes (eGenes) for five PCa risk loci (Fig. 6b). We generated a HNF1B & ERG eGene signature and explored its potential prognostic value in the clinical PCa settings. This analysis showed an elevated risk for bio-chemical relapse and metastasis in the patient groups with inter-mediate Gleason score 7 while having higher eGene signature scores (Fig. 6c, d) but not in PCa patients with Gleason score 6 or 8 (Sup-plementary Fig. 9), suggesting that this eGene signature may serve as molecular stratifier for PCa patients with intermediate risk disease.

We and others have shown that transcription factors can bind to SNP regions, and often regulate SNP-linked eQTL genes which they are co-expressed[4,46,60]. We thus examined the expression pattern of the 17 eGenes in two independent cohorts of 266 or 118 PCa patients[71,72] and

found that most of the eGenes indicated high degree of co-expression with HNF1B and ERG, and interestingly, *FAM57A* together with *GEMIN4* and *VPS53* at the 17p13.3 PCa risk locus consistently displayed the strongest co-expression with both *HNF1B* and *ERG* (Fig. 6e and Sup-plementary Fig. 10a). We then focused on the 17p13.3/rs684232 locus that has been reproducibly reported to be strongly associated with PCa susceptibility[12,73] whereas the underlying biology and functional mechanisms remain elusive.

In agreement with an enrichment of the 17p13.3 locus variants in HNF1B and TMPRSS2-ERG common binding sites, ChIP-seq profiles in ERG fusion-positive VCaP cells display multiple strong binding peaks of HNF1B and TMPRSS2-ERG at the 17p13.3/rs684232 region (Fig. 6f). These chromatin occupancies were further validated through ChIP-qPCR assays (Fig. 6g and Supplementary Fig. 10b). To examine whether variation at rs684232 directly influenced DNA-binding motifs of any transcription factors, we employed bioinformatics analysis using the enhancer element locator tool[74] and a computing program namely affinity testing for regulatory SNP (atSNP)[75] with collected ETS and HNF1B DNA-binding motifs from previous studies[44,76,77]. This analysis showed that the rs684232/17p13.3 appears to alter the HNF1B-ERG heterodimer binding sites (lower panel, Fig. 6f). Given that the rs684232/17p13.3 locus SNPs are functionally mediating HNF1B and ERG in transcriptional regulation, we next examined whether these variants are enriched or exclusive to the TMPRSS2-ERG positive tumors in PCa patients. However, the result indicated no specific enrichment of risk alleles the patients with TMPRSS2-ERG positive and −negative tumors (Supplementary Fig. 10c). Based on these findings, we hypothesized that HNF1B and ERG could functionally influence the expression of 17p13.3 locus genes. We thus performed siRNA-mediated knockdown of *ERG* in VCaP and LNCaP cells treated with DHT and CRISPR-Cas9-mediated genomic deletion of *HNF1B* in the LNCaP-derived castration resistant cell model V16A[78], and found down-regulation of *VPS53*, *FAM57A* and *GEMIN4* mRNA levels (Fig. 6h, i). In contrast, we did not observe expression changes of the 17p13.3 locus genes in a successful siRNA-mediated knockdown experiment in the ERG fusion-negative control cell line LNCaP (Supplementary Fig. 10d). Furthermore, through querying multiple clinical PCa datasets, we found a positive expression correlation between *FAM57A* and *ERG* or *HNF1B* in prostate tumors (Fig. 6j, k), but not in adjacent normal prostate tissues (Supplementary Fig. 10e, f), indicating ERG and HNF1B directed transcriptional reprogramming in human PCa tumorigenesis. ERG has been reported to be highly expressed in TMPRSS2-ERG fusion-positive PCa specimens[31]. In line with this, we found that ERG expres-sion levels were significantly higher in ERG fusion-positive group compared to ERG fusion-negative PCa patient group (Supplementary Fig. 10g, h). We next examined whether the expression levels of *FAM57A* and *HNF1B* are dependent on ERG fusion status in the clinical PCa cohorts. As expected, the mRNA levels of *HNF1B* and *FAM57A* are greatly higher in ERG fusion-positive groups (Fig. 6l, m and

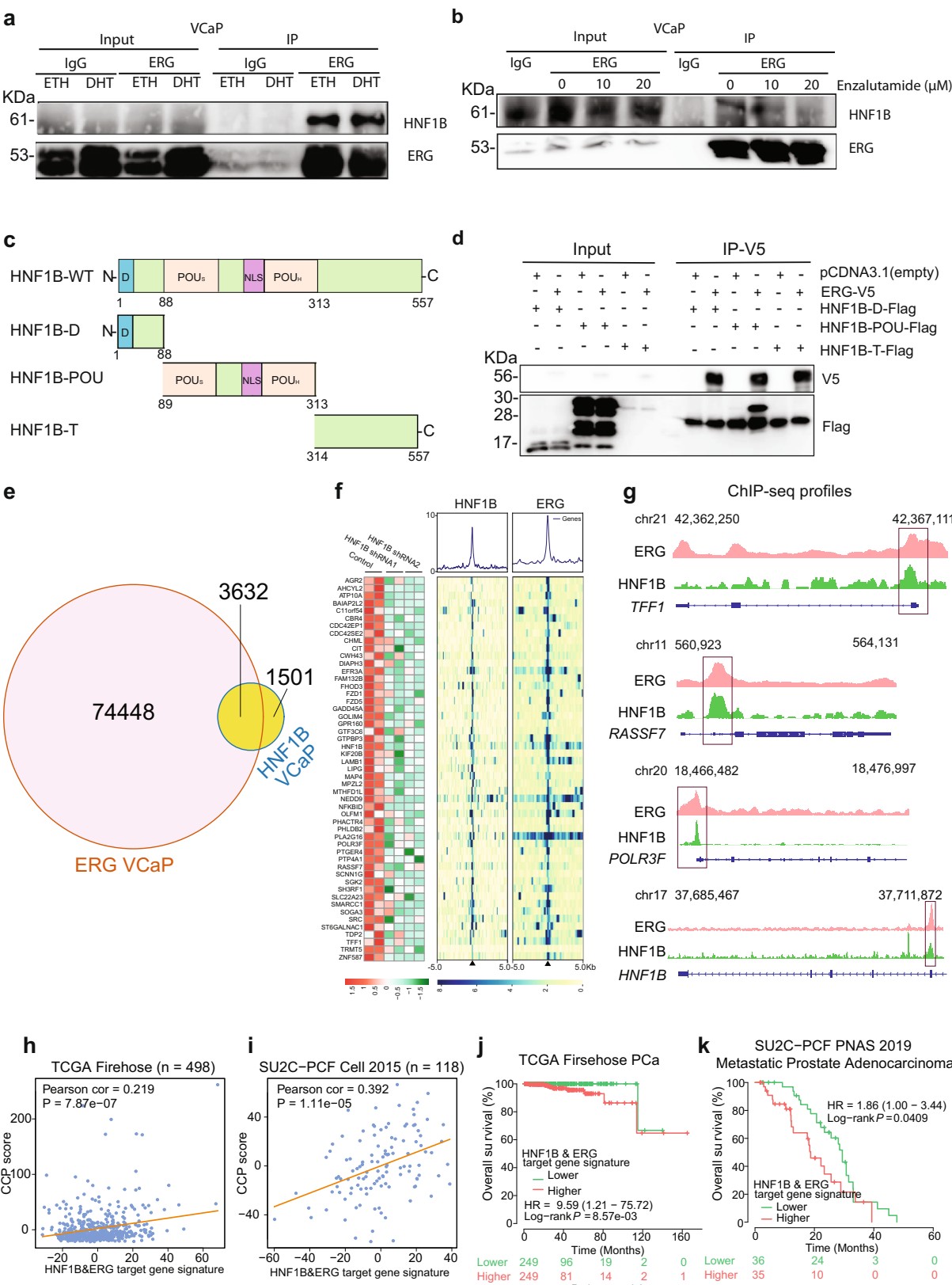

Supplementary Fig. 10i, j). Given that a cluster of cis-regulatory elements were reported to regulate elevated expression of TMPRSS2-ERG[79] and that TMPRSS2-ERG and HNF1B co-bound at a broad H3K27ac-marked super-enhancer region in *HNF1B* (Fig. 3d) and at the rs684232/17p13.3 region (Fig. 6f), we investigated whether known transcription inhibitors could perturb this transcriptional misregulation circuit that drives the expression of several PCa risk-associated genes. Thus, we treated the PCa cell line V16A with different concentration of BET inhibitor that is known to target a family of BRD proteins highly activated in PCa and playing roles in the control of cell-cycle-associated genes[80,81]. The results showed marked reduction of *HNF1B*, *FAM57A*, and *GEMIN4* expression (Supplementary Fig. 10k),

**Fig. 5 | HNF1B physically interacts with ERG and co-opts ERG for chromatin co-occupancy and clinical correlation in PCa. a** IP-WB assay displaying endogenous interaction of HNF1B and ERG in VCaP cells treated with 100 nM DHT and without (ETH-treated) for 24hrs. **b** Endogenous interaction of HNF1B and ERG in VCaP with enzalutamide treatment for 72hrs. **c** Schematic of the full-length HNF1B (NM_000458) with D-dimerization domain (1-32aa), POU$_S$ - POU-specific domain (89-178aa), NLS-nuclear localization signal (229-235aa), POU$_H$ - POU homeodomain (236-313aa), and T-transactivation domain (314-557aa). Three recombinant domains tested for ERG interaction: HNF1B-D (1-88aa), HNF1B-POU (89-313aa) and HNF1B-T (314-557aa). aa: amino acids. **d** Interaction of HNF1B with ectopically expressed ERG in vitro via POU domain determined by co-IP in 293 T cells. **e** Over 70% of HNF1B binding sites are co-occupied by ERG. **f** Heatmaps of RNA-seq and ChIP-seq signals on the direct target genes of HNF1B and ERG. HNF1B and ERG ChIP-seq signals are plotted for the genes shown, where blue color indicates higher enrichment. **g** Chromatin-binding of HNF1B and ERG on the representative genes, *TFF1*, *RASSF7*, *POLR3F*, and *HNF1B*. The chromosome number and position of the peaks are also indicated. **h, i** Pearson correlation tests showing strong positive linear correlation between HNF1B & ERG direct target gene signature scores and CCP scores in the TCGA ($n = 498$) and SU2C-PCF ($n = 118$) cohorts of PCa patients. HNF1B and ERG direct target gene signature based on z-score sum of the 51 differentially upregulated genes by siRNAs against HNF1B also with ERG & HNF1B ChIP-seq coverage. *P* values were assessed by the two-sided Pearson's product-moment correlation test. **j, k** Kaplan-Meier curves depicting the overall survival of PCa patients in the TCGA ($n = 498$) and SUC2-PCF ($n = 71$) cohorts. Patient groups were stratified by median levels of HNF1B & ERG direct target gene signature scores. The log-rank *P* values are denoted in the figures. In (**a**, **b** and **d**), representative experiment of three independent co-immunoprecipitation experiments is shown. Source data are provided in Source Data file.

indicating a translational potential of converged somatic ERG fusion, germline loci 17p13.3/rs684232 and 17q12/HNF1B in the same clinical setting.

## Multiple causal variants and causative genes at 17p13.3 implicating PCa susceptibility

We next examined an enhancer-like function of the 17p13.3 region harboring the SNPs rs2955626, rs684232 and rs461251 for PCa susceptibility genes *VPS53*, *FAM57A* and *GEMIN4*. Hence, we proceeded to test whether these SNPs could directly alter the promoter activity of *VPS53*, *FAM57A* and *GEMIN4*, respectively. We inserted the SNP-containing regions into the upstream of *VPS53*, *FAM57A* or *GEMIN4* promoter in pGL4.10-basic vector, except for rs2955626 already within the promoter region of *VPS53*, and performed enhancer reporter assays in VCaP cells with ETH and DHT treatment. The results showed that SNP-containing regions possess an enhancer activity towards *VPS53*, *FAM57A* and *GEMIN4*. Notably, the rs684232 and rs461251-containing regions have an increased enhancer activity when VCaP cells were treated with DHT compared to that of ETH treatment (Fig. 7a). To investigate whether there are direct chromatin interactions between rs2955626/rs684232/rs461251-containing enhancer and proximal regulatory regions of *VPS53*, *FAM57A* and *GEMIN4*, we performed quantitative chromosome conformation capture assays (3C-qPCR)[82] with the restriction enzyme HindIII. The constant fragment was designed between the two binding sites of HNF1B and ERG with the SNPs rs2955626, rs684232 and rs461251. We determined its interaction with HindIII-digested chromatin fragments in a 45-kb region covering *VPS53*, *FAM57A* and *GEMIN4* promoter regions in VCaP cells with ETH or DHT treatment and the lung cancer cell line A549. The results showed that the SNP regions have higher crosslinking frequencies with *FAM57A* in DHT-treated VCaP compared to ETH-treated VCaP or lung cancer cells A549, suggesting an apparent impact on the observed interactions between the SNP enhancers and *FAM57A* upon androgen stimulation (Fig. 7b).

To further verify if rs2955626, rs684232 and rs461251 are directly involved in the regulation of *VPS53*, *FAM57A* and *GEMIN4*, we applied the CRISPR/Cas9 genome editing approach in PCa cell line V16A and obtained four independent mixed cell clones with partial depletion of each SNP enhancer region at 17p13.3. The results showed reduced transcriptional levels of *VPS53*, *FAM57A* and *GEMIN4* upon deletion of the regulatory region harboring rs2955626, rs684232 or rs461251 compared to the parental V16A line (Fig. 7c), suggesting a causal function of rs2955626, rs684232 and rs461251 for the expression of *VPS53*, *FAM57A* and *GEMIN4*.

Finally, to test the biological relevance of the 17p13.3 locus genes, we pursued tumor cellular assays to demonstrate the effect of VPS53, FAM57A and GEMIN4 on PCa cellular phenotypes. The PCa 22Rv1 cells harboring shRNAs against *VPS53*, *FAM57A* or *GEMIN4* showed markedly attenuated cell growth and viability in comparison with cells harboring control shRNA in the proliferation assays (Fig. 7d–g). Consistent results were obtained via a real-time monitoring of wound-healing assays, indicating that 22Rv1 cells harboring gene-specific shRNAs showed decreased wound closure rates (Fig. 7h–j). Collectively, these results support a mechanistic model of multiple causal variants and causative genes implicating PCa susceptibility and tumor cellular transformation at this 17p13.3 locus.

## Discussion

In this study, we revealed an extensive germline-somatic interaction implicating PCa susceptibility and progression through the oncogenic regulatory circuits, consisting of the most frequent PCa-specific somatic genomic alteration TMPRSS2-ERG, and several germline PCa risk locus genes such as *HNF1B* at 17q12 and *VPS53-FAM57A-GEMIN4* at 17p13.3 (Fig. 7k). Our findings demonstrated not only multiple potential causal SNPs and risk CREs within 17q12/HNF1B locus, and the responsibility of TMPRSS2-ERG fusion for transforming molecular and biological effects of 17q12 and regulating *HNF1B* expression, but also mapped genome-wide binding sites of HNF1B and defined its high rate of chromatin co-occupancy with TMPRSS2-ERG that can explain more of genetic associations discovered by GWASs in PCa. Given that the 17q12/HNF1B has been reported as a cross-cancer pleiotropic genetic risk locus[14–26], our work revealed an understanding of its underlying causation and biological mechanisms implicating in PCa risk prediction and prognosis, while exhibited as an example for the comprehensive evaluation of this locus contributing to risk association and disease progression in other types of cancers.

Here we found *HNF1B*, together with *MYC*, *FOXA1*, *HOXB13* and *AR*, as one of the most essential genes for PCa cell survival and its elevated expression in PCa tumors accompanied with higher grade and Gleason score. Despite the fact that we did not observe significant association between *HNF1B* expression and patient prognosis, which might be due to the limited sample size in current available data sets, we expect to detect strong association with the availability of large patient cohorts in the future. *HNF1B* has three isoforms through alternative splicing and has been shown that the two longer isoforms tend to be transcriptional activators, whereas the shortest isoform is a transcriptional repressor[83]. We proved the relationship between *HNF1B* isoforms and the 17q12 PCa risk SNPs by which an eQTL analysis indicates a strong association between 11 SNPs and *HNF1B* isoform 1, the dominant isoform expressed in prostate specimens and other tissues. Moreover, we discovered the role of *HNF1B* in regulating genes involved in cell cycle progression pathways implicated in PCa progression and clinical severity. It is noteworthy that uncontrolled cell growth and division are a cardinal feature of cancer cells[84], and accordingly, many inhibitory drugs have been developed in clinical use to target cell cycle progression for cancer therapy[85]. For example, CDK4/6 inhibitors targeted the G1/S phase cell-cycle pathway were recently applied and evaluated in the clinical trial for advanced PCa[86]. Here we show that risk genotypes of the 17q12 PCa risk SNPs are associated with an elevated expression of HNF1B, thereby contributing to PCa cell proliferation

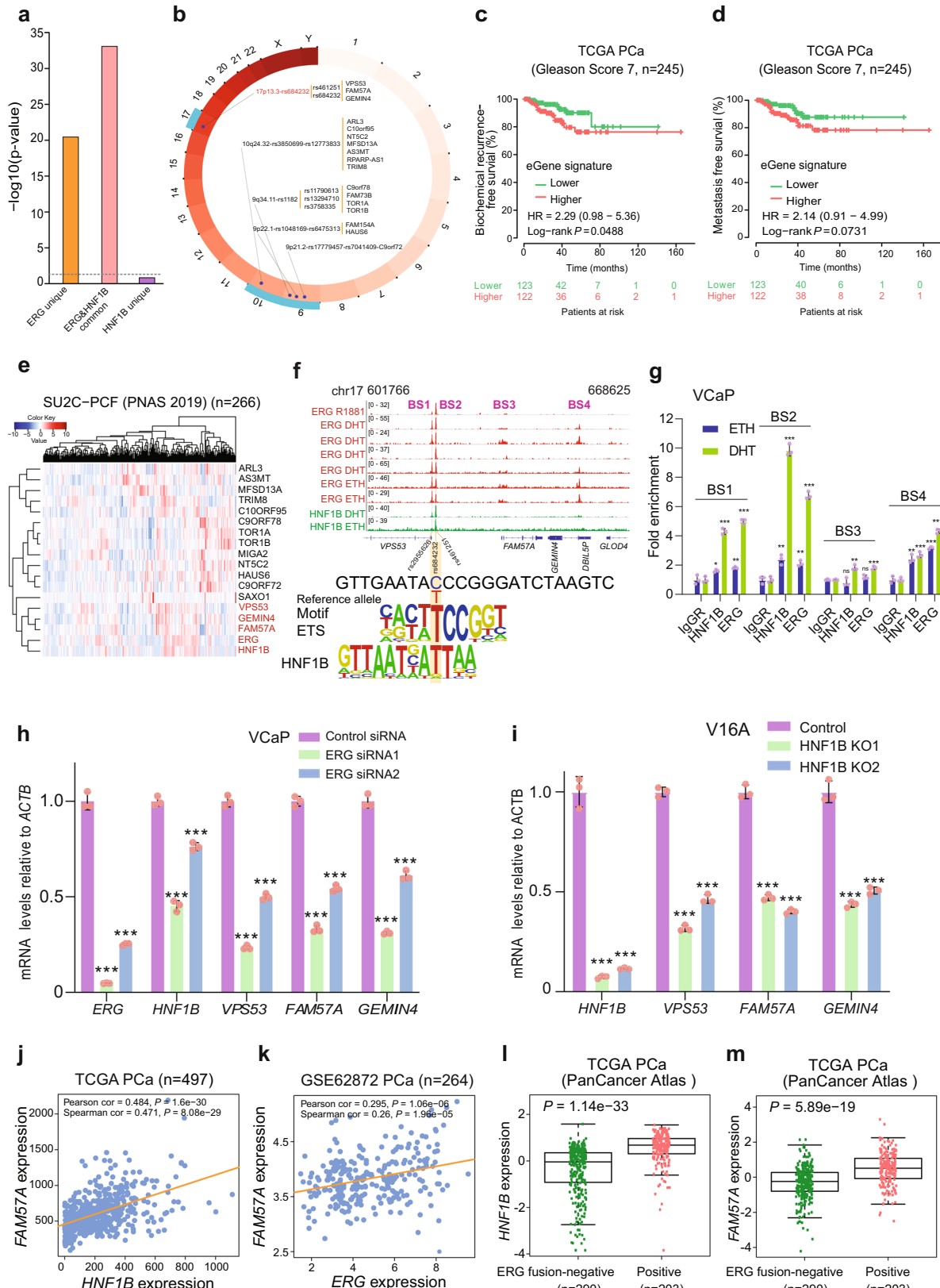

and tumor progression. Thus, the findings may raise an important topic for future study through applying cell cycle pathway inhibitors for the PCa patients with high HNF1B expressing tumors.

Despite the fact that TMPRSS2-ERG has been reported as the most frequent genomic rearrangement in PCa and shown profound roles in PCa initiation and development, for instance through cooperating with

activated PI3K/AKT pathway or PTEN loss[42,43], all these examples are somatic interaction events. Here our study reported a germline-somatic association of TMPRSS2-ERG with this cross-cancer locus of 17q12/HNF1B, mechanistically and biologically implicating PCa risk and progression. Given that TMPRSS2-ERG fusion is a somatic event specific to PCa[31-36], additional transcription regulators remain to be

**Fig. 6 | PCa susceptibility alleles are highly enriched in HNF1B and ERG co-occupied chromatin regions. a** PCa GWAS risk SNPs displays increased enrichment in common binding sites of HNF1B and ERG; Chi-squared test. **b** Display of PCa risk loci enriched in HNF1B and ERG common binding sites. Tag SNPs are followed by proxy SNPs with a cutoff $R^2 \geq 0.8$ and eQTL genes. Kaplan-Meier plots displaying biochemical relapse (**c**) and metastatic (**d**) rates of PCa patients with an intermediate risk of Gleason score 7 ($n = 245$). $P$ values were assessed by a log-rank test. **e** Heatmap demonstrating the expression profile of the eQTL genes in the clinical specimens ($n = 266$). Note that *FAM57A*, *GEMIN4*, and *VPSS3* display a similar expression pattern to *HNF1B* and *ERG*. **f** HNF1B and ERG ChIP-seq signals at the 17p13.3/rs684232 region harboring the eQTL genes *VPSS3*, *FAM57A*, and *GEMIN4*. rs684232 alters binding of HNF1B and ERG. Genomic browser tracks and rs684232-surrounding genome sequence PWM matches are displayed. **g** ChIP-qPCR validation of HNF1B and ERG binding at the regions within 17p13.3 in VCaP cells treated with 100 nM DHT and without (ETH treatment). $n = 6$ samples; $P$ values based on the order of appearance: 0,03, 4E−05, 9,8E−03, 1,5E−05, 4,6E−03, 4,3E−06, 3,8E−03, 6,47E−06, 0,34, 1,6E−03, 0,098, 3,3E−04, 2,5E−03, 2,4E−04, 1E−04,

9,7E−06. **h** ERG knockdown in VCaP cells downregulates expression of *HNF1B*, *VPSS3*, *FAM57A*, and *GEMIN4*. $n = 3$ samples; $P$ values based on the order of appearance: 3E−06, 8E−06, 1E−05, 2E−04, 2E−06, 1E−05, 3E−06, 1E−05, 5E−06, 9E−05. **i** CRISPR/Cas9-mediated deletion of *HNF1B* leads to reduced expression of *VPSS3*, *FAM57A*, and *GEMIN4* in V16A cells. $n = 3$ samples; $P$ values based on the order of appearance: 3E−05, 4E−05, 2E−06, 9E−06, 2E−06, 5E−06, 6E−05, 1E−04. Scatter plots displaying significant expression correlation between *FAM57A* and *HNF1B* (**j**, $n = 497$) or *ERG* (**k**, n = 264). $P$ values were assessed by the two-sided Pearson's product-moment correlation and Spearman's rank correlation rho tests. **l**, **m** *HNF1B* or *FAM57A* expression levels are significantly elevated in *TMPRSS2-ERG* fusion-positive PCa specimens compared to non-fusion group ($n = 493$). $P$ values were evaluated by two-sided Mann–Whitney U test. The interquartile range (IQR) is depicted by the box with the median represented by the center line. Whiskers maximally extend to $1.5 \times$ IQR (with outliers shown). In (**g-i**), $n = 3$ technical replicates, error bars, mean ± SD, * $P < 0.05$, ** $P < 0.01$, *** $P < 0.001$, ns: non-significant; two-tailed Student's $t$ test. Source data are provided in Source Data file.

---

uncovered to explore underlying mechanisms of the 17q12/HNF1B locus associated with other types of cancers[22–26]. Here our results also highlight that the positive correlations of *HNF1B* expression with the 17q12 SNP genotypes are dependent on ERG fusion status. Correspondingly, we observe a moderate enrichment of three HNF1B SNPs in PCa patients with tumors expressing ERG proteins though there are obvious limitations due to small sample size and IHC-stained ERG expression as a surrogate for ERG fusion. We thus expect larger studies with confirmative ERG fusion status might enable the detection of an apparent enrichment of the genotype of SNPs in HNF1B with ERG fusion-positive tumors in humans in the future. Future studies may also test synergistic effects of 17q12 SNP genotypes and HNF1B expression in predicting PCa clinical outcomes similar as previously reported[60].

We also show a physical protein-protein interaction between HNF1B and TMPRSS2-ERG, which in turn co-occupy a large fraction of chromatin regions enriched with various PCa risk-associated non-coding genomic variants, including the SNPs at the 17p13.3 PCa susceptibility locus. Previous studies reported the regulation of *VPSS3*, *FAM57A* and *GEMIN4* within the 17p13.3 regions through regulatory enhancer region harboring SNPs rs2955626 and rs684232 wherein ERG functions as a transcriptional mediator of those CREs[4,87]. Here our observations support those findings and freshly pinpoint the involvement of HNF1B in the regulation of *VPSS3*, *FAM57A* and *GEMIN4* through a synergistic cooperation with TMPRSS2-ERG where ChIP-seq data showed occupancy of common binding sites in the SNP-surrounding enhancer regions with rs2955626, rs684232 and rs461251. All the SNP regions showed enhancer activity for each gene at 17p13.3 locus but strikingly, we observed an increased enhancer activity on *FAM57A* proven by the CRISPR/Cas9-mediated genome editing, and also by enhancer report assay and quantitative analysis via chromosome conformation capture assays. Finally, we tested the biological relevance of *VPSS3*, *FAM57A* and *GEMIN4* expression in PCa cells and demonstrated that knockdown of these genes reduces cell growth and migration. These findings may possess translational application to benefit patients in the future. We show that AR inhibitor enzalutamide attenuates protein-protein interaction between ERG and HNF1B. We also show that BET inhibitor treatment can markedly reduce the expression of HNF1B, FAM57A, and GEMIN4. Together with recent identification of a small molecule selectively inhibiting ERG-positive cancer cells[88], we expect additive effects among these drugs to inhibit the interaction and expression of ERG and HNF1B or their target genes thereby affecting PCa cell growth in the clinical setting as an important topic for future studies.

In summary, our results provide mechanistic insight into how the PCa risk locus 17q12/HNF1B contributes to disease severity and progression through the germline-somatic interplay between HNF1B and

TMPRSS2-ERG with a potential for transcriptionally mediating more genetic variance underpinning PCa susceptibility.

## Methods
The presented study complies with all relevant ethical regulations and was approved by the University of Oulu and the Shanghai Jiao Tong University School of Medicine Affiliated Ruijin Hospital. All participating patients provided written informed consent. Patients were not monetarily compensated.

### Cell culture
The cell lines used in the work (Supplementary Table 4) are 22Rv1 (CRL-2505, ATCC), LNCaP (CRL-1740, ATCC), VCaP (CRL-2876, ATCC), DU145 (HTB-81, ATCC), PC3 (CRL-1435, ATCC), V16A[78], A549 (CCL-185, ATCC), RWPE1 (CRL-11609, ATCC), and 293 T (CRL-11268, ATCC). All cell lines were confirmed to be mycoplasma free during our study. As described above, most of cells lines were originally purchased from ATCC (American Type Culture Collection). Cell morphology and growth rate of the cell lines used in this study were similar to previous reports. These cell lines have been authenticated by STR fingerprinting. The cells were cultured under the conditions of 37 °C and 5% $CO_2$. VCaP and 293 T were grown in Dulbecco's Modified Eagle's Medium (DMEM) (11965092, Thermo FisherInvitrogen), for culturing DU145 we used Eagle's Minimum Essential Medium (EMEM) (30-2003, ATCC). Also, LNCaP, 22Rv1 and V16A were grown in Roswell Park Memorial Institute Medium (RPMI 1640) (R8758, Sigma) and finally A549 and PC3 was grown in F12-K (30-2004, Invitrogen). The cell culture media were supplied with a final concentration of 10% fetal bovine serum (16000044, Thermo Fisher) and 1% of penicillin and streptomycin (15140122, Thermo Fisher)). RWPE1 cells were grown in Keratinocyte-Serum Free Medium. Keratinocyte-SFM Kit including epidermal growth factor (EGF) and bovine pituitary extract (BPE) supplements were purchased from Invitrogen (17005-042, Invitrogen). The VCaP cells were cultured in the charcoal-stripped media wherein activating the androgen receptor signaling through a dihydrotestosterone (DHT) (Olli A. Jänne lab, University of Helsinki) treatment with final concentration of 100 nM for 24 h.

### Plasmids and gene cloning
Human cDNA library was used to amplify HNF1B open reading frame (ORF) that was cloned into pLVET-IRES-GFP and pcDNA3.1 vectors (Supplementary Table 9). Wild type ERG was also amplified from the same library and cloned into pcDNA3.1. The cDNA of TMPRSS2-ERG fusion was cloned from VCaP into pcDNA3.1. HNF1B-D (domains of full-length HNF1B NM_000458), HNF1B-POU and HNF1B-T of HNF1B sub-domains were cloned into pcDNA3.1, respectively, to express three recombinant proteins for testing protein interaction between HNF1B

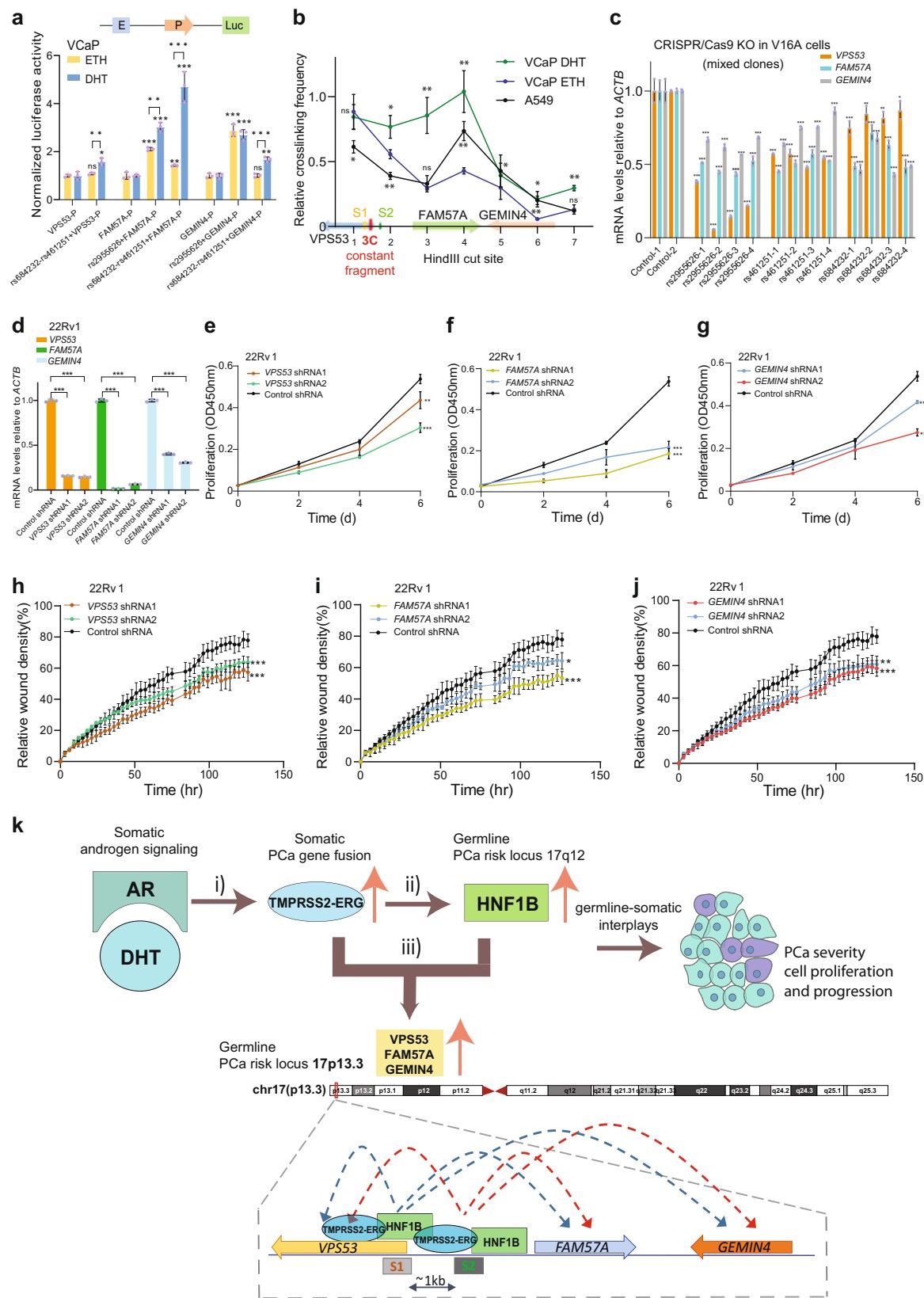

and TMPRSS2-ERG at domain levels. Primer sequences, cloning methods and enzymes are shown in Supplementary Tables 5 and 15.

## Construction of reporter plasmids

Each enhancer or promoter region was amplified from human genomic DNA and cloned into pGL4.10 [luc2] (E6651, Promega) (Supplementary

Table 9) containing a SNP (rs718960, rs7405696, rs11651052, rs9901746, rs11263763 or rs12453443) region. Each of these six SNPs was cloned with two different alleles obtained by site-directed mutagenesis. In addition, three SNP (rs2955626 or rs461251 and rs684232) regions were cloned. The enhancers were cloned into the BamHI site (in both orientations), and the promoters of HNF1B, VPS53, FAM57A or

**Fig. 7 | Effects of the 17p13.3 locus PCa susceptibility alleles on *VPS53*, *FAM57A* and *GEMIN4*. a** Luciferase reporter assays showing elevated enhancer activity of the regions with rs2955626, rs684232 and rs461251 for the promoters of *VPS53*, *FAM57A* and *GEMIN4*, respectively. E: Enhancer; P: Promoter; Luc: Luciferase. $n = 16$ samples; *P* values based on the order of appearance: 0,115, 0,036, (rs684232-rs461251 + VPS53-$P_{ETH \cdot DHT}$: 4,3E−03), 3,1E−04, 5E−05 (rs2955626 + FAM57A-$P_{ETH \cdot DHT}$: 1,3E−03), 8,4E−03, 3,1E−04 (rs684232-rs461251 + FAM57A-$P_{ETH \cdot DHT}$: 8E−04), 3E−04, 2E−04, 0,92, 1E−03 (rs684232- rs461251 + GEMIN4-$P_{ETH \cdot DHT}$: 7,4E−04). **b** 3 C analysis of chromatin interactions between rs2955626, rs684232 and rs461251 region, and the promoters of *FAM57A* or *GEMIN4* in a 45 kb genomic area (chr17:612,242-656,774) at 17p13.3 locus. S1: rs2955626; S2: rs684232 and rs461251. 3 C constant fragment colored in red. $n = 3$ samples; *P* values A549: 1 (0,029), 2 (2,8E−03), 3 (0,42), 4 (2,4E−03), 5 (0,023), 6 (1,7E−03), 7 (0,65); *P* values VCaP DHT: 1 (0,686), 2 (0,018), 3 (2E−03), 4 (3E−03), 5 (0,276), 6 (0,012), 7 (2E−03). **c** RT-qPCR analysis of the mRNA levels of *VPS53*, *FAM57A*, and *GEMIN4* in mixed clones of V16A cells with CRISPR/ Cas9-mediated deletion of rs2955626, rs684232 or rs461251 region. Each group with selected four distinct mixed clones. $n = 14$ samples; *P* values based on the order of appearance: 4,9E−08, 7,3E−07, 6,5E−06, 6,2E−10, 8,3E−07, 8,5E−06, 1,1E−08, 1,2E−06, 2,03E−07, 1,5E−09, 1,7E−05, 1,9E−06, 3,2E−08, 6,2E−07, 1,5E−06, 1,7E−05, 3,8E−06, 8,1E−06, 1,7E−08, 2,01E−05, 4,03E−06, 1,5E−07, 8E−07, 9,3E−04, 7E−04, 7E−06, 1,3E−05, 7E−03, 9E−04, 8,98E−05, 1,7E−03, 5,2E−05, 4,5E−07, 0,022, 7E−05, 2,4E−07. **d** Depletion of *VPS53*, *FAM57A* and *GEMIN4* in 22Rv1 through lentivirus-mediated shRNA knockdown. $n = 9$ samples; *P* values based on the order of appearance: 1,3E−07, 1,6E−07, 6,6E−08, 9,1E−08, 4,7E−07, 1,8E−07. Knockdown of *VPS53* (**e**), *FAM57A* (**f**) or *GEMIN4* (**g**) reduces PCa cell proliferation. $n = 3$ samples; (**e**) *P* values shRNA1: 0,0087, shRNA2: 0,00024; (**f**) *P* values shRNA1: 5,59E−05, shRNA2: 7,3E−05; (**g**) *P* values shRNA1: 1,3E−04, shRNA2: 8,3E−04. Wound healing assay in 22Rv1 cells infected with lentiviruses expressing shRNAs for (**h**) *VPS53*, (**i**) *FAM57A* and (**j**) *GEMIN4*. (**h-j**) n = 3 samples; (**h**) *P* values shRNA1: 6,8E−04, shRNA2: 8,7E−04; (**i**) *P* values shRNA1: 1,65E−06, shRNA2: 0,029; (**j**) *P* values shRNA1: 9,3E−04, shRNA2: 4,9E−03. **k** Model of the germline-somatic interplay at the 17p13.3 locus between TMPRSS2-ERG and HNF1B driving PCa cell growth and tumor severity. (**i**) Androgen signaling implicates the development of *TMPRSS2-ERG* fusion and stimulates its expression via androgen-responsive TMPRSS2 element. (**ii**) Aberrant TMPRSS2-ERG fusion cooperating with the 17q12 PCa susceptibility locus augments the expression of *HNF1B*. (**iii**) HNF1B co-opts TMPRSS2-ERG fusion and synergistically regulates a dozen of PCa risk loci (Fig. 6a, b), including the 17p13.3 PCa susceptibility genes *VPS53*, *FAM57A* and *GEMIN4*, thereby driving PCa cell proliferation and tumor progression and severity. Lower panel summarizes regulatory circuits at the PCa risk locus chr17p13.3 through HNF1B co-option of TMPRSS2-ERG. S1: rs2955626; S2: rs684232 and rs461251. In (**a-j**), $n = 3$ technical replicates, error bars, mean ± SD, * *P* < 0.05, ** *P* < 0.01, *** *P* < 0.001, ns: non-significant, *P* values were evaluated using two-tailed Student's *t* tests. Source data are provided in Source Data file.

GEMIN4 into the EcoRV/HindIII sites of pGL4.10 [luc2] vector, respectively. Both orientations can facilitate testing enhancer activity of SNP-containing regions regardless of the promoter location. The constructs were transient, reversely transfected into LNCaP or VCaP (treated with DHT or ETH) cells with a Renilla Luciferase control plasmid pGL4.75 [hRluc/CMV] (E6931, Promega) by using X-treme GENE HP DNA Transfection Reagent (06366236001, Roche). The experiments were performed on the 96-well white plates with each well containing 100 µl medium of $3 \times 10^5$ 22Rv1 and LNCaP cells/ml or $9 \times 10^5$ VCaP cells/ml. After incubation at 5% $CO_2$ and 37 °C for 48 h, the luciferase activity was measured with Dual-Glo Luciferase Assay System (E2940, Promega). At least three replicate wells were used per construct and the data were statistically analyzed with a two-tailed Student's *t* test. Primer sequences, cloning methods and enzymes are shown in Supplementary Tables 5 and 15.

## Protein blot analysis

Cell pellet was resuspended in lysis buffer (600 mM Nacl, 1% Triton X-100 in PBS, freshly added 1 x protease inhibitor) and sonicated (Q800R sonicator, Q Sonica). The sample was centrifuged, and the supernatant was collected. The amount of protein was measured with Pierce BCA Protein Assay Kit (23225, Thermo Fisher Scientific) based on the manufacturer's protocol and 30 µg of protein lysate of each sample was separated by electrophoresis in 7.5% or 12% SDS-PAGE gel and transferred into 0.45 µm Immobilon-P PVDF Membrane (IPVH00010, Millipore) using a Semi-Dry transfer cell (Trans-Blot SD, Bio-Rad). After transfer, the membrane was blocked for minimum 30 min at room temperature using blocking buffer (5% nonfat milk in TBST) while gently shaking. The blocked membrane then was incubated with antibody diluted in blocking buffer (1:1000 (Ab µl: blocking buffer µl): rabbit polyclonal anti-HNF1B, mouse monoclonal anti-HNF-1B and mouse monoclonal anti-FLAG. 1:5000: mouse monoclonal anti-V5, mouse monoclonal anti-V5-HRP, mouse monoclonal anti-ERG, rabbit monoclonal anti-ERG) at 4 °C for 16 h with gentle rotation. After incubation, the membrane was washed three times each 10 min using TBST. Anti-rabbit IgG or anti-mouse IgG was used as secondary antibody (Thermo Fisher) with 1:5000 dilution into blocking buffer and the incubation took place on a rotor at room temperature for 1 h. Afterwards, the membrane was washed three times each with 15 min using TBST. Finally, the membrane was developed with Lumi-Light Western Blotting Substrate (12015200001, Roche) or SuperSignal West Femto Maximum Sensitivity Substrate (34095, Thermo Fisher Scientific) according to the protocol and exposed with Fujifilm LAS-3000 Imager.

Original blots are provided in the Source data file. For more information about the antibodies see Supplementary Table 6.

## Ectopic expression via transient transfection

293 T cells were used for transient transfection with pcDNA3.1 constructs (Supplementary Tables 9, 14). Mixer A of pcDNA3.1 construct and P3000 reagent (L3000015, Thermo Fisher Scientific) was diluted with Opti-MEM. Mixer B of lipofectamine 3000 reagent (L3000015, Thermo Fisher Scientific) was diluted with Opti-MEM (11058021, Thermo Fisher). We mix A and B which were incubated at room temperature for 15 min and added into 70-80% confluent seeded cells. The cells were incubated 24-48 h before harvesting.

## Co-immunoprecipitation

Co-Immunoprecipitation was performed for examining the endogenous interaction of HNF1B with TMPRSS2-ERG in VCaP with DHT treatment. The ectopic interaction of HNF1B or HNF1B domains with ERG cloned into pcDNA3.1 and transfected in 293 T (cloning primers listed in Supplementary Table 5). Cells were harvested and lysed with 0.5 ml cold immunoprecipitation buffer (50 mM Tris-HCl, pH 7.5, 150 mM NaCl, 1% Triton X-100, 10% Glycerol, 1 mM EDTA and 1% protease inhibitor cocktail). Keep Cell lysates on ice for 30 min with few vortex periods in between. Sonicate in 4 °C water bath for 20 s. Centrifuge for 30 min at 4 °C and supernatant was incubated with 30 µl protein-G Magnetic Beads to pre-clear crude cell extract of proteins which can bind non-specifically to the beads at 4 °C for 1 h. Keep the supernatant and add 5 µg of antibody (5 µg: mouse monoclonal anti-V5, rabbit monoclonal IgG and rabbit monoclonal anti-ERG, mouse monoclonal anti-FLAG) (Supplementary Table 6) with incubation at 4 °C overnight. Add 30 µl of fresh protein-G Magnetic Beads and incubate at 4 °C for 6 h followed by washing five times with Immunoprecipitation buffer. Furthermore, resuspend beads in 30 µl of 2 x SDS sample loading buffer and incubate at 95 °C for 5 min and finally use the supernatant on SDS-PAGE gel for electrophoresis separation.

## siRNA transfections

Individual set of two siRNAs (Qiagen) against *HNF1B* or *ERG* were tested in knockdown efficiency and compared with the siRNA negative-control (Qiagen) by RT-qPCR. For cell proliferation assays, we used the same set of two siRNAs (Qiagen) against *HNF1B* compared with negative and positive control siRNA (Qiagen). $8 \times 10^5$ of VCaP and LNCaP cells were used in reverse transfection for 6-well plate, respectively. siRNA transfection was performed with HiPerFect Transfection

Reagent (301705, Qiagen) with a final concentration of 50 nM siRNA (see Supplementary Table 7 for the siRNAs used).

### RNA isolation and real time quantitative PCR

RNeasy Mini Kit (74106, QIAGEN) was applied for RNA isolation and RNase-Free DNase (79254, QIAGEN) was used during the isolation to remove DNA. cDNA was synthesized from 2 μg RNA by either the High-Capacity cDNA Reverse Transcription Kit (4368814, Applied Biosystems) or the iScript Reverse Transcription Supermix (1708840, Bio-Rad). After cDNA synthesis we used SYBR Select Master Mix (4472920, Applied Biosystems) and high specificity primers for the quantitative RT-PCR reactions. The results were normalized with beta-actin control and for each gene's analysis made triplicates. Primers used for RT-qPCR in Supplementary Table 8.

### CRISPR/Cas9-mediated genome editing analysis

CRISPR design tool (http://crispr.mit.edu/or crispor.tefor.net) was used to prepare the pair of oligos (sgRNA-top and sgRNA-bottom) attached in Supplementary Table 8. Most of the experiment was performed according to the previous protocol[89]. For annealing process 1 μl sgRNA-top (100 μM) and 1 μl sgRNA-bottom (100 μM) were mixed with 1 x T4 ligation buffer, 1 μl T4 PNK and 6 μl ddH$_2$O. The oligos were phosphorylated and annealed in a thermocycler at 37 °C for 30 min followed with 95 °C for 5 min; ramp down to 25 °C with 5 °C/min. Then, the annealed oligos were inserted into pSpCas9 (BB)−2A-Puro and the plasmids were transfected in 22Rv1 and V16A cells with 70-80% confluency. 0.6 μg of total amount of Cas9 plasmids designed for the same SNP region, with 1:1 ratio or 1:1:1:1 ratio, which added into cells by using Lipofectamine 3000 in 24-well plate according to the protocol. Medium was changed after 24 h and replaced with medium containing 1 μg/ml puromycin (P9620, Merck). Afterwards, the successfully transfected cells were isolated to single cells by dilution or FACS. The single cells were seeded into 96-well plates and after 2-3 weeks the positive clones were examined further by genotyping and RT-qPCR determination of gene expression.

### Chromatin immunoprecipitation (ChIP)

ChIP assay was carried out based mainly on previous study[62]. The cells were cross-linked in a final concentration of 1% formaldehyde in medium for 10 min at room temperature with gently shaking. The final concentration of 125 mM glycine was added to stop the reaction and incubated for minimum 5 min with slight shake. Cells were harvested and the pellet was resuspended in hypotonic lysis buffer (20 mM Tris-Cl, pH 8.0, with 10 mM KCl, 10% glycerol, 2 mM DTT, and freshly added cOmplete protease inhibitor cocktail (04693159001, Roche) and incubated up to 1 h on a rotor at 4 °C. Afterwards, the pellet was washed twice with cold PBS and resuspended in SDS lysis buffer (50 mM Tris-HCl, pH 8.1, with 0.5% SDS, 10 mM EDTA, and freshly added cOmplete Protease Inhibitor). Sonication (Q800R sonicator, Q Sonica) was performed as far as the chromatin had size 250-500 bp. Later, 70 μl of Dynabead protein G (10004D, Invitrogen) were washed twice with blocking buffer (0.5% BSA in IP buffer) and incubated with 8 μg of antibody (Rabbit polyclonal anti- HNF1B, Rabbit polyclonal IgG, Rabbit monoclonal IgG, Mouse polyclonal IgG, Rabbit monoclonal anti-ERG, Anti-rabbit Androgen Receptor, H3K4me1, H3K4me2, H3K4me3 and H3K27ac) (examined antibodies in Supplementary Table 6) in 1 ml of 0.5% BSA in IP buffer (20 mM Tris-HCl, pH 8.0, with 2 mM EDTA, 150 mM NaCl, 1% Triton X-100, and freshly added Protease inhibitor cocktail) for 10 h at 4 °C on rotor. After incubation the supernatant was removed, and the sonicated chromatin lysate (200-250 μg) was diluted in 1.3 ml of IP buffer and was added into the beads-antibody complex with incubation at 4 °C for at least 12 hrs on rotor. Afterwards, the beads-antibody complex was washed one time with wash buffer I (20 mM Tris-HCl, pH 8.0, with 2 mM EDTA, 0.1%SDS, 1% Triton X-100, and 150 mM NaCl) and once with buffer II (20 mM Tris-

HCl pH, 8.0, with 2 mM EDTA, 0.1% SDS, 1% Triton X-100, and 500 mM NaCl), followed by two times of washing with buffer III (10 mM Tris-HCl, pH 8.0, with 1 mM EDTA, 250 mM LiCl, 1% deoxycholate, and 1% NP-40) and two times with buffer IV (10 mM Tris-HCl, pH 8.0, and 1 mM EDTA). 50 μl of extraction buffer (10 mM Tris-HCl, pH 8.0, 1 mM EDTA, and 1% SDS) were added to extract from the beads the DNA-protein complex by incubating and shaking at 65 °C for 20 min (repeat same step with another 50 μl of extraction buffer). Proteinase K (AM2548, Thermo Fisher Scientific) with final concentration 1 mg/ml and NaCl with final concentration 0.3 M were added into the extracted DNA-protein complex and incubated at shaking heat block for 16 h at 65 °C in 1000 rpm to reverse-crosslink of the protein-DNA interactions. DNA was purified with MinElute PCR Purification Kit (28006, QIAGEN) followed by ChIP-qPCR with primers that targeted DNA binding genome sequences (see Supplementary Table 8). ChIP library was prepared according to manufacturer's protocol TruSeq Sample Preparation Best Practices and Troubleshooting Guide (Illumina). Finally, the sample were sequenced and analyzed.

### Quantitative analysis of chromosome conformation capture assay

Quantitative analysis of chromosome conformation capture assay (3C-qPCR) was performed as described in the Hagege et al. protocol[82]. The primers used for these assays are listed in Supplementary Table 8. The cells were trypsinized and resuspended in PBS with 10% FBS. $1 \times 10^7$ cells were cross-linked in PBS with 10% FBS and 1% formaldehyde for 10 min at room temperature. To stop the crosslinking reaction, we added 0.57 ml of 2.5 M glycine (ice cold). The pellets of VCaP with treatment and A549 were resuspended in 5 ml cold lysis buffer and incubate for 13 min on ice. Then we centrifuge at 400 g at 4 °C and remove the supernatant and keep the pelleted nuclei, which were collected by centrifugation and used for digestion. We continue on digestion step of sample with HindIII restriction enzyme to digest chromatin DNA and the digestion efficiency was verified. The digested nuclear lysate was used in the ligation step. After ligation, for purification of DNA we increased the volume of sample to dilute DTT presented in the sample with 7 ml distilled water, 1.5 ml of 2 M sodium acetate pH 5.6 and 35 ml ethanol. After washing pellet with 70% ethanol and dry the pellet, and resuspended it in 150 μl of 10 mM Tris pH 7.5. DNA was desalted with centrifuge filters (Microcon DNA Fast Flow) (MRCF0R100, Millipore). For TaqMan qPCR, we used 1 μl of the 3 C sample (100 ng/μl), 5 μl of Quanti tech probe PCR mix (QIAGEN), 1 μl of Taqman probe (1.5 μM), 1 μl of Test + Constant primer (5 μM) and 2 μl distilled H2O. We performed standard curve of each primer using serial dilution of control template, containing amplified fragments across each of 7 HindIII cut sites and mix them together. Values of intercept and slope from the standard curve were used to evaluate the ligation product using the following equation: Value = 10 (Ct-intercept)/slope. These values were finally normalized to ERCC3 (loading control).

### Lentiviral constructs, lentivirus production and infection

HNF1B was cloned into the lentivirus plasmid pLVET-IRES-GFP for ectopic expression (see Supplementary Table 5). Two set of shRNA constructs in the pLKO.1-puro vector targeting HNF1B, VPS53, FAM57A or GEMIN4 (Merck) were applied for knockdown assays. More information on the shRNA constructs can be found in Supplementary Table 7. Lentiviral constructs were produced with the third-generation packaging system in human embryonic kidney (HEK) 293 T cells (ATCC, CRL-11268) which were seeded the previous day into 3.5-cm plate in a 70%−80% confluency. At the day of transfection, the medium was replaced with 1 ml low glucose DMEM (Invitrogen) containing 10% FBS, 0.1% penicillin-streptomycin. A mix of four plasmids was made in a ratio 1:1:1:3 in a total amount of 10 μg (pVSVG-envelope plasmid, pMDLg/pRRE-packaging plasmid, pRSV-Rev-packaging plasmid and

lentiviral transfer vector) (Supplementary Table 9) and diluted in Opti-MEM with Lipofectamine 2000. 24 h later the medium was replaced with 2 ml fresh medium. After that time, the virus-containing medium was collected every 24 h for 3 d and then was centrifuged at 95 g for 5 min and the supernatant was filtered with 0.45 μm filter unit place on syringe. Then the sample was collected and frozen with liquid nitrogen before stored at −80 °C. For virus transduction into the desired cells seeded 24 h before transduction in 3.5 cm plate, the final concentration of 8 μg/ml polybrene (Sigma) was added in 1.4 ml medium and 0.6 ml lentivirus-containing medium. Then the culture medium of target cells was replaced with the above prepared mix and incubate for 24 h at 37 °C and 5% $CO_2$. In case of puromycin (Sigma) selection construct, after 24 h the medium was replaced with pre-warmed medium, and 48 h after transduction the medium was changed with fresh medium containing puromycin in a final concentration of 2 μg/ml. Cells without virus transduction were used as control to determine cell survival status upon puromycin selection. For the GFP expression constructs, 48 h after transduction the cells were sorted positively by fluorescence activated cell sorting (FACS) using BD FACS Aria flow cytometer (BD Biosciences).

### Cell proliferation assays

The experiments were performed on the 96-well plates with each well containing 100 μl medium of $2 \times 10^3$ V16A, PC3, DU145, RWPE1 or 22Rv1 cells and $8 \times 10^3$ VCaP cells per well, respectively, in an incubation period of 4-6 d with 5% $CO_2$ and 37 °C. The Cell Proliferation Kit II XTT (11465015001, Roche) was used according to the manufacturer. Cell proliferation was examined at indicated time points by XTT colorimetric assay (absorbance at 450 nm). At least three replicate wells were prepared per condition and the data were statistically analyzed with a two-tailed Student's t test. The information on critical commercial assays can be available in Supplementary Table 12.

### Wound healing assays

Cells were seeded into 96-well imageLock plates with the appropriate culture medium that can allow to grow near 100% confluence. Then we used WoundMaker tool to create homogenous scratch wounds and cells were washed twice with PBS. Culture medium was added into each well. The wound areas of each well were imaged every 2 h for max 180 h using Essen BioScience IncuCyte Live-Cell Imaging System.

### Gene expression correlation analysis

We performed the co-expression analysis to evaluate the expression correlation between HNF1B, *ERG* and *FAM57A* from multiple independent cohorts with benign and cancerous prostate tissues. The co-expression tests were also applied in scenarios considering TMPRSS2-ERG status. Both Pearson's product-moment correlation and Spearman's rank correlation rho methods were applied in all linear expression correlation tests. Genes were ranked according to Pearson coefficient value in a descending order to identify the gene that is most co-expressed with *HNF1B* in a genome-wise scale.

### Survival analysis

Survival analysis was applied to assess the impact of HNF1B cell cycle signature, ERG & HNF1B target gene signature and ERG & HNF1B eGene signature on PCa prognosis and survival in multiple independent cohorts. The survival analyses were performed and visualized as Kaplan-Meier plots using R package "Survival" (v.3.2.3)[90,91]. Patients were stratified into two groups based on the median value of the z-score summed signature scores. Function "Surv" was first employed to create the survival models with "time-to-event" and "event status" as input from clinical cohorts. Then signature scores was further followed to fit to the models by function "survfit". The Cox proportional-hazards model[92] was applied to investigate the hazard ratio for assessing the association between patients' survival time and gene expression or signature scores.

### Expression quantitative trait loci (eQTL) analysis

To evaluate the associations between genotypes of SNPs and HNF1B expression level, we performed the expression quantitative trait loci (eQTL) analysis by R package "MatrixEQTL" v.2.2[93] in Wisconsin and TCGA cohorts[94,95], which comprised of 466 normal and 389 prostate tumor samples, respectively. To examine whether T2E fusion affects the eQTL signal, we matched available T2E information to the existing TCGA cohort and further stratified patients into fusion-positive and -negative PCa tumors consisting of 160 and 228 samples, respectively. The eQTL analysis was applied by fitting a linear regression model between the expression and the genotype data, other parameters were left as default (pvOutputThreshold = 0.05, errorCovariance = numeric ())". The transcriptional profiling in TCGA cohort was assessed by RNA-Seq. The TCGA cohort was genotyped on Affymetrix SNP array 6. The relevant SNP genomic locations are listed in Supplementary Table 13.

### Enrichment analysis of HNF1B SNPs in ERG-positive PCa tumors

To investigate whether the SNPs were associated with TMPRSS2-ERG fusion-positive tumors in PCa, we performed an independent association study in a Chinese prostate biopsy cohort[96] and radical prostatectomy cohort. Briefly, a consecutive prostate biopsy cohort and prostatectomy cohort with biospecimen started from October 2017 to December 2021 at a tertiary hospital in Shanghai, China. ERG was regularly stained in biopsy tissue samples via IHC. IHC was performed on the 4-μm-thick FFPE tissue sections using commercially available antibodies against ERG (Agilent Technologies Singapore). Antibody staining was detected using a universal immunoperoxidase polymer method (Envision-kit; Dako, Carpinteria, CA, US). A Dako automated immunohistochemistry system (Dako, Carpinteria, CA, US) was used according to the manufacturer's protocol. Likewise, the IHC results were independently interpreted by two experienced pathologists: Xiaoqun Yang and Chaofu Wang. Genotyping was performed in most of the samples using Illumina Asian Screening Array. Imputation was performed thereafter[97]. A posterior probability of >0.90 was applied to call genotypes during imputation and the same quality control procedure for excluding genotyped SNPs was applied to imputed SNPs. Genotyping data of HNF1B with a ±100 kb window was extracted (n = 1,662). SNPs were excluded if they had: (1) genotype call rate < 90% (n = 1,335); (2) minor allele frequency (MAF) <0.01 (n = 53); or (3) p < .05 for the Hardy–Weinberg Equilibrium (HWE) test. The study was approved by the Institutional Review Board (IRB) of Ruijin Hospital, Shanghai, China. A total of 1,543 samples with ERG expression information were found to have genotyping data (October 2017-December 2021). The demographic characteristics of these patients are shown in Supplementary Table 1. Assuming that positive ERG expression based on IHC is due to ERG fusion, 136 ERG-positive PCa cases out of 791 (17.2%) cases were observed in this cohort (30 ERG-positive biopsies negative to PCa to be excluded). This frequency was similar to the reported ERG fusion frequencies in Asian population[33–36].

### RNA-sequencing (RNA-seq) and differential expression analysis

Preparation of RNA samples was made with VCaP cells, which were treated with 100 nM DHT, and reversely transfected with two different siRNAs targeting HNF1B and negative siRNA control and incubated for 72 h at 37 °C each with two biological replicates. For the RNA sequencing in VCaP cells treated with either negative control siRNAs or siRNAs against HNF1B, raw sequence data were first pre-processed with FastQC (v.0.11.4) to assess read quality. SortMeRna was applied to identify and filter rRNA[98] to limit the rRNA quantity in FastQ files. The filtered data was resubmitted for a QC assessment by FastQC (v.0.11.4) to ensure the validity of the filtering steps. Trimmomatic v.0.39[99] was employed to process reads for quality trimming and adapter removal

with default parameters: TruSeq3-SE.fa:2:30:10 SLIDINGWINDOW: 5:20. A final FastQC (v.0.11.4) run was performed to ensure the success of previous quality control steps. The processed reads were aligned against the human genome assembly hg19 using TopHat2 v.2.1.1[100] with default settings; parameter for library type was set as "fr-firststrand". HTSeq v.0.11.0 (htseq-count) was employed to quantitate aligned sequencing reads against gene annotation from UCSC and with parameters "-s reverse, –i gene_id". Differential expression analysis was performed from read count matrix using Bioconductor package DESeq2 v.1.16.1[101]. Genes with low expressions (<5 cumulative read count across samples) were filtered out before analysis. A threshold of $P < 0.05$ was applied to generate the differentially expressed gene list. Statistical test was applied to control or treatment to ensure high correlations between biological replicates. Data was normalized using method variance Stabilizing Transformation (VST) and the heatmap presenting differentially expressed genes between siRNA Control and siRNAs HNF1B samples was generated using R package "pheatmap" v.1.0.12. All the relevant software and algorithms are listed in Supplementary Table 10.

## Gene set enrichment analysis

We applied Gene Set Enrichment Analysis (GSEA) v.4.0.3 to interpret the RNA-Seq results upon knockdown of *HNF1B*. The pre-ranked gene list was obtained by calculation of data following formula sign (logFC) *-log(p value), and data were sorted in a descending order. GSEA-Preranked test[102] was used to test the enrichment of genes with phenotype in Hallmark gene sets. Parameters were set as follows: Enrichment statistic = "weighted", Max size (exclude larger sets) = 5000, number of permutations =1000. All other parameters were remained as default. The GSEA enrichment plots were generated using R packages "clusterProfiler" v.3.14.3[103] and "enrichplot" v.1.12.0[104].

## Chromatin immunoprecipitation sequencing (ChIP-seq)

The HNF1B ChIP-seq library was sequenced to generate 35-76 bp single-end reads. The HNF1B and ERG replicates were sequenced and produced 150 bp single-end reads. FastQC (v.0.11.4) was applied to assess the quality of raw data and followed by Trimmomatic v.0.39[99] for quality control. The trimmed reads were mapped into the human genome assembly hg19 using Bowtie2 v.2.4.4[105]. MACS2 v. 2.2.7.1[106] was employed for peak calling using default parameters. HOMER v.4.11[107], UCSC, samtools v.1.9[108], bedtools v.2.27.1[109], deepTools v.3.3.2[110] and IGV v.2.4.10 tools were used for peak annotation and generating big wiggle and TDF formats. Bioconductor package ChIPseeker v.1.18.0[111] was applied to perform downstream peak annotation analysis.

## Development of the HNF1B/ERG derived signatures

The HNF1B cell cycle signature, composed of 33 genes, was derived from the five top enriched cell cycle related pathways via GSEA, then further being intersected with the 207-upregulated genes from the RNA-Seq upon HNF1B knockdown. We defined the differentially expressed genes from the RNA-Seq as HNF1B knockdown signature. The HNF1B knock-down upregulated signature score was defined as a z-score sum of the 207 HNF1B upregulated genes by RNA-seq measurement. For the HNF1B and ERG direct target gene signature, we first converted the 207 upregulated gene symbols to Entrez IDs. Bedtools v.2.27.1[109] was used to identify common peaks from HNF1B and ERG ChIP-Seq binding signals. Function "annotatePeaks.pl" from HOMER v.4.11 was applied for annotating HNF1B and ERG common peaks. The 207-upregulated gene list and the gene list from HNF1B and ERG common binding peaks were intersected, and thus resulted in a 51-gene list, defined as HNF1B and ERG direct target gene signature. For the eQTL gene (eGene) signature, we screened 13 proxy SNPs enriched in HNF1B and ERG common binding sites from Haploreg v.4.1[112]. We then set $R^2 \geq 0.8$ as a threshold, which resulted in a total eight proxy SNPs with 17 corresponding eQTL genes. We defined these 17 genes as

HNF1B and ERG eGene signature. Signature scores were calculated as weighted sums of normalized expression of the genes from each signature.

## Meta-analysis

The pooled HR was calculated by a fixed effect model[113], as the $I^2$ statistic was less than 30% or the fixed effects $P$ value for the $I^2$ statistic was greater than 0.10, indicating insignificant heterogeneity across studies[114]. The meta-analysis for investigation of the association between the HNF1B cell cycle signature and patient prognosis across studies was performed using the "metafor" package v.3.4.0[115] in R environment v.4.2.0.

## Multivariate analysis

We investigated the association of the PCa patient overall survival and biochemical recurrence with the HNF1B cell cycle signature and clinical variables including age, Gleason score, PSA, tumor stage, ERG-fusion status, seminal vesical status and extraprostatic extension status. The Cox proportional hazards model was applied for to investigate the relation between patient prognosis and the HNF1B cell cycle signature together with a set of covariates described above. Samples were stratified into two groups with higher and lower expression by comparing to the median value of the HNF1B cell cycle signature or by the continuous value of the HNF1B cell cycle signature score.

## Statistical analysis and data visualization

All statistical analyses were performed using RStudio[116,117] v.1.2.5033 with R environment v.3.6.3 or unless specified. Statistical analyses were applied across normal prostate, tumor and metastatic tissues from multiple cohorts. Mann–Whitney U test was used for gene expression in clinical cohorts with two groups, while Kruskal-Wallis H test was applied for cohorts having three groups or more. R package "Survival" was applied in all Survival analysis. Statistical analyses for all Kaplan-Meier curves were calculated using log-rank test. HNF1B signature scores were calculated from the z-score sum of panels of gene expression levels. For microarray-based expression profiling, we selected gene probes with lowest p values. Circos maps were generated using Circos (v.0.67)[118]. Asterisks indicate the significance level (*p < 0.05; **p < 0.01; ***p < 0.005). P value <0.05 was considered to be statistically significant.

## Reporting summary

Further information on research design is available in the Nature Portfolio Reporting Summary linked to this article.

# Data availability

The publically available GWAS data in PCa used in this study was obtained from the GWAS catalog. The publically available RNA-seq or microarray data including Welsh, Yu (GSE6919), Bittner (GSE2109), Wallace (GSE6956), TCGA, MSKCC, SU2C-PCF, Grasso, DKFZ, GSE62872, Fred Hutchinson CRC, SMMU, Broad/Cornel, CPGEA were retrieved from public databases including cBioPortal for Cancer Genomics[95,119], Oncomine database[120] and GEO database[121,122]. The publically available ERG ChIP-seq profiling data in VCaP cells were obtained from the Cistrome Data Browser[64,123]. To capture the comprehensive genome-wide map of ERG chromatin binding sites, we retrieved and merged 15 ERG ChIP-seq publically available datasets in VCaP from the Cistrome Data Browser, and obtained a union peak set of ERG in VCaP cells (Supplementary Table 11; GEO: GSM717395, GSM717396, GSM717397, GSM1193658, GSM1328978, GSM1328980, GSM1328981, GSM2086315, GSM2086314, GSM2086313, GSM2086312, GSM2086311, GSM2086310, GSM2086309, GSM353637)[37,124–127]. The publically available data used for the eQTL analyses described in this manuscript were obtained from GTEx portal[68], PancanQTL[69] and ncRNA-eQTL[70]. The publically available FinnGen data used in this

research are available to qualified researchers and detailed documentation is provided on the FinnGen study website. Both raw and processed data from the RNA-seq profiling upon HNF1B depletion and HNF1B ChIP-seq generated in this study have been deposited in the ENA (European Nucleotide Archive) database under accession codes PRJEB46082 and PRJEB46088, respectively. The raw and processed data from the ChIP-seq of HNF1B and ERG replicates generated in this study are available at ENA under accession code PRJEB49662. Source data are provided with this paper. The remaining data supporting the findings of this study are available within the Article, Supplementary Information or Source Data file. A reporting summary for this article is available as a Supplementary Information file. Source data are provided with this paper.

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

## Acknowledgements
We want to acknowledge the participants and investigators of the FinnGen study. This work was supported by the grants from the National Natural Science Foundation of China (82073082), the Jane and Aatos Erkko Foundation, the Finnish Cancer Foundation, the Sigrid Juseliuksen Saatio, and the Fudan University Recruit Funding. The work was also supported by a grant from National Institute of Health (1R01CA250018) to L.W. Lentiviral vector and virus preparations were done at the Biocenter Oulu Virus Core Laboratory. Linux High-Performance Computing servers were provided by the CSC – IT CENTER FOR SCIENCE LTD, and also supported by the Medical Research Data Center of Fudan University.

## Author contributions
Conceptualization, G.-H.W.; N.G. designed and performed most of the experiments and analyzed the data. Q.Z. designed, performed, and interpreted most of the bioinformatics analysis. N.G., Q.Z., and G.-H.W. prepared figures. X.Y.Y. performed ChIP-seq assays and together with N.G. for ChIP-qPCR experiments. R.N. provided SNP genotype data in Chinese cohorts and performed the association study. Methodology, N.G., Q.Z., X.Y.Y., R.N., Y.T., Y.Y., X.R., D.H., X.Q.Y., C.W., P.Z., A.M., L.W., and G.-H.W.; Software, Q.Z., R.N., G.-H.W.; Validation, N.G., X.Y.Y., Y.Y.; Formal analysis, Q.Z., G.-H.W.; Investigation, N.G., Q.Z., X.Y.Y., R.N., Y.Y., and G.-H.W.; Resources, R.N., A.M., L.W., G.-H.W.; Data curation, N.G., Q.Z., G.-H.W.; Writing – original draft, N.G., Q.Z., G.-H.W.; Writing – review & editing, N.G., Q.Z., G.-H.W. with inputs from all authors; Visualization, Q.Z., G.-H.W.; Supervision, G.-H.W.; Project administration, R.N., Y.Y., A.M., L.W., G.-H.W.; Funding acquisition, G.-H.W.

## Competing interests
The authors declare no competing interests.
