## [Peer Review File · Nature Communications]

Extensive germline-somatic interplay contributes to prostate cancer progression through HNF1B co-option of TMPRSS2-ERGReviewers' Comments:

Reviewer #1:

Remarks to the Author:

In this study, the authors found a strong eQTL for HNF1B and multiple potential causal variants involved in the regulation of HNF1B expression in PCa, and revealed the HNF1B eQTL signal is TMPRSS2-ERG fusion status dependent. They then investigated the role of HNF1B and found its involvement in several pathways related to cell cycle progression and PCa severity. Furthermore, HNF1B interacted with TMPRSS2-ERG to co-occupy large proportion of genomic regions with a remarkable enrichment of additional PCa risk alleles. They finally showed that HNF1B co-opts ERG fusion to mediate mechanistic and biological effects of the PCa risk-associated locus 17p13.3/VPS53/FAM57A/GEMIN4. They concluded that an extensive germline-somatic interaction between TMPRSS2-ERG fusion and genetic variations underpins PCa risk association and progression. The work seems interesting. However, there are many vague descriptions of experimental details so that it is hard for a reviewer to follow sometimes. I have the following comments.

Major

1. The authors need to specify the properties of each of PCa cell lines used in this study. How is each relevant to TMPRSS2-ERG fusion? In other words, the authors should use a TMPRSS2-ERG negative PCa as a control for many of experiments conducted in this study.
2. Why do authors perform one type of experiments in on one cell line, then others in another cell line? For example, in Fig 2, CRISPR/Cas9-mediated genome editing was performed in V16A cell line, enhancer report assays was done in LNCaP cell line, then genomic deletion of the SNP-enhancer regions through CRISPR/Cas9-mediated KO at single cell levels was conducted in 22Rv1 cell line. These functional experiments should be designed in one cell line then expand to other cell lines.
3. In Fig 3d, ChIP-seq of histone marks used were from LNCaP cells. The authors should use the data from VCaP cells since these two cell lines are very different.
4. In Fig 3e-g, what cell line were the ChIP-qPCR performed?
5. For co-IP experiment, the authors should use a Nuclease (like Benzonase) treatment to rule out that the interaction between ERG and HNF1B mediated by a DNA/RNA moiety.
6. For ChIP-seq, what are the antibodies and sequencing depth, replicates?
7. For 3C-qPCR, how many technique replicates were performed for each point? What is p-value? What is EHT-treatment (ethanol)? How does this experiment strengthen the conclusion?
8. The discussion is oversimplified. The authors should expand it and emphasize the advancement of this study in comparison to other studies by giving some examples. The authors may discuss a little bit about the translational application in the future.

Minor

9. The title seems too bigger and needs to be toned down such as using "contribute to prostate cancer progression" kinda of words instead of "drive"...
10. Suppl. Tables should be included in Suppl. Information with Suppl. Figures.
11. What sequencing data are generated and have the data deposited into GEO?

Reviewer #2:

Remarks to the Author:

This study by Giannareas et al., entitled "Extensive germline-somatic interplay drives prostate cancer through HNF1B co-option of TMPRSS2-ERG" provides evidence of how co-occurrence of some germline variants at the HNF1B genomic locus and somatic TMPRSS2-ERG fusion in prostate cancer (PCa) may

be involved in tumor progression. The work is important and highlights HNF1B as a PCa risk associated gene and prioritizes risk variants at the 17q12 and 17p13.3 loci. The authors show that in fusion-positive PCa, ERG is able to bind the 17q12 locus and induce HNF1B expression. Expression of HNF1B increases especially when the locus presents certain SNPs alleles which are located at HNF1B enhancers regions. In turn, HNF1B overexpression regulates cell-cycle genes that correlates with PCa progression in the clinical setting. The authors also report physical interaction between ERG and HNF1B at the chromatin level and overlapping cistromes enriched in further risk SNPs within the 17p13.3 affecting ERG/HNF1B downstream target genes (VPS53, FAM57A, and GEMIN4) that are important for PCa cells growth.

The manuscript is clearly written and observations in cultured cells are linked to clinical data on patients. The data is novel, robust, and accompanied by logical interpretations of the results. Identifying interplay between germline variation and observed somatic mutations in cancer is of great importance since it will provide a deeper understanding of mechanisms underlying disease initiation and progression with possible clinical applications.

By reading the article I was left wandering about few major aspects:

1) It remains unclear whether it is only ERG or also some other transcription factors that can cooperate with ERG to regulate HNF1B gene. For instance, in fig 6f the authors illustrate how the ERG-HNF1B binding to chromatin is possibly affected by the alternative allele of rs684232. Among the HNF1B and ERG co-bound genes in fig 5f, HNF1B itself is also listed, but this is not reported by the authors (the heatmap shows couple of strong binding sites). Observing Fig 3b, I wonder whether the HNF1B binding site in the HNF1B locus could be in proximity of any of the ERG binding site validated? Is the HNF1B binding site in any of the reported enhancers? Is the binding site for HNF1B at the HNF1B locus at any of the risk SNPs studied? Maybe a detailed analysis of the DNA binding motifs in the locus could help to clarify. The co-binding of ERG-HNF1B at the HNF1B locus could be reported/validated as it is done for TFF1, RASSF7, and POLR3F in Fig. 5g.

2) Wang et al (<https://www.nature.com/articles/s41388-019-1065-2>) and Lu et al (<https://onlinelibrary.wiley.com/doi/full/10.1111/jcmm.16081>) reported functional studies in AR negative cell lines showing that HNF1B overexpression is leading to inhibition of cell proliferation. How do the authors explain these results in light of their functional validations? The authors should clarify what is the role of androgens and the androgen receptor (AR) in the mechanisms (chromatin-related and non) uncovered in this study and whether the androgens/AR axis can interfere with HNF1B functions. Also, is the interaction between ERG and HNF1B abolished in absence of androgens (line 283-284)? If that is the case, this could be shown and emphasized as this might have therapeutic implications. Can enzalutamide be used to see whether the interactions are affected?

3) Lastly, a question that remains to be answered is the frequency of co-occurrence of fusion positive PCa tumors that present SNPs alleles increasing risks of aggressive disease. Can the authors comment on this or provide an estimate?

..and some minor details:

1) Line 94-96: the study does not provide evidence of the involvement of HNF1B in PCa development but rather on its involvement in progression as well as a predisposing factor to aggressive disease. I suggest to amend the sentence.

2) In Wang et al, HA-Tagged-HNF1B-ChIP-seq was performed. What is the overlap between the dataset presented in this study and the dataset in Wang et al? Is the cistrome of HNF1B in AR negative PCa cells extensively overlapping with the cistrome of ERG?

Reviewer #3:

Remarks to the Author:

As the authors note, prostate cancer has amongst the highest estimates of heritability of any malignancy. GWAS studies have identified 269 risk variants consistently associated with risk in multiethnic populations. The current investigation focuses on understanding the mechanistic aspects of a subset of these loci, specifically HNF1B. The study uses both experimental and computational approaches to investigate. The methods for the computational studies are clear and there is level of innovation in the study's focus.

While understanding the functional significance of these risk SNPs is a key next step, the overall rationale for the focus on TMPRSS2:ERG's connection of this study is not clearly laid out. The authors select HNF1B given its role more broadly across cancer types, which is reasonable. However, the selection to focus on TMPRSS2:ERG is not clearly motivated, particularly since it is a somatic event specific to prostate cancer. The framing of the introduction is out of balance with the description of the findings in the TCGA cohorts.

Given the focus on TMPRSS2:ERG, it is then unclear the inclusion of many of the cell lines for investigation, since most of these do not contain the gene fusion event. A study by Mertz et al, 2007 describes this in detail. The VCaP cell line is reasonable given it contains TMPRSS2:ERG, but also molecularly is characterized by AR V7, high AR, etc. This should be taken into account in interpreting these cell line studies.

The authors motivate the study in part by "clinical significance" of this locus, yet the data in Supp 1 are weak and based on unclear, small datasets likely cherry-picked for this findings. Similarly, the clinical significance of the HNF1B signature -- is there associations with survival simply because it is correlated with Gleason score? The outcomes in TCGA are a mixed bag and not validated. Moreover, since TMPRSS2:ERG on its own is not prognostic, how does this translate to the apparent co-operation between ERG and HNF1B?

One key piece that is missing from this study is whether SNPs in HNF1B are enriched/exclusive to TMPRSS2:ERG positive tumors in humans. This would help establish the relevance of the experimental findings to humans.

Below are some specific comments:

Introduction:

line 51: the 57% refers to estimate of heritability and not familial (familiar) risk.

line 79-81: the majority of studies do not find TMPRSS2:ERG to be prognostic or associated with high grade cancers.

Results:

line 118-120: Why is this finding remarkable? This has been demonstrated previously in GWAS studies.

line 129-131: While the comparison of HNF1B shows higher expression in prostate cancer, it is not strikingly different and appears to be in the top 25% of cancer types (against an idea of specificity).

lines 148-151: Would suggest greater caution in interpreting survival analyses given the very small sample size, lack of significance, and lack of clarity on outcomes in this particular clinical dataset (there are better datasets in cBIOPORTAL to investigate survival. Why this one?)

NCOMMS-21-23625: Extensive germline-somatic interplay contributes to prostate cancer progression through HNF1B co-option of TMPRSS2-ERG

General response to the reviewers' comments.

We thank the Referees for their valuable questions and suggestions to improve our manuscript. We have now considerably strengthened the data analysis, repeated and added extensive experiments in response to the points raised by the referees (see the following pages).

We feel that the added data both further validates and strengthens our original findings, and significantly increases the impact of the manuscript. In the following pages is our point-by-point response to all specific criticisms of the reviewers. The reviewers' comments are in *italic*, and our response to them is in roman text, with the changes made to the manuscript indicated in **Bold**. Affected lines, paragraphs, figures and datasets are also indicated.

REVIEWER COMMENTS

Reviewer #1 (Remarks to the Author):

In this study, the authors found a strong eQTL for HNF1B and multiple potential causal variants involved in the regulation of HNF1B expression in PCa, and revealed the HNF1B eQTL signal is TMPRSS2-ERG fusion status dependent. They then investigated the role of HNF1B and found its involvement in several pathways related to cell cycle progression and PCa severity. Furthermore, HNF1B interacted with TMPRSS2-ERG to co-occupy large proportion of genomic regions with a remarkable enrichment of additional PCa risk alleles. They finally showed that HNF1B co-opts ERG fusion to mediate mechanistic and biological effects of the PCa risk-associated locus 17p13.3/VPS53/FAM57A/GEMIN4. They concluded that an extensive germline-somatic interaction between TMPRSS2-ERG fusion and genetic variations underpins PCa risk association and progression. The work seems interesting. However, there are many vague descriptions of experimental details so that it is hard for a reviewer to follow sometimes. I have the following comments.

Major

1. The authors need to specify the properties of each of PCa cell lines used in this study. How is each relevant to TMPRSS2-ERG fusion? In other words, the authors should use a TMPRSS2-ERG negative PCa as a control for many of experiments conducted in this study.

Response: Thank you for the helpful comments and suggestions. For PCa cell lines used in this study, in addition to VCaP cells harboring TMPRSS2-ERG fusion, the others lack the fusion. According to your suggestions, we have now added the TMPRSS2-ERG negative PCa cell line, LNCaP as control in several experiments. Similar to VCaP, LNCaP cells are androgen-sensitive human PCa cell line. Relative mRNA levels of ERG in LNCaP is nearly

10000-fold lower than that of VCaP **as shown in the figure below**. We performed siRNA-mediated knockdown of ERG in LNCaP and observed no association of ERG expression with any of HNF1B, VPS53, FAM57A and GEMIN4 (**Supplementary Fig. 3f** line 236 and **Supplementary Fig. 10c** line 425); in contrast, the results showed positive associations in the TMPRSS2-ERG positive VCaP cells (**Fig. 3c** line 236 and **Fig. 6h** line 422). Moreover, we performed also the ChIP-qPCR analysis in LNCaP. In contrast to strong enrichment of ERG, HNF1B, and active histone modifications H3K4me1/2/3 at given HNF1B locus variants in VCaP (**Fig. 3e-i** line 247), we observed no occupancy of ERG and HNF1B, and much less enrichment of active histone marks at these regions (**Supplementary Fig. 3g-k** line 249). We found similar results, with strong binding of ERG and HNF1B at the 17q12 locus (**Supplementary Fig. 8h** line 366) and 17p13.3 locus variants in VCaP (**Fig. 6g** line 412) but not in LNCaP (**Supplementary Fig. 8i** line 366 and **Supplementary Fig. 10b** line 412). Similar results were obtained when compared the chromatin occupancies of ERG and HNF1B at the genes TFF1, RASSF7, and POLR3F in VCaP and LNCaP, respectively (**Supplementary Fig. 8j,k** line 366).

2. Why do authors perform one type of experiments in on one cell line, then others in another cell line? For example, in Fig 2, CRISPR/Cas9-mediated genome editing was performed in V16A cell line, enhancer report assays was done in LNCaP cell line, then genomic deletion of the SNP-enhancer regions through CRISPR/Cas9-mediated KO at single cell levels was conducted in 22Rv1 cell line. These functional experiments should be designed in one cell line then expand to other cell lines.

Response: We agree with the Referee's suggestions and have now performed these experiments in more than one PCa cell line. Notably, CRISPR/Cas9-mediated genome editing followed by single clone isolation (**Fig. 2e** line 213) could only be carried out in 22Rv1 cells since only this cell line can survive and be expanded from a single selected clone. We thus could not manage to expand this single-clone selection experiment in additional PCa cell lines.

We expanded CRISPR/Cas9-mediated genome editing experiments with mixed clone selection (initially in V16A cells) and enhancer report assays (initially in LNCaP cells) in 22Rv1 cell line (**Fig. 2c** line 196 and **Fig. 2d** lines 203, 209). The results showed a similar pattern but with rather lower expression levels of HNF1B in the mixed clones of 22Rv1 (**Fig. 2c** line 196) compared to that of V16A cells line (**Supplementary Fig. 2p** line 196) upon

deletion of a given SNP-containing genomic region. For the expanded enhancer reporter assays in 22Rv1 cells, the results showed that, compared to HNF1B promoter, rs11651052, rs12453443, rs9901746, rs7405696 or rs11263763-containing region indicates enhancer-like function to activate luciferase gene transcription (**Fig. 2d** lines 203, 209). Moreover, the rs9901746 region shows a 3'-5' orientation-dependent enhancer activity for both alleles in 22Rv1 cells (**Fig. 2d** lines 203, 209) while the results in LNCaP cells show a slight different rs9901746 allele-specific activity (**Supplementary Fig. 2q** lines 203), probably due to expression of different TFs that can bind to the SNP region in the two cell lines. Overall, the enhancer assay results in both PCa cell lines are well consistent and support our observed association of the studied SNPs with the target gene HNF1B (as shown in the figures below).

Giannareas & Zhang et al Figure 2

Giannareas & Zhang et al Figure S2

3. In Fig 3d, ChIP-seq of histone marks used were from LNCaP cells. The authors should use the data from VCaP cells since these two cell lines are very different.

Response: We thank the reviewer for pointing out this issue. We do agree that ChIP-seq data of histone marks from VCaP cells should be shown. We have now replaced data on histone marks from LNCaP with VCaP cell line (**Fig. 3d** lines 242, 438), and observed similar pattern of active histone marks at regions of the 17p13 HNF1B locus.

4. In Fig 3e-g, what cell line were the ChIP-qPCR performed?

Response: These ChIP-qPCR experiments were performed in VCaP cell line. Relevant to the comments made by Reviewer #2, we repeated the experiments in VCaP under the conditions of ethanol (ETH) or dihydrotestosterone (DHT) treatment to identify if there are any difference for transcription factor binding and histone mark enrichment at each SNP region (**Fig. 3e-i** line 247).

5. For co-IP experiment, the authors should use a Nuclease (like Benzonase) treatment to rule out that the interaction between ERG and HNF1B mediated by a DNA/RNA moiety.

Response: We thank the Reviewer for this insightful comments on the physical interaction between ERG and HNF1B. We have now performed Benzonase treatment, and can still observe the interaction of ERG and HNF1B, thereby concluding that the interaction between ERG and HNF1B was not mediated by a DNA/RNA moiety (**Supplementary Fig. 8a** line 339).

6. For ChIP-seq, what are the antibodies and sequencing depth, replicates?

Response: We thank the reviewer for this question. We have now performed ERG and HNF1B replicate. For HNF1B ChIP-seq data in the **Fig. 5e** (lines 347, 356), the experiment was conducted in 2014 (**HNF1B 2014**), and the antibody against HNF1B was purchased from Santa Cruz Biotechnology (sc-22840X). The assays generated 19137704 single-end reads of 32 bp in length. For the input control, it generated 13814734 single-end reads of 33bp in length. The ERG ChIP-seq data used in the **Fig. 5e** (lines 347, 356) was a combination of 15 ERG ChIP-seq experiments in VCaP retrieved from cistrome DB to obtain a union of ERG binding sites (**shown as the table below**). We have now performed independent replicate for HNF1B (2021) and ERG (2021) with corresponding inputs in VCaP cells under DHT and ETH treatment conditions, respectively. The summary of each ChIP-seq experiment is listed in the table below. For the 15 ERG ChIP-seq retrieved from published data, the ERG antibodies were originated from Santa Cruz or Epitomics. The ERG antibody that we used over the current paper revision was purchased from Abcam (ab92513). The HNF1B antibody used in the HNF1B ChIP-seq experiment (HNF1B 2014) was purchased from Santa Cruz (sc-22840X); however, this HNF1B antibody was discontinued

(<https://www.scbt.com/p/hnf-1beta-antibody-h-85>), similar as the above ERG antibody and other rabbit polyclonal antibodies from Santa Cruz Biotechnology.

We applied the same HNF1B antibody (sc-22840X) that has been purchased and stored since 2014 in our new replicate of HNF1B ChIP-seq experiment, the total number of identified peaks in newly-performed HNF1B experiment (HNF1B 2021) was relatively less. This is likely to be attributed to the reduced activity of HNF1B antibody with over 7-years' storage. Nonetheless, we still observed the most enriched motif as HNF1B (**Fig. A**) and comparable proportion of peak overlaps between ERG and HNF1B ChIP-seq data. As shown below in **Fig. B**, there was 43.4% of HNF1B peaks (HNF1B 2014) overlapping with that of newly-generated ERG ChIP-seq peaks (top-enriched motif as ERG, **Fig. A**). Consistently, we found 49.5% of newly-generated HNF1B peaks (HNF1B 2021) sharing common binding regions with that of the newly-generated ERG peaks (**Fig. C**), which is highly comparative to that of the first HNF1B experiment performed in 2014 (**Fig. 5e** lines 347, 356). The proportion of peak overlapping between ERG and HNF1B in ETH condition was 41.15% (**Fig. D**), which is comparable to that of under DHT condition (**Fig. C**). We also checked the common chromatin binding sites between our newly-performed ERG ChIP-seq and the ERG ChIP-seq data curated from the published ones shown in **Fig. 5e** (lines 347, 356). The result indicated high overlap with 83.44% under DHT (**Fig. E**) and 85.87% under ETH conditions (**Fig. F**), respectively.

ChIP-seq assays in VCaP	GEO/ENA ID	Antibody vendor	Source	Identification	Library Layout	Length	Reads	Treatment time
HNF1B DHT	PRJEB46088	Santa Cruz (sc-22840X)	This study	2014 January	SINGLE	32	19137704	24hr (DHT)
Input DHT	PRJEB46088		This study	2014 January	SINGLE	33	13814734	
HNF1B DHT	PRJEB49662	Santa Cruz (sc-22840X)	This study	2021 October	SINGLE	150	86058984	24hr (DHT)

ERG DHT	PRJEB49662	Abcam (ab92513)	This study	2021 October	SINGLE	150	93137288	24hr (DHT)
Input DHT	PRJEB49662		This study	2021 October	SINGLE	150	107422118	
HNF1B ETH	PRJEB49662	Santa Cruz (sc-22840X)	This study	2021 October	SINGLE	150	93847628	24hr (ETH)
ERG ETH	PRJEB49662	Abcam (ab92513)	This study	2021 October	SINGLE	150	105600402	24hr (ETH)
Input ETH	PRJEB49662		This study	2021 October	SINGLE	150	94491797	
ERG VCaP	GSM717395	Santa Cruz (sc-353)	Cistrome DB	8654	SINGLE	76	18592769	0hr (DHT)
ERG VCaP DHT	GSM717396	Santa Cruz (sc-353)	Cistrome DB	8655	SINGLE	76	17236806	2hr (DHT)
ERG VCaP DHT	GSM717397	Santa Cruz (sc-353)	Cistrome DB	8656	SINGLE	76	21108187	18hr (DHT)
ERG VCaP R1881	GSM1193658	Santa Cruz (sc-353)	Cistrome DB	47409	SINGLE	36	88996460	(N.A)hr (R1881)
ERG VCaP DHT	GSM1328978	Epitomics (2805-1)	Cistrome DB	44603	PAIRED	111	25396931	12hrs (DHT)
ERG VCaP DHT	GSM1328980	Epitomics (2805-1)	Cistrome DB	44605	PAIRED	111	27917153	12hrs (DHT)
ERG VCaP DHT	GSM1328981	Epitomics (2805-1)	Cistrome DB	44606	PAIRED	111	31109617	12hrs (DHT)
ERG VCaP DHT	GSM2086315	Epitomics (2805-1)	Cistrome DB	69266	SINGLE	75	22075079	24hr (DHT)
ERG VCaP DHT	GSM2086314	Epitomics (2805-1)	Cistrome DB	69267	SINGLE	75	26469470	24hr (DHT)
ERG VCaP DHT	GSM2086313	Epitomics (2805-1)	Cistrome DB	69268	SINGLE	75	19790485	24hr (DHT)
ERG VCaP DHT	GSM2086312	Epitomics (2805-1)	Cistrome DB	69269	SINGLE	75	20510758	24hr (DHT)
ERG VCaP DHT	GSM2086311	Epitomics (2805-1)	Cistrome DB	69270	SINGLE	75	19481746	24hr (DHT)
ERG VCaP DHT	GSM2086310	Epitomics (2805-1)	Cistrome DB	69271	SINGLE	75	24419416	24hr (DHT)
ERG VCaP DHT	GSM2086309	Epitomics (2805-1)	Cistrome DB	69272	SINGLE	75	25995622	24hr (DHT)
ERG VCaP R1881	GSM353637	Santa Cruz (SC354X)	Cistrome DB	2918	SINGLE	36	8419264	16hr (R1881)

7. For 3C-qPCR, how many technique replicates were performed for each point? What is p-value? What is EHT-treatment (ethanol)? How does this experiment strengthen the conclusion?

Response: Thank you for the comments. Given a rather complex procedure of 3C-qPCR and to ensure reproducibility of this assay, we have already performed three times of independent experiments for 3C-qPCR (**shown as the figure below**), which are overall highly consistent. In each independent experiment, three technical replicates were performed and the p-values were calculated with significance marked in **Fig. 7b**, line 470 (shown as one independent assay) in comparison with that of VCaP cells treated with ethanol (ETH). The 3C-qPCR showed the direct chromatin interactions between rs2955626/rs684232/rs461251-containing enhancer and proximal regulatory regions of VPS53, FAM57A and GEMIN4, especially the 3rd HindIII cut site (GRCh37/hg19 chr17:637,677) and the 4th HindIII cut site (GRCh37/hg19 chr17:641,016) located in FAM57a locus show a relative high crosslinking frequency with the rs2955626/rs684232/rs461251-containing enhancer.

3C-qPCR

8. The discussion is oversimplified. The authors should expand it and emphasize the advancement of this study in comparison to other studies by giving some examples. The authors may discuss a little bit about the translational application in the future.

Response: Thank you for this valuable comment. We have now expanded the discussion and propose potential translational application in the future. Please find the updated discussion on pages 14-16.

Minor

9. The title seems too bigger and needs to be toned down such as using “contribute to prostate cancer progression” kinda of words instead of “drive”...

Response: We agree with the Referee and changed the title of the manuscript as “Extensive germline-somatic interplay contributes to prostate cancer progression through HNF1B co-option of TMPRSS2-ERG”.

10. Suppl. Tables should be included in Suppl. Information with Suppl. Figures.

Response: We thank the reviewer for the kind advice. We have now converted and included the 17 supplementary tables and supplementary figures in supplementary information.

11. What sequencing data are generated and have the data deposited into GEO?

Response: We thank the reviewer for the kind suggestion. We generated ChIP-seq and RNA-seq data in this study. The siRNA-mediated HNF1B knockdown followed by RNA-seq was performed with two biological replicates, and the data has been deposited in the European Nucleotide Archive (ENA) under accession number PRJEB46082. For the ChIP-seq data on HNF1B that was performed in 2014 has been deposited in ENA under accessions PRJEB46088. The ChIP-seq data of HNF1B and ERG performed during the manuscript revision has been archived under accession ID PRJEB49662. The data releasing date was set on 30.04.2022.

Reviewer #2 (Remarks to the Author):

This study by Giannareas et al., entitled “Extensive germline-somatic interplay drives prostate cancer through HNF1B co-option of TMPRSS2-ERG” provides evidence of how co-occurrence of some germline variants at the HNF1B genomic locus and somatic TMPRSS2-ERG fusion in prostate cancer (PCa) may be involved in tumor progression. The work is important and highlights HNF1B as a PCa risk associated gene and prioritizes risk variants at the 17q12 and 17p13.3 loci. The authors show that in fusion-positive PCa, ERG is able to bind the 17q12 locus and induce HNF1B expression. Expression of HNF1B increases especially when the locus presents certain SNPs alleles which are located at HNF1B enhancers regions. In turn, HNF1B overexpression regulates cell-cycle genes that correlates with PCa progression in the clinical setting. The authors also report physical interaction between ERG and HNF1B at the chromatin level and overlapping cistromes enriched in further risk SNPs within the 17p13.3 affecting ERG/HNF1B downstream target genes (VPS53, FAM57A, and GEMIN4) that are important for PCa cells growth.

The manuscript is clearly written and observations in cultured cells are linked to clinical data on patients. The data is novel, robust, and accompanied by logical interpretations of the results.

Identifying interplay between germline variation and observed somatic mutations in cancer is of great importance since it will provide a deeper understanding of mechanisms underlying disease initiation and progression with possible clinical applications.

By reading the article I was left wandering about few major aspects:

1) It remains unclear whether it is only ERG or also some other transcription factors that can cooperate with ERG to regulate HNF1B gene. For instance, in fig 6f the authors illustrate how the ERG-HNF1B binding to chromatin is possibly affected by the alternative allele of rs684232. Among the HNF1B and ERG co-bound genes in fig 5f, HNF1B itself is also listed, but this is not reported by the authors (the heatmap shows couple of strong binding sites). Observing Fig 3d, I wonder whether the HNF1B binding site in the HNF1B locus could be in proximity of any of the ERG binding site validated? Is the HNF1B binding site in any of the reported enhancers? Is the binding site for HNF1B at the HNF1B locus at any of the risk SNPs studied? Maybe a detailed analysis of the DNA binding motifs in the locus could help to clarify. The co-binding of ERG-HNF1B at the HNF1B locus could be reported/validated as it is done for TFF1, RASSF7, and POLR3F in Fig. 5g. It remains unclear whether it is only ERG or also some other transcription factors that can cooperate with ERG to regulate HNF1B gene.

Response: We appreciate for the valuable suggestions. We have now added HNF1B and ERG chromatin co-binding profile at HNF1B gene as shown in **Fig. 5g** (lines 362, 366). To see whether there is additional transcription factor binding at HNF1B, we searched for the Cistrome DB, representing thus far largest collection database of published ChIP-seq data^{1,2}. As shown in the table below, in addition to ERG, FOXA1 shows chromatin occupancy at the HNF1B regions (**shown as the table and Fig. A below**), suggesting it might be involved in

the regulation of HNF1B. We thus performed co-expression analysis similar as what we have done for ERG and HNF1B in the clinical prostate cancer expression profiling data. However, unlike ERG that is top correlated with HNF1B expression in PCa tumor samples, this co-expression analysis indicated no clear expression correlation between FOXA1 and HNF1B in multiple independent PCa datasets (**Fig. B-D**). Moreover, we performed siRNA-mediated knockdown of FOXA1 in LNCaP and VCaP cells. While the results showed positive regulation of FOXA1 for HNF1B in LNCaP (**Fig. E**), the results from two different siRNA against FOXA1 indicated different effect on the expression of HNF1B (**Fig. F**). Here with these inconsistent trends on FOXA1 to HNF1B in either the clinical PCa datasets or the experimental PCa cell lines, we concluded that ERG is the most plausible transcriptional regulator of HNF1B as we documented in the manuscript.

GSM_ID	Factor	Biosource	RP_score
GSM2537226	FOXA1	VCaP;Epithelium;Prostate	0.660546
GSM1354836	FOXA1	VCaP;Epithelium;Prostate	0.616284
GSM1354837	FOXA1	VCaP;Epithelium;Prostate	0.593755
GSM2537227	FOXA1	VCaP;Epithelium;Prostate	0.587659
GSM2537225	FOXA1	VCaP;Epithelium;Prostate	0.581905
GSM717397	ERG	VCaP;Epithelium;Prostate	0.580632
GSM2537229	FOXA1	VCaP;Epithelium;Prostate	0.573766
GSM2537230	FOXA1	VCaP;Epithelium;Prostate	0.554813
GSM1354839	FOXA1	VCaP;Epithelium;Prostate	0.415635
GSM1068136	FOXA1	LNCaP;Epithelium;Prostate	0.594602
GSM1068137	FOXA1	LNCaP;Epithelium;Prostate	0.513472
GSM1691164	FOXA1	LNCaP;Epithelium;Prostate	0.484069
GSM2219861	FOXA1	LNCaP;Epithelium;Prostate	0.476365
GSM2219867	FOXA1	LNCaP;Epithelium;Prostate	0.453855
GSM2219860	FOXA1	LNCaP;Epithelium;Prostate	0.451737
GSM916524	FOXA1	LNCaP;Epithelium;Prostate	0.448772
GSM1891830	FOXA1	LNCaP;Epithelium;Prostate	0.448329
GSM2219863	FOXA1	LNCaP;Epithelium;Prostate	0.435243
GSM1691165	FOXA1	LNCaP;Epithelium;Prostate	0.433547
GSM2219864	FOXA1	LNCaP;Epithelium;Prostate	0.432157
GSM2219866	FOXA1	LNCaP;Epithelium;Prostate	0.414227
GSM1691142	FOXA1	LNCaP;Epithelium;Prostate	0.396627

With newly-generated HNF1B ChIP-seq replicate, we identified two more HNF1B binding sites in the HNF1B locus (see **Fig. 3d** lines 242, 438). We experimentally validated the three common binding (CB) sites of ERG and HNF1B at HNF1B locus with ChIP-qPCR in VCaP cells (**Supplementary Fig. 8g** lines 362, 364 and **Supplementary Fig. 8h** line 366).

We found also several SNPs including rs11649743, rs9901746, rs11263761, rs4430796, and rs11263763 residing in close proximity to HNF1B binding sites in the HNF1B locus. By performing ChIP-qPCR in VCaP cell line, we verified that HNF1B is significantly enriched at the rs11263763-containing region (**Fig. 3g** line 247). This holds true for the binding of HNF1B at the rs11263763 region in LNCaP cells (**Supplementary Fig. 3h** line 249). In addition, by ChIP-qPCR, we observed the occupancy of HNF1B in VCaP cells at additional HNF1B variants rs12453443 and rs718960 that are not close to the HNF1B ChIP-seq peaks in the HNF1B locus (**Fig. 3e** line 247 and **Fig. 3i** line 247).

We next performed the DNA binding motif analysis for the three common binding (CB) sites of ERG and HNF1B at the 17q12 HNF1B locus (**Supplementary Fig. 8g** lines 362, 364). As expected, the motif analysis shows that both ERG and HNF1B motifs exist at the ChIP-seq peaks (**Supplementary Fig. 8g, lower panel** lines 362, 364).

Through the motif analysis for the 17q12 SNPs in the HNF1B locus, we identified four SNPs including rs4430796, rs718960, rs8064454, and rs11651052 might alter HNF1B chromatin binding (**Supplementary Fig. 3k** line 249). In the manuscript, we have validated the SNPs in the HNF1B locus that are likely to be causal variants in regulating HNF1B expression based on the eQTL analysis (**Fig. 2a** lines 189,190, 262) via CRISPR/Cas9 KO (**Fig. 2c** line 196 and **Fig. 2e** line 213) and enhancer reporter assays (**Fig. 2d** line 203, 209). Taken together, these results suggest that the four SNPs might influence HNF1B expression by modulating DNA binding motif of HNF1B and ERG in the HNF1B locus despite that these are not perfectly residing in their ChIP-seq peaks. It has been proposed that variations at SNPs may influence local, proximal, or distal TF-DNA binding motifs³ thereby leading to variations in gene expression and ultimately causing phenotypic diversities.

Lastly, we have added the common binding site of HNF1B and ERG in the HNF1B gene to **Fig. 5g** (lines 362, 366) as described above. We have also performed ChIP-qPCR verification for the common binding sites of HNF1B and ERG at the genes TFF1, RASSF7 and POLR3F (**Fig. 5g** lines 362, 366) in both of VCaP and LNCaP cell lines (**Supplementary Fig. 8j,k** line 366).

2) Wang et al (<https://www.nature.com/articles/s41388-019-1065-2>) and Lu et al (<https://onlinelibrary.wiley.com/doi/full/10.1111/jcmm.16081>) reported functional studies in AR negative cell lines showing that HNF1B overexpression is leading to inhibition of cell proliferation. How do the authors explain these results in light of their functional validations? The authors should clarify what is the role of androgens and the androgen receptor (AR) in the mechanisms (chromatin-related and non) uncovered in this study and whether the androgens/AR axis can interfere with HNF1B functions. Also, is the interaction between ERG and HNF1B abolished in absence of androgens (line 283-284)? If that is the case, this could be shown and emphasized as this might have therapeutic implications. Can enzalutamide be used to see whether the interactions are affected?

Response: Thank you for your valuable comments. Based on the pinpointed publications, we have now performed the cell proliferation assays upon siRNA-mediated knockdown of HNF1B in the AR negative cell lines PC3 and DU145 as described in the two studies. In parallel, we have performed similar experiments in the AR positive cell lines VCaP and LNCaP. To ensure the reproducibility, we have now performed four independent cell

proliferation experiments (shown as upper panel of the figures below). Our data consistently showed that knockdown of HNF1B expression markedly attenuate cell growth and viability in comparison with cells harboring control siRNA in the proliferation assays. One of the independent results are presented in **Supplementary Fig. 1b-i** (line 161). Thus, the results in the AR negative cell lines PC3 and DU145 are inconsistent with that described in the two articles. We reasoned that the differences may come from these issues: according to the previous study by Bach and Yaniv⁴, there are three isoforms of HNF1B, of the two isoforms tend to be transcriptional activators, whereas the shortest isoform is a transcriptional repressor. Therefore, we proposed that the targeting siRNAs to specific HNF1B isoform may cause the difference. We thus tried to find the detailed information about the siRNAs used in Wang's paper, but unfortunately, we could not find these siRNA sequence information. On our side, the siRNAs against HNF1B we used are targeting all the three isoforms of HNF1B. Thus, without details in siRNAs against HNF1B in the published work, we could not conclusively explain the inconsistent results among the studies.

Independent experiment 1 (Supplementary Fig. 1c, e, g, i)

Independent experiment 2

Independent experiment 3

Independent experiment 4

Next, according to your insightful recommendations we tried to clarify the role of androgens and AR in our study. We thus performed ChIP-qPCR assays in VCaP and LNCaP treated with or without androgens (DHT). Relevant results are presented in **Fig. 3e-i** (line 247), **Fig. 6g** (line 412), **Supplementary Fig. 3g-k** (line 249), **Supplementary Fig. 8h-k** (line 366) and **Supplementary Fig. 10b** (line 412). In some cases, we can observe stronger binding affinity of the TFs upon androgen treatment. The notable changes are all the TFs ERG, HNF1B and AR occupancy at rs12453443, rs7405696, rs11651052 or rs718960 SNP regions as shown in **Fig. 3e, f, h and i** (line 247). In addition, HNF1B and ERG at their binding sites within 17p13.3 locus and HNF1B locus had higher fold enrichment in VCaP cells under DHT treatment than that of without androgen stimulation (vehicle control ethanol, ETH) (**Fig. 6g** line 412, **Supplementary Fig. 8h,i** line 366 and **Supplementary Fig. 10b** line 412). Moreover, we observed slight enrichments of AR occupancy at rs12453443, rs7405696, rs11263763, rs11651052 and rs718960 SNP regions in VCaP and LNCaP cell lines (see **Fig. 3e-i** line 247 and **Supplementary Fig. 3g-k** line 249).

Lastly, we revealed interesting results according to your recommendation to test whether the interaction between ERG and HNF1B can be influenced by androgen treatment. While we did not observe that the interaction between ERG and HNF1B abolished in absence of androgens, we detected partially abolished interaction upon treated with enzalutamide (**Fig. 5a** line 337, **Fig. 5b** line 340), and slight downregulation of ERG and HNF1B (**Supplementary Fig. 8b** line 341), indicating possible therapeutic implications. Consistent to this, below we also presented another independent experimental result upon enzalutamide treatment (0 μ M, 10 μ M).

3) Lastly, a question that remains to be answered is the frequency of co-occurrence of fusion positive PCa tumors that present SNPs alleles increasing risks of aggressive disease. Can the authors comment on this or provide an estimate?

Response: Thank you for this insightful question. We firstly investigated whether the 13 PCa risk-associated SNPs within 17p12 HNF1B locus are greatly enriched in TMPRSS2-ERG (T2E) fusion positive patients in the TCGA dataset consisting of 388 samples with matched genotype and expression data previously obtained (Nat Genet 2016). There are 228 fusion negative and 160 T2E positive samples. We checked the enrichment of SNP alleles in T2E fusion positive and negative PCa patients, respectively. However, the results showed no particular enrichment pattern for the SNP alleles in TMPRSS2:ERG (T2E) fusion-positive or -negative tumors (**shown as Fig. A below**). To examine whether the disease aggressiveness might be implicated in the co-occurrence, we next stratified patients

according to clinical features of biochemical relapse (BCR) (**Fig. B, C**) or metastasis status (**Fig. D, E**), and then checked enrichment of SNP alleles in T2E fusion-positive versus -negative tumors. The result indicated quite similar distribution pattern for SNP alleles in T2E fusion-positive and -negative tumors with/without BCR or metastasis.

To further investigate whether the SNPs were associated with TMPRSS2-ERG fusion, we performed an independent association study in a Chinese prostate biopsy cohort⁵ and radical prostatectomy cohort. Briefly, a consecutive prostate biopsy cohort and prostatectomy cohort with biospecimen started from October 2017 to December 2021 at a tertiary hospital in Shanghai, China. ERG protein abundance was measured in biopsy tissue samples via immunohistochemistry (IHC). Genotyping were performed in most of the samples from October 2017 to December 2021 using Illumina Asian Screening Array. Imputation was performed thereafter⁶. A total of 1069 samples with ERG protein expression information were found to have matched genotyping data (October 2017-December 2021). The demographic characteristics of these patients are shown as **Table 1** below, also can be found on line 270 as **Supplementary Table 2**.

We first performed a case-control (580 PCa vs 475 non-PCa) association analysis to see whether the HNF1B SNPs shows any associations with prostate cancer. We observed that the SNPs including rs4239217, rs8064454, rs7405696, rs9901746, and rs12453443 were significantly associated with prostate cancer (**Table 2** as shown below, also can be found on line 273 as **Supplementary Table 3**).

Assuming that positive ERG expression based on IHC is due to the TMPRSS2-ERG fusion, a total of 120 (11.2%) positive cases were observed in this cohort. Among the 120 samples, biopsy results of 13 men were negative to prostate cancer. Thus, there are 107 ERG expression positive tumors. We next performed a case only study (107 ERG+ PCa vs 473 ERG- PCa), and found a moderate but significant association of the HNF1B SNPs rs9901746 and rs12453443 with ERG expression positive tumors (**Table 3** as shown below, also can be found on line 277 as **Supplementary Table 4**). Dataset with larger sample size with stronger statistical power may reveal different impact on the co-occurrence of T2E fusion-positive and SNPs in disease severity, though a limitation of this association analysis is that IHC-derived positive ERG protein expression was used as surrogate for TMPRSS2-ERG fusion.

Table 1. Descriptive characteristics of Ruijin Biopsy cohort.

Characteristics	PCa Biopsy Group	Prostatectomy Group
Total number of patients	907	162
Age, yrs		
Median (IQR)	68 (63-75)	68.5 (64-73)
tPSA, ng/mL		
Median (IQR)	12.36 (7.58-25.05)	12.83 (7.48-27.92)
Prostate volume, mL		
Median (IQR)	81.55 (58.50-111.54)	63.95 (48.30-95.47)

ERG		
Negative	826 (91.1%)	123 (75.9%)
Positive	81 (8.9%)	39 (24.1%)
Family history of PCa		
Negative	884 (97.5%)	159 (98.1%)
Positive	23 (2.5%)	3 (1.9%)
Missing	0	0
Biopsy result		
Negative	489 (53.9%)	0
Positive	418 (46.1%)	162 (100%)
Grade Group		
1 (GS≤3+3)	95 (10.5%)	15 (9.3%)
2 (GS=3+4)	73 (8.0%)	55 (34.0%)
3 (GS=4+3)	80 (8.8%)	45 (27.8%)
4 (GS=8)	96 (10.6%)	10 (6.2%)
5 (GS=9 or 10)	60 (6.6%)	19 (11.7%)
Missing	8	18

Abbreviations: IQR, Interquartile range; tPSA, total prostate-specific antigen; PCa, prostate cancer; GS, Gleason Score.

Table 2. Association between HNF1B SNPs and PCa status (580 PCa vs 475 non-PCa)

SNP	Alt allele	Ref allele	Risk allele frequency	OR (95%CI)	P
SNPs at promoter region					
rs11649743	A	G	0.37	1.16(0.97-1.40)	0.11
rs718960	T	C	0.35	1.04(0.86-1.25)	0.71
rs11263761	G	A	0.22	1.08(0.86-1.36)	0.52
rs4430796	G	A	0.23	0.91(0.73-1.13)	0.39
rs4239217	G	A	0.05	10.44(4.98-21.88)	5.15E-10
rs8064454	A	C	0.06	2.94(1.91-4.53)	1.04E-06
rs7405696	G	C	0.15	1.75(1.35-2.27)	2.87E-05
rs11651052	A	G	0.25	NA	NA
rs9901746	G	A	0.15	1.65(1.27-2.14)	1.50E-04

rs11263763	G	A	0.24	NA	NA
rs11658063	C	G	0.24	NA	NA
rs12453443	C	G	0.16	1.60(1.24-2.06)	3.15E-04
rs3760511	Not able to be genotyped		-	-	-

Association was tested using logistic regression by adjusting for age

Table 3. Association between HNF1B SNPs and ERG expression tumors in PCa (107 ERG+ vs 473 ERG-)

SNP	Alt allele	Ref allele	Risk allele frequency	OR (95%CI)	P
SNPs at promoter region					
rs11649743	A	G	0.35	1.08 (0.80-1.46)	0.61
rs718960	T	C	0.32	1.03 (0.75-1.42)	0.85
rs11263761	G	A	0.24	0.76 (0.51-1.12)	0.16
rs4430796	G	A	0.24	1.05 (0.73-1.50)	0.81
rs4239217	G	A	0.11	1.19 (0.73-1.95)	0.48
rs8064454	A	C	0.12	1.37 (0.85-2.21)	0.20
rs7405696	G	C	0.20	1.34 (0.93-1.95)	0.12
rs11651052	A	G	0.25	0.96 (0.51-1.82)	0.90
rs9901746	G	A	0.20	1.45 (1.00-2.10)	4.82E-02
rs11263763	G	A	0.23	1.01 (0.53-1.93)	0.97
rs11658063	C	G	0.23	1.03 (0.54-1.96)	0.93
rs12453443	C	G	0.20	1.54 (1.07-2.22)	2.08E-02
rs3760511	Not able to be genotyped		-	-	-

..and some minor details:

1) *Line 94-96: the study does not provide evidence of the involvement of HNF1B in PCa development but rather on its involvement in progression as well as a predisposing factor to aggressive disease. I suggest to amend the sentence.*

Response: We fully agreed and amended the sentence as “We therefore focused on the 17q12/HNF1B locus and sought to identify functional causal variants of this region as well as investigate role of HNF1B in PCa progression and predisposition to aggressive disease...”

2) *In Wang et al, HA-Tagged-HNF1B-ChIP-seq was performed. What is the overlap between*

the dataset presented in this study and the dataset in Wang et al? Is the cistrome of HNF1B in AR negative PCa cells extensively overlapping with the cistrome of ERG?

Response: We thank the reviewer for suggesting a comparison of HNF1B ChIP-seq datasets across different PCa cell lines. The ChIP-seq experiment of HNF1B shown in **Fig. 5e** (lines 347, 356) was performed in 2014 with HNF1B antibody (sc-22840X) purchased from Santa Cruz. We recently conducted ERG and HNF1B ChIP-seq replicate under DHT or ETH conditions in VCaP cells (**please see our response to Reviewer #1, point #6**). The HNF1B antibody used in the recent HNF1B ChIP-seq replicate (HNF1B 2021) has been stored since 2014 as Santa Cruz stopped manufacturing rabbit polyclonal antibodies. The amount of identified peaks in recently performed HNF1B replicate was relatively less in comparison with HNF1B ChIP-seq conducted in 2014; the reason might be attributable to the loss of HNF1B antibody efficiency due to over 7-year' storage. Nonetheless, we still observed HNF1B as top ranked motif and comparable proportion of peak overlaps between ERG and HNF1B ChIP-seq data (see **Table A in reviewer #1 point #6**). As shown in **Fig. B in reviewer #1 point #6**, there was 43.4% of HNF1B peaks (HNF1B 2014) overlapping with that of ERG. Consistently, we found 49.5% of HNF1B peaks (HNF1B 2021) sharing common binding regions with that of ERG (**Fig. C in reviewer #1 point #6**), which is highly comparative to that of the first HNF1B experiment performed in 2014 (**Fig. B reviewer #1 point #6**). The proportion of peak overlapping between ERG and HNF1B in ETH condition was 41.1% (**Fig. D in reviewer #1 point #6**), which is slightly less but comparable to that of in DHT condition (**Fig. B-C in reviewer #1 point #6**). We also performed the peak overlap between our newly performed ERG ChIP-seq with the ERG ChIP-seq (**Fig. 5e** lines 347, 356) that was retrieved from Cistrome DB. The result indicated high overlap with 83.44% and 85.87% in DHT (**Fig. E in reviewer #1 point #6**) and ETH (**Fig. F in reviewer #1 point #6**) conditions, respectively.

HNF1B ChIP-seq in Wang et al was performed using HA-tag antibody in DU145 cells. As shown in **Fig. A (Supplementary Fig. 8e** line 352), there was 19.13% peak overlaps between HNF1B VCaP (DHT, 2014) and HA-tagged HNF1B in DU145 cells from Wang et al. From our recent HNF1B ChIP-seq replicate dataset in VCaP, we observed a comparable percentage of peak overlaps with HA-tagged HNF1B in DU145, which is 21.8% (**Fig. B**) and 23.7% (**Fig. C**) under DHT and ETH conditions in VCaP cells, respectively. For the overlap between cistromes of HA-tagged HNF1B in DU145 and ERG in VCaP cells, we found 47.95% (**Fig. D, Supplementary Fig. 8f** line 354) of HA-tagged HNF1B in DU145 overlapping with the ERG ChIP-seq data that was presented in **Fig. 5e** (lines 347, 356). Comparably, we identified 35.1% (**Fig. E**) and 34.1% (**Fig. F**) common overlapping cistrome of HA-tagged HNF1B in DU145 with recent performed ERG ChIP-seq in VCaP DHT and ETH conditions, respectively.

Reviewer #3 (Remarks to the Author):

As the authors note, prostate cancer has amongst the highest estimates of heritability of any malignancy. GWAS studies have identified 269 risk variants consistently associated with risk in multiethnic populations. The current investigation focuses on understanding the mechanistic aspects of a subset of these loci, specifically HNF1B. The study uses both experimental and computational approaches to investigate. The methods for the computational studies are clear and there is level of innovation in the study's focus.

While understanding the functional significance of these risk SNPs is a key next step, the overall rationale for the focus on TMPRSS2:ERG's connection of this study is not clearly laid out. The authors select HNF1B given its role more broadly across cancer types, which is reasonable. However, the selection to focus on TMPRSS2:ERG is not clearly motivated, particularly since it is a somatic event specific to prostate cancer. The framing of the introduction is out of balance with the description of the findings in the TCGA cohorts.

Response: Thank you for raising this important point. Our study focus on the somatic TMPRSS2-ERG fusion mainly due to an unbiased genome-wide co-expression analysis of HNF1B gene in the clinical PCa data sets and ChIP-seq data mining. We thus found that ERG is the most co-expressed gene with HNF1B where HNF1B SNP regions have several ERG binding sites observed by ChIP-seq data. For details, please refer to our description in the manuscript text (**lines 96-98; 218-223**). We also fully agree with you that TMPRSS2-ERG fusion is a somatic event specific to prostate cancer, and have expanded the discussion by saying "Given that TMPRSS2-ERG fusion is a somatic event specific to PCa31-36, additional transcription regulators remain to be uncovered to explore underlying mechanisms of the 17q12/HNF1B locus associated with other types of cancers²²⁻²⁶." (**lines 96-98; 529-532**)

Given the focus on TMPRSS2:ERG, it is then unclear the inclusion of many of the cell lines for investigation, since most of these do not contain the gene fusion event. A study by Mertz et al, 2007 describes this in detail. The VCaP cell line is reasonable given it contains TMPRSS2:ERG, but also molecularly is characterized by AR V7, high AR, etc. This should be taken into account in interpreting these cell line studies.

Response: Thank you for the kind comments. We agree that TMPRSS2:ERG gene fusion event across existing PCa cell lines has been well documented in the study of Mertz et al, 2007. In addition to VCaP cells harboring the gene fusion, Mertz et al also reported the other neuroendocrine PCa cell model NCI-H660 harboring the fusion. To see if this cell line is a good plus for current study, we performed RT-qPCR measurement of ERG and HNF1B expression levels. We found that the expression levels of ERG is over 15000-fold lower in NCI-H660 than that of VCaP cells (**Figure as below, left panel**), which is consistent with the report by Beltran et al in 2011, showing no detectable protein levels of ERG in NCI-H660 in comparison with VCaP⁷. This is most likely due to lack of AR expression and signaling in NCI-H660. We also observed HNF1B expression with near 15-fold lower in NCI-H660

compared to that in VCaP (**Figure as below, right panel**). We thus concluded that NCI-H660 is not suitable for our study of ERG-HNF1B interplay.

For the reasons to include many of the cell lines for investigation, briefly, we applied LNCaP cell line as negative control in ChIP-qPCR experiments as similar to VCaP, LNCaP is also an androgen-responsive cell line but with *TMPRSS2:ERG* fusion. We performed siRNA-mediated ERG knockdown VCaP and LNCaP cells, respectively, confirming the regulatory association of only ERG fusion with HNF1B. Moreover, we used VCaP, LNCaP, and also androgen-unresponsive cell lines PC3 and DU145 to investigate the role of HNF1B in PCa cell growth. 22RV1 cell line was applied to select single clones for CRISPR-Cas9-mediated deletion of HNF1B SNP regions as this is the only cell line that can survive from a single selected clone (**please see also our response to Reviewer #1, point #2**). We applied 22RV1 and V16A cells to do lentivirus-mediated shRNA knockdown assays, as VCaP is vulnerable to lentivirus transduction.

The authors motivate the study in part by "clinical significance" of this locus, yet the data in Supp 1 are weak and based on unclear, small datasets likely cherry-picked for this findings.

Response: We thank the reviewer for pointing out this issue. We have now added the source/original publication of datasets used in **Supplementary Fig. 1**. We examined the clinical relevance and HNF1B expression level in a variety of independent PCa datasets. We additionally supplied two more datasets comparing the HNF1B expressing level between PCa tumor and normal tissues despite of marginal significance (**Supplementary Fig. 1k**, l line 172).

We compared HNF1B expression with PSA, tumor stage in multiple datasets and summarized the results **as below in Table 1**. Although the association between HNF1B expression and PSA/tumor stage was not strong in most of datasets, HNF1B expression showed significant association with PSA in the MSKCC (Taylor) PCa dataset (**Supplementary Fig. 1n** line 175) and tumor stage in the Vanaja dataset (**Supplementary Fig. 1o** line 175). **As shown in Table 2 below**, we tested the associations of HNF1B

expression with patient survival in multiple PCa data sets. We did not observe statistically significant associations between HNF1B expression and survival in datasets with bigger samples sizes, such as the TCGA. However, in addition to the results showing in **Supplementary Fig. 1p** line 178 (**Ebiomedicine 2015**), we found that patients with higher HNF1B expression exhibited a clear trend of associations with worse survival in two more datasets **Glinsky (J Clin Invest 2004, Supplementary Fig. 1q** line 178) and **Lapointe (PNAS 2004, Supplementary Fig. 1r** line 178). These analyses together demonstrating HNF1B expression and clinical relevance shown in **Supplementary Fig. 1** was based on a broad deep investigation across a big collection of datasets.

Table 1. HNF1B expression vs PSA & Tumor stage

HNF1B expression vs PSA & Tumor stage			
	PSA	Tumor stage	
Bittner (GSE2109)	0.184	0.75	
Broad/Cornell (Nat Genet 2012)	0.399	0.536	
Camcap (Ebiomedicine 2015)	0.642	0.955	
CPGEA (Nature 2020)	0.223	0.657	
FHCRC (Nat Med 2016)	0.762	NA	
Glinsky (J Clin Invest 2004)	0.878	0.355	
Lapointe (PNAS 2004)	NA	0.874	
MSKCC (Cancer Cell 2010)	9.53E-03	0.972	Fig.S1n
SMMU (Eur Urol 2017)	0.345	0.607	
SU2C-PCF (PNAS 2019)	0.727	NA	
TCGA	0.706	0.0542	
Wallace (Cancer Res 2008)	NA	0.33	
Vanaja (Cancer Res 2003)	NA	0.0171	Fig.S1o
Welsh (Cancer Res 2001)	NA	0.518	
Yu (J Clin Oncol 2004)	NA	0.957	

Table 2. HNF1B expression vs patient survival

HNF1B expression vs patient survival					
	Overall survival	Biochemical recurrence	Disease-specific survival	Metastasis	
TCGA	$P= 0.62$; HR= 0.72	$P= 0.42$; HR= 0.84	$P= 0.46$; HR= 0.49	$P= 0.38$; HR= 0.83	
MSKCC (Cancer Cell 2010)	NA	$P= 0.92$; HR= 1.04	NA	NA	
SU2C-PCF (PNAS2019)	$P= 0.74$; HR= 0.906	NA	NA	NA	
CPGEA (Nature 2020)	$P= 0.164$; HR=0.616	NA	NA	NA	
Camcap (Ebiomedicine 2015)	NA	$P= 0.077$; HR= 2.23	NA	NA	Fig.S1p

Glinsky (J Clin Invest 2004)	NA	$P= 0.36$; HR= 1.35	NA	NA	Fig.S1q
Lapointe (PNAS 2004)	NA	$P= 0.36$; HR= 2.78	NA	NA	Fig.S1r
Seitlur (J Natl Cancer Inst 2008)	$P= 0.662$; HR= 1.06	NA	NA	NA	
Yu (J Clin Oncol 2004)	$P= 0.644$; HR= 1.75	NA	NA	NA	

Similarly, the clinical significance of the HNF1B signature -- is there associations with survival simply because it is correlated with Gleason score?

Response: We thank the reviewer for this comment. Gleason Score has long been used to describe prostate cancer severity in clinic, and higher Gleason score indicates the more aggressive disease. We found positive correlations between the HNF1B signature and Gleason score in multiple independent PCa cohorts, indicating the HNF1B signature may possess predictive value in other clinical features of prostate cancer. For example, in SU2C PNAS2019 cohort shown in **Fig. 4n** (line 310), the HNF1B signature displayed significant prognostic value with patient overall survival. We further tested the correlation between HNF1B cell cycle signature and Gleason score in this cohort, but the correlation between HNF1B cell cycle signature and Gleason score was not statistically significant (**Fig. A as below**). Moreover, we found that the Gleason score could not stratify patients with overall survival risk in this cohort (**Fig. B**). To further examine the effect of multiple variates on patient survival, we performed a multivariate analysis. While Gleason score reached a marginal significant P value, the result indicated HNF1B cell cycle signature outperformed other variables and significantly associated with patient overall survival (**Fig. C**). To exclude Gleason score as a confounding factor, we additionally stratified patients according to Gleason score classification and performed survival analysis with HNF1B cell cycle signature in each patient group with Gleason score ≤ 6 (**Fig. D**), 7 (**Fig. E**) or ≥ 8 (**Fig. F**). The results demonstrated a trend showing patients with higher HNF1B signature score underwent poorer overall survival in Gleason score group 7 and ≥ 8 . The trend in Gleason score ≤ 6 was not obvious, which might be attributable to the small sample size.

To further validate these findings, we replicated similar analysis in another cohort namely CPGEA comprising of 208 pairs of PCa tumor and adjacent normal tissues published in Nature 2020⁸. The results indicated significant correlation between HNF1B cell cycle signature and Gleason score, although the HNF1B cell cycle signature score was not the lowest in the Gleason score group ≤ 6 (**Fig. G**). The multivariate analysis indicated HNF1B cell cycle signature significantly associating with biochemical recurrence compared to other variables, indicating its potential in predicting patient survival independently (**Fig. H**). The survival analysis testing the prognosis of HNF1B signature by pre-stratifying patients by Gleason score of ≤ 6 (**Fig. I**), 7 (**Fig. J**) or ≥ 8 (**Fig. K**) indicated similar results as that of observed in the SU2C-PCF PNAS 2019 cohort. Taken together, these integrated results revealed that the association between HNF1B cell cycle signature and patient survival is not solely due to the correlation between HNF1B cell cycle signature and Gleason score.

The outcomes in TCGA are a mixed bag and not validated.

Response: We thank the reviewer for suggesting the validation of the outcomes observed in TCGA. We now have validated the findings from the TCGA dataset that showed positive significant association between HNF1B signature and clinical variables, such as tumor stage, Gleason score, and lymph node in multiple independent PCa cohorts. We supplied new data shown in **Fig. 4j** line 305 (**Grasso, Nature 2012**) to support the finding that HNF1B cell cycle signature is upregulated upon prostate cancer progression to advanced stage. Moreover, we provided additional data to support the association between HNF1B cell cycle signature and Gleason score in **Supplementary Fig. 7d** line 308 (**Bittner GSE2109**), biochemical recurrence in **Supplementary Fig. 7f** line 308 (**MSKCC, Cancer Cell 2010**), lymph node in **Supplementary Fig. 7g** line 308 (**Yu, J Clin Oncol 2014**), and tumor stage in **Supplementary Fig. 7j** line 308 (**Bittner GSE2109**). Hence, in addition to the outcomes in TCGA, we provided additional supports from different datasets to solidify the observation on the positive correlation of HNF1B cell cycle signature with PCa severity.

Moreover, since TMPRSS2:ERG on its own is not prognostic, how does this translate to the apparent co-operation between ERG and HNF1B?

Response: We appreciate the reviewer for this valuable question. We agree that the association between ERG fusion and PCa prognosis is not conclusive based on the current evidence. A systematic review summarized the published studies and performed meta-analysis regarding this topic⁹. Studies with large sample sizes showed consistent association between ERG fusion and higher T stages (T3-4), as well as metastatic PCa (M1). Meta-analysis also confirmed such relationships. However, ERG fusion seems to have very weak and insignificant association with PCa recurrence and death. This could be due to the limited number of reported study investigating this topic comparing with those evaluating T stage and M stage.

Despite that it is not conclusive for TMPRSS2:ERG PCa prognosis, in current study we hypothesized that the co-operation between ERG and HNF1B could promote PCa progression, where the impact of the cooperation might be functioned via other mechanisms. For instance, in **Fig. 3l-n** (lines 263, 266, 267), the eQTL association between the 17q12 locus SNPs and HNF1B expression was significant only in ERG fusion-positive PCa tumors (**Fig. 3n** line 267), while neither in ERG fusion-negative tumors (**Fig. 3m** line 266) nor in PCa patients without ERG fusion stratification (**Fig. 3l** line 263). Moreover, in **Fig. 5f** (lines 360, 361), we developed a HNF1B&ERG target gene signature, which was derived from genes obtained from common binding sites of HNF1B and ERG ChIP-seq profiles and further integrated with HNF1B upregulated genes based on RNA-seq analysis. In **Fig. 5j, k** (line 371), the HNF1B&ERG target gene signature demonstrated prognostic value in stratifying PCa patients with overall survival outcomes. Additionally, the eGene signature in **Fig. 6c, d** (line 393), derived from target genes of eQTL SNPs that were enriched in HNF1B and ERG common binding sites, showed significant elevated risks with biochemical relapse and metastasis in patient group with an intermediate risk (Gleason score 7). Strikingly, the prognostic value of the HNF1B&ERG eGene signature was not found in patients with lower risk (Gleason score ≤ 6 , **Supplementary Fig. 9a, c** line 394) or higher risk (Gleason score ≥ 8 , **Supplementary Fig. 9b, d** line 394). Collectively, the co-operation between ERG and

HNF1B together with the involvement of additional factors such as eQTL risk SNPs and HNF1B target genes demonstrated prognostic value in PCa.

One key piece that is missing from this study is whether SNPs in HNF1B are enriched/exclusive to TMPRSS2:ERG positive tumors in humans. This would help establish the relevance of the experimental findings to humans.

Response: We appreciate for this valuable suggestion that might help establish the connection between the co-occurrence of SNPs in TMPRSS2:ERG (T2E) fusion-positive tumors and disease aggressiveness. Please kindly see our above response to the **reviewer #2 point #3**.

Below are some specific comments:

Introduction:

line 51: the 57% refers to estimate of heritability and not familial (familiar) risk.

Response: Thank you for comment. We made the changes based on your suggestion (line 77).

line 79-81: the majority of studies do not find TMPRSS2:ERG to be prognostic or associated with high grade cancers.

Response: Thank you for your comments. We have now re-organized the wording (lines 105-108).

Results:

line 118-120: Why is this finding remarkable? This has been demonstrated previously in GWAS studies.

Response: We fully agree with your statement, and made the changes to balance the description (lines 145-150).

line 129-131: While the comparison of HNF1B shows higher expression in prostate cancer, it is not strikingly different and appears to be in the top 25% of cancer types (against an idea of specificity).

Response: Thank you for your suggestion. We edited the sentence in accordance with your suggestion (lines 155-157).

lines 148-151: Would suggest greater caution in interpreting survival analyses given the very small sample size, lack of significance, and lack of clarity on outcomes in this particular clinical dataset (there are better datasets in cBIOPORTAL to investigate survival. Why this one?)

Response: We thank the reviewer for suggesting datasets from the cBioportal to examine the prognostic value of HNF1B. We have now investigated the association between HNF1B expression and patient survival using multiple datasets from cBioPortal. However, none of the cBioPortal datasets showed significant association between HNF1B expression and patient survival (**shown as Fig. A-F below**). In addition to the existing result shown in the Camcap¹⁰ cohort that was published in the EBioMedicine 2015 (**Supplementary Fig. 1p** line 178), we observed additional two cohorts demonstrating that patients with high HNF1B expression levels displayed a trend with poorer prognosis despite that the P values were not statistically significant (**Supplementary Fig. 1q, r** line 178). We used the median of HNF1B expression value as a cutoff in stratifying patients and then tested the association with survival. Perhaps a more specific cutoff might help the identification of associations of HNF1B expression with patient prognosis.

References:

1. Mei, S. *et al.* Cistrome Data Browser: A data portal for ChIP-Seq and chromatin accessibility data in human and mouse. *Nucleic Acids Res.* (2017) doi:10.1093/nar/gkw983.
2. Zheng, R. *et al.* Cistrome Data Browser: Expanded datasets and new tools for gene regulatory analysis. *Nucleic Acids Res.* (2019) doi:10.1093/nar/gky1094.
3. Deplancke, B., Alpern, D. & Gardeux, V. The Genetics of Transcription Factor DNA Binding Variation. *Cell* (2016) doi:10.1016/j.cell.2016.07.012.
4. Bach, I. & Yaniv, M. More potent transcriptional activators or a transdominant

inhibitor of the HNF1 homeoprotein family are generated by alternative RNA processing. *EMBO J.* (1993) doi:10.1002/j.1460-2075.1993.tb06107.x.

5. Huang, D. *et al.* Genetic polymorphisms at 19q13.33 are associated with [-2]proPSA (p2PSA) levels and provide additional predictive value to prostate health index for prostate cancer. *Prostate* (2021) doi:10.1002/pros.24192.
6. Das, S. *et al.* Next-generation genotype imputation service and methods. *Nat. Genet.* (2016) doi:10.1038/ng.3656.
7. Beltran, H. *et al.* Molecular characterization of neuroendocrine prostate cancer and identification of new drug targets. *Cancer Discov.* (2011) doi:10.1158/2159-8290.CD-11-0130.
8. Li, J. *et al.* A genomic and epigenomic atlas of prostate cancer in Asian populations. *Nature* (2020) doi:10.1038/s41586-020-2135-x.
9. Song, C. & Chen, H. Predictive significance of TMRPSS2-ERG fusion in prostate cancer: A meta-analysis. *Cancer Cell International* (2018) doi:10.1186/s12935-018-0672-2.
10. Ross-Adams, H. *et al.* Integration of copy number and transcriptomics provides risk stratification in prostate cancer: A discovery and validation cohort study. *EBioMedicine* (2015) doi:10.1016/j.ebiom.2015.07.017.

Reviewers' Comments:

Reviewer #1:

Remarks to the Author:

The authors have addressed my concerns.

Reviewer #2:

Remarks to the Author:

The authors have beautifully addressed my comments. It is a very good study and well done to all.

I only have a minor clarification to make:

Reviewer 3 asked an enrichment analysis of the risk SNPs in T2E+ vs - tumors, while my concern (point 3) was simply to provide the reader with an estimate percentages of T2E positive prostate cancers with the risk alleles. This information is now partially in panel A page 14 in your rebuttal letter, but the panel should have included all the SNPs (also e.g. the SNPs at 17p13.3), and marking the relevant progression-associated alleles. Once the modifications are made to that panel, the figure could be presented e.g. in Supplementary.

Reviewer #4:

Remarks to the Author:

Summary

The work investigates germline-somatic interactions in prostate cancer, between germline risk variants in the HNF1B locus and the somatic TMPRSS2:ERG fusion. This is a fascinating and largely unstudied topic and could have important implications both for risk modeling and in the clinic. The authors have conducted a number of new analyses to address concerns from Reviewer #3, which are extensive and have improved the manuscript. However, given the large number of datasets being integrated, the lack of directly stated multiple test correction makes it difficult to assess whether these additional analyses actually confirm the initial claims. Additionally, the new GWAS analysis in a Chinese cohort produced associations that were highly discordant with previous studies and were not explained. As such, while it's possible the new data could be conclusive, the lack of statistical rigor leaves this in doubt.

Major comments

1. To establish clinical relevance, the authors looked at association between HNF1B expression and survival across 9 cohorts (response p.21). Do any of these associations pass multiple test correction: it does not appear so, the most significant association reported is p-value=0.077 which is not significant? Alternatively, is a single meta-analyzed association statistically significant? Likewise for the association between HNF1B expression and PSA/stage. If these associations are not significant, it is not accurate to report HNF1B expression as being clinically relevant ("The results suggested that patients with higher expression levels of HNF1B bear a trend for increased risk of biochemical recurrence. Together, these findings illustrated a potential role for HNF1B in PCa severity and progression, indicating that HNF1B is a plausible causative gene underlying the effects of the 17q12 PCa susceptibility locus variants") and I recommend avoiding terms like "trend of association" for associations that are not statistically significant, since there is no statistical evidence of a trend.

2. The HNF1B signature association appears robust, and the use of a multivariate model incorporating stage/gleason score is appropriate. However it's still not clear whether the association is merely significant in a single cohort or meta-analyzed across all cohorts investigated. In particular, the

association was identified in the SU2C cohort with overall survival but replicated in the CPGEA cohort with biochemical recurrence; why were different outcomes used and are the results concordant when the same outcomes are used. Lastly, what is the justification for testing the cell signature "high" group (which is dependent on a user-defined threshold) rather than the continuous cell signature value? Is the continuous signature score also associated or else why not?

3. To address whether the risk SNPs near HNF1B are associated with TMPRSS2-ERG fusion the study conducted additional GWAS in a Chinese cohort. For prostate cancer risk (i.e. cancer cases versus controls) Supplementary Table 3 shows an incredibly significant association with OR=10 at rs4239217, which is completely out of line with the previously reported OR=1.2 for this locus from much larger GWAS. Either this cohort must be ascertained in a very unusual way, or something is wrong with the GWAS analysis that yields these impossibly large associations.

4. Turning to the association with T2E: in Supplementary Table 4, two SNPs were nominally ($p < 0.05$) associated with T2E but these do not pass multiple test correction for the number of SNPs tested. Either permutation testing must additionally be performed to show that this association is significant after accounting for all SNPs tested, or it should not be reported as significant. Given lack of association between these SNPs in TCGA, it is most accurate to report that there is no current evidence for a direct germline-somatic association.

In sum, it appears that in the new data there is no significant association between HNF1B expression and survival; the HNF1B cell cycle signature is nominally associated in two cohorts but with different clinical outcomes; and the risk SNPs are not significant after multiple test correction. Perhaps the manuscript would benefit from reframing as an interesting biological relationship between HNF1B and ERG rather than a clinically relevant germline-somatic interaction?

NCOMMS-21-23625A: Extensive germline-somatic interplay contributes to prostate cancer progression through HNF1B co-option of TMPRSS2-ERG

REVIEWER COMMENTS

Reviewer #1 (Remarks to the Author):

The authors have addressed my concerns.

Reviewer #2 (Remarks to the Author):

The authors have beautifully addressed my comments. It is a very good study and well done to all.

I only have a minor clarification to make:

Reviewer 3 asked an enrichment analysis of the risk SNPs in T2E+ vs - tumors, while my concern (point 3) was simply to provide the reader with an estimate percentages of T2E positive prostate cancers with the risk alleles. This information is now partially in panel A page 14 in your rebuttal letter, but the panel should have included all the SNPs (also e.g. the SNPs at 17p13.3), and marking the relevant progression-associated alleles. Once the modifications are made to that panel, the figure could be presented e.g. in Supplementary.

Response: We appreciate the reviewer's advice for making the result visible to the readers. We have now included the 17p13.3 locus SNPs and also marked the risk alleles to the panel as below. The 17q13 HNF1B locus SNPs and the 17q13.3 locus variants are now presented in the **Supplementary Fig. 5o** and **Supplementary Fig. 10c**, respectively.

Reviewer #4 (Remarks to the Author):

Summary

The work investigates germline-somatic interactions in prostate cancer, between germline risk variants in the HNF1B locus and the somatic TMPRSS2:ERG fusion. This is a fascinating and largely unstudied topic and could have important implications both for risk modeling and in the clinic. The authors have conducted a number of new analyses to address concerns from Reviewer #3, which are extensive and have improved the manuscript. However, given the large number of datasets being integrated, the lack of directly stated multiple test correction makes it difficult to assess whether these additional analyses actually confirm the initial claims. Additionally, the new GWAS analysis in a Chinese cohort produced associations that were highly discordant with previous studies and were not explained. As such, while it's possible the new data could be conclusive, the lack of statistical rigor leaves this in doubt.

Major comments

1. To establish clinical relevance, the authors looked at association between HNF1B expression and survival across 9 cohorts (response p.21). Do any of these associations pass multiple test correction: it does not appear so, the most significant association reported is $p\text{-value}=0.077$ which is not significant? Alternatively, is a single meta-analyzed association statistically significant? Likewise for the association between HNF1B expression and PSA/stage. If these associations are not significant, it is not accurate to report HNF1B expression as being clinically relevant ("The results suggested that patients with higher expression levels of HNF1B bear a trend for increased risk of biochemical recurrence. Together, these findings illustrated a potential role for HNF1B in PCa severity and progression, indicating that HNF1B is a plausible causative gene underlying the effects of the 17q12 PCa susceptibility locus variants") and I recommend avoiding terms like "trend of association" for associations that are not statistically significant, since there is no statistical evidence of a trend.

Response: We thank the reviewer for pointing out the issue and giving the advice for additional analysis. Those P values from the PSA/TS and survival analyses presented in **Table 1** and **Table 2** on page 22 in the previous rebuttal letter were not adjusted by multiple testing corrections. Therefore, the P value 0.077 obtained from the survival analysis in the Camcap cohort (**Ebiomedicine 2015**) was a raw P value. We now applied the multiple testing correction approach on these raw P values. However, the obtained adjusted P values were not significant as the original raw P values were not significant on their own. We alternatively performed a meta-analysis by incorporating the nine studies with survival outcomes to comprehensively overview the association of the HNF1B expression with patient overall survival and biochemical recurrence across different studies. However, the pooled HR from the meta-analysis displayed that the prognostic effect of HNF1B expression was neither statistically significant for overall survival (**Figure A** below, $P=0.79$, $\ln(\text{HR}): 0.03$; $\ln(95\% \text{ CI}): -0.20\text{--}0.26$) nor for biochemical recurrence (**Figure B** below, $P=0.96$, $\ln(\text{HR}): -0.01$; $\ln(95\% \text{ CI}): -0.27\text{--}2.62$) in PCa patients. We also applied meta-analysis to examine the association between the HNF1B expression level and PSA/tumor stage. As the

measurement basis of tumor stages/PSA in those data sets presented in the **Table 1** in the previous rebuttal letter are different. For example, some data sets contain tumor stage of T1, T2 and T3, while other include T2, T3 and T4, We therefore compared the association of HNF1B expression with tumor stage/PSA pair-wisely. We first applied limma package (**Ritchie et al., 2015**) to compute for the *P* value and fold change between two conditions. We next fed these inputs to MetaVolcanoR package (**Prada et al., 2022**) for conducting the meta-analysis to systematically estimate the correlation between HNF1B expression level and TS/PSA in multiple PCa data sets. However, the meta-analysis results indicated non-significant correlation of HNF1B expression with tumor stage (**Table 1** below) or PSA (**Table 2** below) in those PCa data sets investigated. As these associations were non-significant, we thereby excluded these statistically weak observations (**Supplementary Fig. 1m-r**) from the manuscript and modified the text accordingly. The limitations are probably due to these currently available cohorts with small sample size and limited amount of patients, thus lacking statistical power and leading to the observations of non-significant associations between HNF1B expression and patient prognosis or TS/PSA. We expect to observe potentially significant associations in the future when studies with large sample size become available.

Table 1. HNF1B expression correlation between tumor stages and the Meta-analysis.

Comparison type	P.Value	logFC	CI.L	CI.R	Dataset
T2 vs. T1	0.96517	-0.0016	-0.0763	0.07297	Camcap
T2 vs. T1	0.10897	-0.5076	-1.1298	0.11465	CPGEA
T2 vs. T1	0.65967	0.02839	-0.0995	0.15627	Glinsky
T2 vs. T1	0.81024	0.07196	-0.5194	0.6633	MSKCC
T2 vs. T1	0.74067	-0.0367	-0.2542	0.18093	TCGA
T3 vs. T1	0.99154	0.00052	-0.0968	0.09784	Camcap
T3 vs. T1	0.56306	-0.2286	-1.0086	0.55144	CPGEA
T3 vs. T1	0.11557	-0.3241	-0.7295	0.08139	Glinsky
T3 vs. T1	0.95472	0.04176	-1.4096	1.49316	MSKCC
T3 vs. T1	0.02137	-0.3751	-0.6942	-0.0559	TCGA
T3 vs. T2	0.4805	-0.1866	-0.7132	0.34001	Bittner
T3 vs. T2	0.57982	0.31063	-0.8468	1.46805	Broad-Cornell

T3 vs. T2	0.96773	0.00217	-0.1039	0.10823	Camcap
T3 vs. T2	0.4404	0.27901	-0.4343	0.99236	CPGEA
T3 vs. T2	0.08565	-0.3525	-0.7555	0.05064	Glinsky
T3 vs. T2	0.98179	0.00477	-0.412	0.42158	Lapointe
T3 vs. T2	0.96768	-0.0302	-1.5007	1.44028	MSKCC
T3 vs. T2	0.56823	0.16329	-0.4058	0.73237	SMMU
T3 vs. T2	0.03811	-0.3384	-0.6582	-0.0186	TCGA
T3 vs. T2	0.06463	-0.1636	-0.3378	0.01067	Vanaja
T3 vs. T2	0.95533	-0.0125	-0.4558	0.43085	Wallace
T3 vs. T2	0.89686	0.17295	-2.5908	2.93672	Welsh
T3 vs. T2	0.85709	-0.0097	-0.1172	0.09777	Yu
T4 vs. T2	0.89003	0.0664	-0.8919	1.0247	Bittner
T4 vs. T2	0.29255	0.52865	-0.467	1.52429	SMMU
T4 vs. T2	0.3317	-0.9007	-2.7396	0.93826	Wallace
T4 vs. T2	0.95554	-0.0085	-0.3136	0.29656	Yu
T4 vs. T3	0.61532	0.25301	-0.7506	1.25665	Bittner
T4 vs. T3	0.48566	0.36536	-0.6761	1.40681	SMMU
T4 vs. T3	0.34003	-0.8882	-2.7334	0.95702	Wallace
T4 vs. T3	0.99378	0.00118	-0.3002	0.30259	Yu
Meta-analysis type	Metap	Metafc	idx		
T2 vs. T1	0.78439	-0.0891	-0.0094		
T3 vs. T1	0.20919	-0.1771	-0.1203		
T3 vs. T2	0.6307	-0.0124	-0.0025		
T4 vs. T2	0.75872	-0.0785	-0.0094		
T4 vs. T3	0.80081	-0.0672	-0.0065		

Table 2. HNF1B expression correlation between PSA and the Meta-analysis.

Comparison type	P.Value	logFC	CI.L	CI.R	dataset
PSA 10-20 vs. 4-10	0.8019	0.1958	-1.4381	1.8296	Broad-Cornell
PSA 10-20 vs. 4-10	0.8692	0.0063	-0.0697	0.0824	Camcap
PSA 10-20 vs. 4-10	0.1025	-0.6831	-1.505	0.1388	CPGEA
PSA 10-20 vs. 4-10	0.928	-0.0072	-0.1655	0.1511	Glinsky
PSA 10-20 vs. 4-10	0.5962	-0.1872	-0.89	0.5156	SMMU
PSA 10-20 vs. 4-10	0.457	-0.1874	-0.6839	0.3092	SU2C-PCF
PSA 10-20 vs. 4-10	0.0392	-0.234	-0.4562	-0.0118	Taylor
PSA 10-20 vs. 4-10	0.8634	-0.1114	-1.4329	1.2102	TCGA
PSA 20+ vs. 10-20	0.2272	-1.7517	-4.7171	1.2136	Broad-Cornell
PSA 20+ vs. 10-20	0.46	-0.1312	-0.4817	0.2194	Camcap
PSA 20+ vs. 10-20	0.679	0.1354	-0.5106	0.7814	CPGEA
PSA 20+ vs. 10-20	0.7308	-0.0372	-0.2523	0.1778	Glinsky
PSA 20+ vs. 10-20	0.3904	-0.2616	-0.8665	0.3433	SMMU
PSA 20+ vs. 10-20	0.1802	0.2914	-0.1363	0.719	SU2C-PCF
PSA 20+ vs. 10-20	0.0306	0.328	0.0312	0.6249	Taylor

PSA 20+ vs. 10-20	0.4237	-0.6592	-2.3308	1.0125	TCGA
PSA 20+ vs. 4-10	0.2313	-1.556	-4.2142	1.1023	Broad-Cornell
PSA 20+ vs. 4-10	0.4764	-0.1248	-0.4711	0.2214	Camcap
PSA 20+ vs. 4-10	0.1575	-0.5477	-1.31	0.2145	CPGEA
PSA 20+ vs. 4-10	0.4078	0.4002	-0.5528	1.3532	FHCRC
PSA 20+ vs. 4-10	0.6429	-0.0444	-0.2348	0.1459	Glinsky
PSA 20+ vs. 4-10	0.185	-0.4488	-1.1183	0.2207	SMMU
PSA 20+ vs. 4-10	0.5977	0.104	-0.2847	0.4926	SU2C-PCF
PSA 20+ vs. 4-10	0.4281	0.0941	-0.1402	0.3283	Taylor
PSA 20+ vs. 4-10	0.3509	-0.7706	-2.4422	0.9011	TCGA
Meta-analysis type	Metap	Metafc	idx		
PSA 10-20 vs. 4-10	0.5395	-0.151	-0.0405		
PSA 20+ vs. 10-20	0.224	-0.2608	-0.1694		
PSA 20+ vs. 4-10	0.3931	-0.3216	-0.1304		

2. *The HNF1B signature association appears robust, and the use of a multivariate model incorporating stage/gleason score is appropriate. However it's still not clear whether the association is merely significant in a single cohort or meta-analyzed across all cohorts investigated. In particular, the association was identified in the SU2C cohort with overall survival but replicated in the CPGEA cohort with biochemical recurrence; why were different outcomes used and are the results concordant when the same outcomes are used. Lastly, what is the justification for testing the cell signature "high" group (which is dependent on a user-defined threshold) rather than the continuous cell signature value? Is the continuous signature score also associated or else why not?*

Response: We appreciate the reviewer for pointing out this issue. We indeed performed the multivariate analysis testing the association of HNF1B cell cycle signature score and other clinical variables with the patients' overall survival in the SU2C-PCF cohort, while the biochemical recurrence in the CPGEA cohort. The reason why different outcomes used was due to the different defined endpoints in these two cohorts. Specifically, the overall survival data was the only outcome available in the SU2C cohort but not the biochemical recurrence. Conversely, the only available measured outcome in the CPGEA cohort was the patients' biochemical recurrence but not the overall survival. Therefore, we used different outcomes (overall survival or biochemical recurrence) for examining the association of HNF1B cell cycle signature, clinical features and patient prognosis in the SU2C-PCF and CPGEA cohorts, respectively. We performed the multivariate analysis to solidify the association between HNF1B cell cycle signature and patient prognosis, and validated that the association was not merely due to the significant correlation between HNF1B signature and the Gleason score (as the reviewer#3 concerned). As the reviewer now suggested, we additionally performed a meta-analysis to systematically review, integrate, and provide an overall interpretation of the association between HNF1B cell cycle signature and patient prognostic outcomes across different studies. We estimated the hazard ratio (HR) from each study and then pooled these HRs (Tierney et al., 2007) in a meta-analysis with the same outcome, including overall survival (OS) and biochemical recurrence (BCR). As heterogeneity among the studies was not significant ($P= 0.454$, $I^2 = 0\%$ for OS; $P= 0.931$,

$I^2 = 0\%$ for BCR), a fixed-effects model was applied. The pooled HRs from the meta-analysis showed that the elevated HNF1B cell cycle signature was significantly associated with poor OS (**Figure A** below, $P = 3.5e-03$, $\ln(\text{HR})$: 0.85; $\ln(95\% \text{ CI})$: 0.28–1.41) and higher risk for BCR (**Figure B** below, $P = 6.1e-03$, $\ln(\text{HR})$: 0.50; $\ln(95\% \text{ CI})$: 0.14–0.85) in PCa patients. The meta-analysis further confirmed the robustness of the HNF1B cell cycle signature with poor prognosis in PCa patients. We have now included these results from the meta-analysis to **Supplementary Fig. 7m** and **7n**.

As for testing the impact of the HNF1B cell cycle signature on patient prognosis, we treated the HNF1B signature as a categorical variable by stratifying patients into two groups with high and low HNF1B signature scores. Prostate cancer patients were classified into two groups based on the median expression value of HNF1B cell cycle signature, which is a standard cutoff/approach for examining patient survival outcomes used by many high-quality research work published in top journals (Hua et al., 2018; Zhu et al., 2010) (**Zhu et al. J Clin Oncol. 2010, PMID: 20823422; Hua et al. Cell. 2018, PMID: 30033362**). We chose patients expressing low HNF1B score as the reference level with defined hazard ratio 1.0, and we found patients displaying high HNF1B signature score with HR above one, which indicates a worse prognosis. We now also tested the association using the continuous value of the HNF1B cell cycle signature. We replicated similar analysis showing on page 24 **Figure C** and **F** from the previous rebuttal letter using the continuous value of the HNF1B cell cycle signature score as recommended. The results (**Figure C** and **D** below) were relatively concordant to that of categorical value of the HNF1B cell cycle signature score. We found the continuous value of the HNF1B cell cycle signature score as one of the variables significantly influencing PCa patient prognosis. We have now supplied the results from the multivariate analysis using both the categorical and continuous value of the HNF1B cell cycle signature to **Supplementary Fig. 7o** and **7p**.

(i.e. cancer cases versus controls) Supplementary Table 3 shows an incredibly significant association with OR=10 at rs4239217, which is completely out of line with the previously reported OR=1.2 for this locus from much larger GWAS. Either this cohort must be ascertained in a very unusual way, or something is wrong with the GWAS analysis that yields these impossibly large associations.

Response: Thank you for your concern. We checked our data among the patients who had ERG status information (580 PCa vs. 475 non-PCa). We found that these samples involved two sets of genotyping data using different reference populations for imputation. One set of data (n=768) was imputed based on Chinese Han in Beijing (CHB, n=89), the other (n=775) were imputed using Eastern Asian (EAS, n=504). This causes the missing or mismatching of the genotypes in some of the SNPs (for example, rs4239217, rs11651052, etc.). This phenomenon is usually rare which only happens in 5%~9% of the genome (**Howie et al., 2012**). To address this issue, we redo the imputation in the entire cohort based on the reference population of EAS. We have updated our results thereafter (please refer to the updated **supplementary table 3**). The allelic frequencies, risk alleles, as well as the OR directions are concordant to those in reported GWASs. To further confirm our results, we performed analysis with additional 472 samples genotyped recently with ERG data (1543 in total). Similar results were observed (**Table below**).

Previous Supplementary Table 3 with corrected calculation. Association between PCa status and SNPs (580 PCa vs 475 non-Pca*)

SNP	Alt allele	Ref allele	Risk allele frequency	OR (95%CI)	P
rs11649743	A	G	0.34	0.98 (0.81-1.19)	0.85
rs718960	T	C	0.30	1.07 (0.85-1.34)	0.57
rs11263761	G	A	0.27	0.81 (0.65-0.99)	0.04
rs4430796	G	A	0.24	0.86 (0.70-1.06)	0.15
rs4239217	G	A	0.18	0.59 (0.47-0.73)	3.09E-06
rs8064454	A	C	0.18	0.59 (0.47-0.74)	4.60E-06
rs7405696	G	C	0.26	0.65 (0.53-0.80)	4.20E-05
rs11651052	A	G	0.24	0.86 (0.67-1.10)	0.22
rs9901746	G	A	0.26	0.63(0.52-0.78)	1.40E-05
rs11263763	G	A	0.23	0.80 (0.62-1.04)	0.10
rs11658063	C	G	0.23	0.80 (0.62-1.04)	0.10
rs12453443	C	G	0.26	0.64 (0.52-0.80)	4.44E-05
rs3760511	T	G	0.26	0.74 (0.61-0.91)	3.37E-03

*Two samples were excluded due to the missing age

Association was tested using logistic regression by adjusting for age

New Supplementary Table 3. Association between PCa status and SNPs (791 PCa vs 752 non-PCa).

SNP	BP	Alt allele	Ref allele	Freq (alt)	Freq in gnomAD [#]	OR (95%CI)	P
rs11649743	36074979	A	G	0.32	0.34	0.78 (0.66-0.91)	1.89E-03
rs718960*	36077279	T	C	0.29	0.34	N/A	N/A
rs11263761	36097775	G	A	0.30	0.32	0.91 (0.77-1.07)	0.25
rs4430796	36098040	G	A	0.25	0.28	0.87 (0.74-1.03)	0.11
rs4239217	36098987	G	A	0.25	0.29	0.85 (0.72-1.01)	0.06
rs8064454	36101586	A	C	0.24	0.28	0.80 (0.68-0.95)	0.01
rs7405696	36102035	G	C	0.32	0.36	0.80 (0.68-0.95)	8.89E-03
rs11651052	36102381	A	G	0.24	0.28	0.82 (0.69-0.97)	0.02
rs9901746	36103149	G	A	0.32	0.38	0.81 (0.69-0.96)	0.01
rs11263763	36103565	G	A	0.24	0.26	0.80 (0.67-0.95)	0.01
rs11658063	36103872	C	G	0.23	0.26	0.81 (0.68-0.97)	0.02
rs12453443	36104121	C	G	0.32	0.38	0.82 (0.70-0.97)	0.02
rs3760511	36106313	T	G	0.31	0.36	0.86 (0.73-1.01)	0.06
Additional							
rs79882976	36083710	C	A	0.05	0.06	0.93 (0.65-1.33)	0.69
rs11651496	36084244	C	T	0.05	0.06	0.93 (0.65-1.33)	0.69
rs3744764	36090396	C	T	0.07	0.06	1.12 (0.84-1.50)	0.43

*this SNP failed to pass quality control (80%<genotype call <90%) in the new dataset.

[#]Frequency of alt allele in East Asian population by gnomAD dataset (<https://gnomad.broadinstitute.org/>).

Association results were adjusted for patient's age at diagnosis.

Abbreviation: N/A, not applicable; Freq, frequency; PCa, prostate cancer.

4. Turning to the association with T2E: in Supplementary Table 4, two SNPs were nominally ($p < 0.05$) associated with T2E but these do not pass multiple test correction for the number of SNPs tested. Either permutation testing must additionally be performed to show that this association is significant after accounting for all SNPs tested, or it should not be reported as

significant. Given lack of association between these SNPs in TCGA, it is most accurate to report that there is no current evidence for a direct germline-somatic association.

Response: Thank you for your comments. We updated the results based on the newly imputed data and additional data. Results from the new data show that three SNPs were associated with ERG expression status (**New Supplementary Table 4 below**). Among them, rs79882976 and rs11651496 have relatively strong LD with rs11649743 ($D'=0.87$) and rs718960 ($D'=0.72$). And rs3744764 is LD with multiple SNPs ($D'>0.8$) including rs7405696, rs11651052, rs9901746, rs11263763, and rs11658063.

Indeed, multiple testing issue is a major concern in the genetic association study. However, the important assumption of the most commonly used method of multiplicity correction, the Bonferroni correction, should base on the independence of each variable. In the current situation, the SNPs in HNF1B region ($n=63$ from our genotyping data) are not independent---Strong LD is observed among the SNPs. **As shown in the matrix plot below (under the tables), SNPs in HNF1B region could be grouped to 2-3 clusters based on the LD (D' or R^2 , indicated in red dash line).** This suggests that, despite the 63 SNPs we tested, only 2-3 independent tests were performed. In this case ($0.05/63$), Bonferroni correction is too stringent. Using $0.05/3$ or $0.05/2$ as the significant level here would be more appropriate. Here we observed 3 SNPs which has relatively strong LD with other SNPs across the HNF1B region reached significant P value.

Accordingly, we have updated the methods session on “**Enrichment analysis of HNF1B SNPs in ERG-positive PCa tumors**”, and the **Supplementary Table 2-4**.

**Previous Supplementary Table 4 with corrected calculation.
Association between ERG expression and SNPs in PCa sample (107 ERG+ PCa vs 473 ERG- PCa)**

SNP	Alt allele	Ref allele	Alt allele frequency	OR (95%CI)	P
rs11649743	A	G	0.34	0.91 (0.66-1.25)	0.54
rs718960	T	C	0.30	1.17 (0.82-1.65)	0.40
rs11263761	G	A	0.25	0.97 (0.68-1.39)	0.85
rs4430796	G	A	0.23	1.02 (0.71-1.45)	0.97
rs4239217	G	A	0.15	0.95 (0.63-1.43)	0.73
rs8064454	A	C	0.15	0.89 (0.58-1.36)	0.51
rs7405696	G	C	0.23	1.06 (0.74-1.52)	0.86
rs11651052	A	G	0.23	1.14 (0.71-1.83)	0.59
rs9901746	G	A	0.22	0.96 (0.66-1.39)	0.72
rs11263763	G	A	0.21	0.97 (0.57-1.67)	0.92
rs11658063	C	G	0.21	0.98 (0.57-1.67)	0.93
rs12453443	C	G	0.22	0.99 (0.68-1.44)	0.88

rs3760511	T	G	0.24	0.71 (0.49-1.04)	0.07
-----------	---	---	------	------------------	------

New Supplementary Table 4. Association between ERG expression and SNPs in PCa samples (136 ERG+ vs 655 ERG-).

SNP	BP	Alt allele	Ref allele	Freq (alt)	Freq in gnomAD [#]	OR (95%CI)	P
rs11649743	36074979	A	G	0.30	0.34	0.84 (0.63-1.12)	0.23
rs718960*	36077279	T	C	0.29	0.34	N/A	N/A
rs11263761	36097775	G	A	0.29	0.32	1.07 (0.80-1.43)	0.64
rs4430796	36098040	G	A	0.24	0.28	0.97 (0.72-1.31)	0.84
rs4239217	36098987	G	A	0.24	0.29	0.95 (0.71-1.29)	0.76
rs8064454	36101586	A	C	0.22	0.28	0.96 (0.70-1.30)	0.77
rs7405696	36102035	G	C	0.30	0.36	1.10 (0.82-1.47)	0.52
rs11651052	36102381	A	G	0.23	0.28	0.94 (0.69-1.28)	0.68
rs9901746	36103149	G	A	0.30	0.38	1.06 (0.78-1.42)	0.72
rs11263763	36103565	G	A	0.22	0.26	0.89 (0.64-1.23)	0.47
rs11658063	36103872	C	G	0.22	0.26	0.88 (0.64-1.22)	0.45
rs12453443	36104121	C	G	0.30	0.38	1.06 (0.78-1.42)	0.72
rs3760511	36106313	T	G	0.30	0.36	0.86 (0.73-1.01)	0.78
Additional							
rs79882976	36083710	C	A	0.05	0.06	2.00 (1.15-3.48)	0.01
rs11651496	36084244	C	T	0.05	0.06	2.00 (1.15-3.48)	0.01
rs3744764	36090396	C	T	0.07	0.06	1.67 (1.08-2.60)	0.02

*this SNP failed to pass quality control (80%<genotype call <90%) in the new dataset.

[#]Frequency of alt allele in East Asian population by gnomAD dataset (<https://gnomad.broadinstitute.org/>).

Abbreviation: N/A, not applicable; Freq, frequency; PCa, prostate cancer.

In sum, it appears that in the new data there is no significant association between HNF1B expression and survival; the HNF1B cell cycle signature is nominally associated in two cohorts but with different clinical outcomes; and the risk SNPs are not significant after multiple test correction. Perhaps the manuscript would benefit from reframing as an interesting biological relationship between HNF1B and ERG rather than a clinically relevant germline-somatic interaction?

Response: Thank you for the comments. We indeed did not observe significant associations between HNF1B expression level and patient survival nor with TS/PSA, which is might be e.g. due to limited sample size in current available data sets. We now have excluded these weak observations from the manuscript. However, we expect to detect strong association in the further when high quality data sets with large sample size become available. As for the HNF1B cell cycle signature, we validated the association with patient prognosis by using meta-analysis by incorporating different data sets with same outcomes, and the meta-analysis results revealed the significant association. We also performed the multivariate analysis to further examine the association of HNF1B cell cycle signature using both the categorical and continuous values with patient survival among a set of clinical variables, which confirmed the robustness of the association.

Despite that “the risk SNPs are not significant after multiple test correction”, we feel there are several reasons for us to make the conclusions. First, the SNPs in HNF1B region are not independent (only 2~3 independent clusters across the gene). Using a Bonferroni corrected P-value with the total number of SNPs ($n=63$) would be too stringent and might raise false negative. A P-value $<0.05/3$ or $0.05/2$ would be a more appropriate corrected cutoff value in the current situation. Second, a series of significant SNPs ($n=3$) were observed in this region which suggested that they were unlikely to be detected by chance. Thirdly, germline-somatic interaction (for example, germline common variants such as SNPs) are usually mild. Despite the strong association between SNPs and the expression of the nearby genes (cis-eQTL), it is very unlikely to observe a similarly strong association between the SNPs with genes at long distances. In the present study, we have provided the evidence that: (1) SNPs in HNF1B regions are associated with expression of HNF1B (eQTL); (2) HNF1B is associated with ERG; (3) SNPs in HNF1B regions are mildly but significantly associated with ERG expression. These evidences are not only from the association study, but also from a series of bioinformatics investigations and functional studies.

Bibliography:

- Howie, B., Fuchsberger, C., Stephens, M., Marchini, J., & Abecasis, G. R. (2012). Fast and accurate genotype imputation in genome-wide association studies through pre-phasing. *Nature Genetics*. <https://doi.org/10.1038/ng.2354>
- Hua, J. T., Ahmed, M., Guo, H., Zhang, Y., Chen, S., Soares, F., Lu, J., Zhou, S., Wang, M., Li, H., Larson, N. B., McDonnell, S. K., Patel, P. S., Liang, Y., Yao, C. Q., van der Kwast, T., Lupien, M., Feng, F. Y., Zoubeydi, A., ... He, H. H. (2018). Risk SNP-Mediated Promoter-Enhancer Switching Drives Prostate Cancer through lncRNA PCAT19. *Cell*. <https://doi.org/10.1016/j.cell.2018.06.014>

- Prada C, Lima D, Nakaya H (2022). *MetaVolcanoR: Gene Expression Meta-analysis Visualization Tool*. R package version 1.10.0.
- Ritchie, M. E., Phipson, B., Wu, D., Hu, Y., Law, C. W., Shi, W., & Smyth, G. K. (2015). Limma powers differential expression analyses for RNA-sequencing and microarray studies. *Nucleic Acids Research*. <https://doi.org/10.1093/nar/gkv007>
- Tierney, J. F., Stewart, L. A., Gherzi, D., Burdett, S., & Sydes, M. R. (2007). Practical methods for incorporating summary time-to-event data into meta-analysis. *Trials*. <https://doi.org/10.1186/1745-6215-8-16>
- Zhu, C. Q., Ding, K., Strumpf, D., Weir, B. A., Meyerson, M., Pennell, N., Thomas, R. K., Naoki, K., Ladd-Acosta, C., Liu, N., Pintilie, M., Der, S., Seymour, L., Jurisica, I., Shepherd, F. A., & Tsao, M. S. (2010). Prognostic and predictive gene signature for adjuvant chemotherapy in resected non-small-cell lung cancer. *Journal of Clinical Oncology*. <https://doi.org/10.1200/JCO.2009.26.4325>

Reviewers' Comments:

Reviewer #5:

Remarks to the Author:

The manuscript by Giannareas et al presents analyses showcasing the role of HNF1B locus in prostate cancer. As a new reviewer, I will focus only on the authors response to reviewer #4 from previous review. Reviewer #4 notes several concerns with statistical rigor that reduce the impact of the claimed advances of the HNF1B association analyses.

In their response, the authors acknowledge the limitations and agree with the reviewer in that association between HNF1B expression with survival across the 9 cohorts is not significant (either in each cohort or in meta-analyses) thus removing these analyses from the revision with appropriate text clarifications. Second, the authors perform follow-up analyses to clarify the association of HNF1B cell-signature score with survival and/or recurrence; the revision shows significant association in meta-analyses thus improving rigor of statistical analyses. Third, the authors do find issues with the genotype data (as noted by reviewer #4) and re-update the QC to obtain accurate genotype calls with reasonable association statistics. Fourth, the authors investigate multiple testing for SNP effects and provide some evidence as why Bonferoni multiple testing should not be used in this case; under a relatively relaxed multiple testing of just 2 tests, 3SNPs remain significant.

Overall, I find the authors addressed reviewer #4 comments adequately while reducing the claims in main text and thus I support the publication. I also encourage the authors to be more deliberate in including the limitations of their study in the discussion section, similar to their response to reviewer #4: "We ... did not observe significant associations between HNF1B expression level and patient survival nor with TS/PSA, which is might be due to limited sample size in current available data sets ..."

NCOMMS-21-23625B: Extensive germline-somatic interplay contributes to prostate cancer progression through HNF1B co-option of TMPRSS2-ERG

REVIEWERS' COMMENTS

Reviewer #5 (Remarks to the Author):

The manuscript by Giannareas et al presents analyses showcasing the role of HNF1B locus in prostate cancer. As a new reviewer, I will focus only on the authors response to reviewer #4 from previous review. Reviewer #4 notes several concerns with statistical rigor that reduce the impact of the claimed advances of the HNF1B association analyses.

In their response, the authors acknowledge the limitations and agree with the reviewer in that association between HNF1B expression with survival across the 9 cohorts is not significant (either in each cohort or in meta-analyses) thus removing these analyses from the revision with appropriate text clarifications. Second, the authors perform follow-up analyses to clarify the association of HNF1B cell-signature score with survival and/or recurrence; the revision shows significant association in meta-analyses thus improving rigor of statistical analyses. Third, the authors do find issues with the genotype data (as noted by reviewer #4) and re-update the QC to obtain accurate genotype calls with reasonable association statistics. Fourth, the authors investigate multiple testing for SNP effects and provide some evidence as why Bonferoni multiple testing should not be used in this case; under a relatively relaxed multiple testing of just 2 tests, 3SNPs remain significant.

Overall, I find the authors addressed reviewer #4 comments adequately while reducing the claims in main text and thus I support the publication. I also encourage the authors to be more deliberate in including the limitations of their study in the discussion section, similar to their response to reviewer #4: "We ... did not observe significant associations between HNF1B expression level and patient survival nor with TS/PSA, which is might be due to limited sample size in current available data sets ... "

Response: We appreciate the reviewer's comments and advice for helping improve the overall manuscript. We have now expanded and discussed the limitations of our study in the discussion section.

Page #17: 'Here we found HNF1B, together with MYC, FOXA1, HOXB13 and AR, as one of the most essential genes for PCa cell survival and its elevated expression in PCa tumors accompanied with higher grade and Gleason score. Despite the fact that we did not observe significant association between HNF1B expression and patient prognosis, which might be due to the limited sample size in current available data sets, we expect to detect strong association with the availability of large patient cohorts in the future.'